# GWL_FCS30: global 30 m wetland map with fine classification system using multi-sourced and time-series remote sensing imagery in 2020

Xiao Zhang[1, 2], Liangyun Liu[1, 2, 3, *], Tingting Zhao[1, 4], Xidong Chen[5], Shangrong Lin[6], Jinqing Wang[1, 2, 3], Jun Mi[1, 2, 3] and Wendi Liu[1, 2, 3]

5 1 International Research Center of Big Data for Sustainable Development Goals, Beijing 100094, China

2 Key Laboratory of Digital Earth Science, Aerospace Information Research Institute, Chinese Academy of Sciences, Beijing 100094, China

3 School of Electronic, Electrical and Communication Engineering, University of Chinese Academy of Sciences, Beijing 100049, China

10 4 College of Geomatics, Xi'an University of Science and Technology, Xi'an 710054, China

5 North China University of Water Resources and Electric Power, Zhengzhou 450046, China

School of Atmospheric Sciences, Southern Marine Science and Engineering Guangdong Laboratory (Zhuhai), Sun Yat-sen University, Zhuhai 519082, Guangdong, China

* Corresponding author Email: liuly@radi.ac.cn

**Abstract**

Wetlands, often called the "kidneys of the earth", play an important role in maintaining ecological balance, conserving water resources, replenishing groundwater, and controlling soil erosion. Wetland mapping is very challenging because of its complicated temporal dynamics and large spatial and spectral heterogeneity. An accurate global 30-m wetland dataset that can simultaneously cover inland and coastal zones is lacking. This

study proposes a novel method for wetland mapping by combining an automatic sample extraction method, multisource existing products, time-series satellite images, and a stratified classification strategy. This approach allowed for the generation of the first global 30-m wetland map with a fine classification system (GWL_FCS30), including five inland wetland sub-categories (permanent water, swamp, marsh, flooded flat, and saline) and three coastal tidal wetland sub-categories (mangrove, salt marsh, and tidal flats), which was developed using

Google Earth Engine platform. We first combined existing multi-sourced global wetland products, expert knowledge, training sample refinement rules, and visual interpretation to generate a large and geographically distributed wetland training samples. Second, we integrated the time-series Landsat reflectance products and Sentinel-1 SAR imagery to generate various water-level and phenological information to capture the complicated temporal dynamics and spectral heterogeneity of wetlands. Third, we applied a stratified

classification strategy and the local adaptive random forest classification models to produce the wetland dataset with a fine classification system at each $5° \times 5°$ geographical tile in 2020. Lastly, the GWL_FCS30, mosaicked by 961 $5° \times 5°$ regional wetland maps, was validated using 25,708 validation samples, which achieved an overall accuracy of 86.44% and a kappa coefficient of 0.822. The cross-comparisons with other global wetland products demonstrated that the GWL_FCS30 dataset performed better in capturing the spatial patterns of wetlands and

had significant advantages over the diversity of wetland subcategories. The statistical analysis showed that the global wetland area reached 6.38 million $km^2$, including 6.03 million $km^2$ of inland wetlands and 0.35 million $km^2$ of coastal tidal wetlands, approximately 72.96% of which were distributed poleward of 40°N. Therefore, we can conclude that the proposed method is suitable for large-area wetland mapping and that the GWL_FCS30 dataset is an accurate wetland mapping product that has the potential to provide vital support for wetland

management. The GWL_FCS30 dataset in 2020 is freely available at https://doi.org/10.5281/zenodo.7340516 (Liu et al. 2022).

## 1. Introduction

The Ramsar Convention defines a wetland as an "areas of marsh, fen, peatland or water, whether natural or artificial, permanent or temporary, with water that is static or flowing, fresh, brackish or salt, including areas of marine water the depth of which at low tide does not exceed six meters" (Gardner and Davidson, 2011). Wetlands not only provide humans with a large amount of food, raw materials and water resources (Ludwig et al., 2019; Zhang et al., 2022b) but also play an important role in maintaining ecological balance, conserving water resources, replenishing groundwater, and controlling soil erosion (Hu et al., 2017a; Mao et al., 2021; Wang et al., 2020; Zhu and Gong, 2014). Therefore, they are also called the "kidneys of the earth" (Guo et al., 2017). However, due to increasing human activities, including agriculturalization, industrialization and urbanization (McCarthy et al., 2018; Xi et al., 2020), and climatic changes such as sea-level rise and coastal erosion (Cao et al., 2020; Wang et al., 2021), wetlands have been seriously degraded and threatened over the past few decades (Mao et al., 2020). Thus, having access to timely and accurate wetland mapping information is pivotal for protecting biodiversity and supporting the sustainable development goals.

Along with the rapid development of remote sensing techniques and computing abilities, a variety of regional or global wetland datasets have been produced with spatial resolutions ranging from 30 m to 1° (~112 km) (Chen et al., 2022; Gumbricht et al., 2017; Lehner and Döll, 2004; Mao et al., 2020; Matthews and Fung, 1987; Tootchi et al., 2019). Recently, Tootchi et al. (2019) and Hu et al. (2017a) have systematically reviewed the generation process of global wetland datasets with various spatial and temporal resolutions and wetland categories and found significant uncertainties and inconsistencies among these datasets. For example, the global total wetland area reviewed by Hu et al. (2017a) ranged from 2.12 to 7.17 million $km^2$ based on remote sensing products. Therefore, great uncertainties among global wetland datasets directly hindered wetland applications and analysis. Furthermore, from the perspective of spatial resolution, although many wetland products have been produced, at regional or global scales, using various remote sensing imagery and different methods (Guo et al., 2017; Tootchi et al., 2019), most of them were coarse spatial resolution datasets, ranging from 100 m to 25 km. Recently, with the improvement of computing power and storage abilities, three global 30-m land-cover products (including GlobeLand30 (Chen et al., 2015), FROM_GLC (Gong et al., 2013) and GLC_FCS30 (Zhang et al., 2021c)) and several 10-m land-cover products (WorldCover (Zanaga et al., 2021), Dynamic World (Brown et al., 2022) and FROM_GLC10 (Gong et al., 2019)), containing an independent wetland layers, were produced, but their classification algorithms were not specifically designed for the wetland environment, so the wetland usually suffered from low accuracy in these products. In addition, several global coastal tidal wetland products have been developed, including the global mangrove extent (Bunting et al., 2018; Hamilton and Casey, 2016) and global 30 m tidal flat datasets from 1984 to 2016 (Murray et al., 2019), but these only covered the intertidal zones. Thus, an accurate global 30 m thematic wetland dataset, with fine wetland categories and covering both inland and coastal zones, is still lacking.

One of the largest challenges of current state-of-the-art methods for large-area wetland mapping is to collect massive amount of training samples (Liu et al., 2021; Ludwig et al., 2019). Zhang et al. (2021b) mentioned two options for collecting training samples, including the visual interpretation method and deriving training samples from pre-existing products. First, since the visual interpretation method had significant advantage over the confidence of training samples, it was widely used for local or regional wetland mapping (Amani et al., 2019; Wang et al., 2020). However, collecting accurate and sufficient training samples is usually a time-consuming process and involves a large amount of manual work, so it was impractical and nearly impossible to use the

visual interpretation for collecting global wetland samples. Comparatively, deriving training samples from existing products and applying some rules or refinement methods to identify these high confidence samples from existing products shows promise (Zhang et al., 2021c). So this approach is practical in that it could quickly large and geographically diverse distribution of training samples without much manual effort. Thus, the second option attached increasing attention and has been successfully used for large-area land-cover mapping (Zhang and Roy, 2017; Zhang et al., 2021c; Zhang et al., 2020). For example, Zhang et al. (2021b) used derived global training samples from the combination of the CCI_LC and MCD43A4 NBAR datasets to produce a global 30-m land-cover product with a fine classification system in 2015 and 2020 (GLC_FCS30) with an overall accuracy of 82.5%. Therefore, if we take effective measures to fuse these existing products and then derive high confidence training samples using some refinement rules, the deriving approach would exude great potential for global wetland mapping.

Another major challenge inherent to wetland mapping is the complicated temporal dynamics and spatial and spectral heterogeneity. The spectral characteristics of the wetlands would quickly change with the seasonal or daily water levels of the underlying surface (Ludwig et al., 2019; Mahdianpari et al., 2020). Therefore, many studies proposed to combine multi-sourced, time-series remote sensing imagery for capturing the spatial extent and temporal dynamics of wetlands (LaRocque et al., 2020; Ludwig et al., 2019; Murray et al., 2019; Wang et al., 2021; Zhang et al., 2022b). For example, Zhang et al. (2022b) and Murray et al. (2019) used the time-series Landsat imagery to generate tidal-level and phenological features for identifying coastal tidal wetlands and successfully produced the coastal tidal wetlands in China with an overall accuracy of 97.2% (Zhang et al., 2022b) and global trajectory tidal flats with the overall map accuracy of 82.3% (Murray et al., 2019). Except for optical imagery, synthetic aperture radar (SAR) data, which was sensitive to soil moisture, vegetation structure, and inundation, enabled data acquisition regardless of solar illumination, clouds, or haze and was also widely used for wetland mapping, especially after the open-access of Sentinel-1 data became available (Li et al., 2020; Slagter et al., 2020; Zhang et al., 2018). For example, Li et al. (2020) used the time-series Sentinel-1 imagery to discriminate wetlands with and without trees and achieved an overall accuracy of 86.0±0.2%. Therefore, the fusion of multi-sourced and time-series remote sensing imagery is vital for accurate wetland mapping.

Due to the complicated temporal dynamics and spatial and spectral heterogeneity of wetlands, there is very few global thematic wetland dataset covering both inland and coastal regions with fine classification system and high spatial resolution, which also cause that global 30 m wetland mapping with a fine classification system remains a challenging task. In this study, we combined several existing wetland products and multi-sourced time-series remote sensing imagery to (1) derive a large and geographically distributed wetland training samples from multi-sourced pre-existing global wetland products to minimize the manual participation; (2) develop a robust method to capture the temporal dynamics of wetlands and then produce the first global 30-m wetland dataset with a fine classification system (GWL_FCS30); (3) quantitatively analyze the spatial distribution of different wetland categories and assess the accuracy of the GWL_FCS30 in 2020.

## 2. Datasets

### 2.1 Multi-sourced remote sensing imagery

Three types of remote sensing imagery were collected to capture the temporal dynamics and spatial and spectral heterogeneity of wetlands. These include Landsat optical data, Sentinel-1 SAR, and ASTER GDEM topographical data. First, all available Landsat imagery, including Landsat 7 ETM+ and Landsat 8 OLI missions,

during 2019–2021 was obtained for the nominal year of 2020 via the Google Earth Engine platform for minimizing the influence of frequent cloud contamination in the tropics and snow and ice in the high latitudes. To minimize the effect of atmosphere, each Landsat image was atmospherically corrected to the surface reflectance by the United States Geological Survey using Land Surface Reflectance Code (LaSRC) method (Vermote et al., 2016) and then archived on the GEE platform. And these 'bad quality' observations (shadow, cloud, snow, and saturated pixels) in Landsat imagery were masked using CFmask cloud detection method, which built a series of decision rules, using temperature, spectral variability, brightness and geometric relationship between cloud and shadow, to identify these 'poor quality' pixels and achieved the overall accuracy of 96.4% (Zhu et al., 2015; Zhu and Woodcock, 2012). In this study, six optical bands, including: blue, green, red, NIR (near infrared), SWIR1 (shortwave infrared 1) and SWIR2 (shortwave infrared 2) bands, were used for wetland mapping. Totally, 764,239 Landsat scenes were collected to capture various water-level and phenological features according to the spectral characteristics of various land-cover types, presented in Section 4. Figure 1a illustrates the spatial distribution of all clear-sky observations for all Landsat scenes, and it can be seen that there were more than 10 clear observations after masking these 'poor quality' observations at each region even if in the tropics.

Then, the Sentinel-1 SAR data, which was demonstrated to be sensitive to the soil moisture, vegetation structure, and inundation information (Li et al., 2020), used dual-polarization C-band backscatter coefficients to measure the incident microwave radiation scattered by the land surface (Torres et al., 2012). This study obtained the time-series Sentinel-1 imagery archived on the GEE platform in 2020 in Interferometric Wide Swath mode with a dual-polarization of VV and VH. Notably, all Sentinel-1 SAR imagery on the GEE platform has been pre-processed by the Sentinel-1 Toolbox with thermal noise removal, radiometric calibration, and terrain correction using 30-m elevation data (Veci et al., 2014). Figure 1b also illustrates the spatial distribution of all available Sentinel-1 SAR imagery, there were enough Sentinel-1 SAR observations in each area to capture the water-level dynamics of wetlands because it was immune to the cloud and shadow and had a revisit time of 6 days after launching the Sentinel-1B mission. Lastly, as many studies have demonstrated that the topography would directly affect the spatial distribution of wetlands, which are mainly distributed in low-lying areas (Hu et al., 2017b; Ludwig et al., 2019; Tootchi et al., 2019), the ASTER GDEM elevation and derived slope and aspect were used as auxiliary information for wetland mapping. It had a spatial resolution of 30 m and covered the entire global land area (Tachikawa et al., 2011a). Quantitative assessment indicated that the GDEM achieved an absolute vertical accuracy of 0.7 m over bare areas and 7.4 m over forested areas (Tachikawa et al., 2011b).

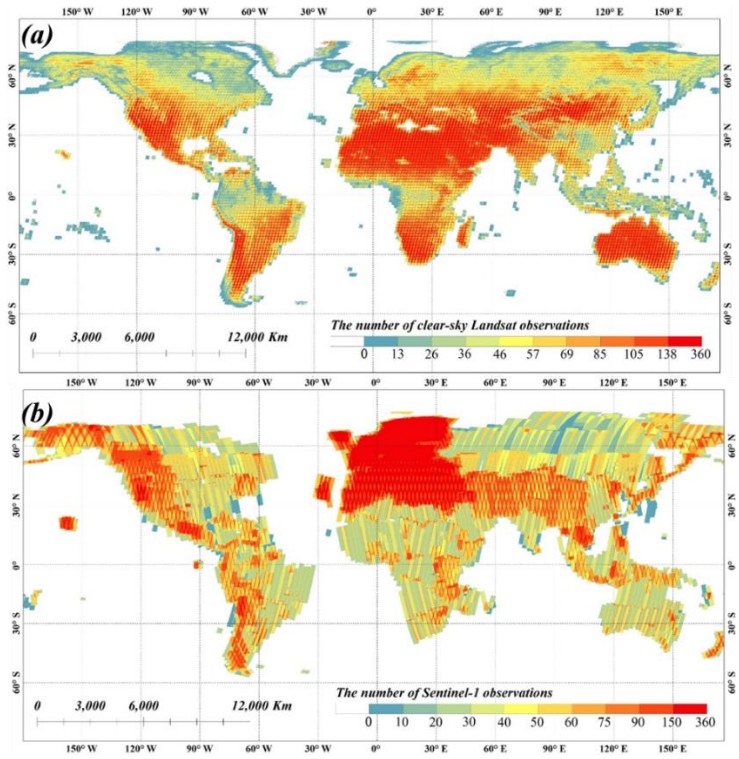

**Figure1**. The spatial distribution of clear observations after masking these 'poor quality' observations during 2019-2021 (a), and availability of time-series Sentinel-1 SAR observations in 2020 (b).

## 2.2 Global prior wetland datasets

To achieve the goal of deriving a large and geographically diverse distribution of training samples with minimum manual labor, we propose combining various prior global wetland datasets for generating high-confidence training samples. Table 1 lists the characteristics of several global wetland datasets. Specifically, we collected five global mangrove forest products with different spatial resolutions and time spans, and all of them achieved desirable accuracy. For example, the Global Mangrove Watch (GMW) was validated to reach an overall accuracy of 95.25%, and the user and producer accuracies of mangrove forest were 97.5% and 94.0%, respectively (Thomas et al., 2017). Furthermore, to derive the samples of salt marsh and tidal flats, we collected the time-series global 30-m tidal flats products from 1984 to 2016 with an interval of three years, achieving an overall map accuracy of 82.3% (Murray et al., 2019). The global salt marsh dataset, containing 350,985 individual occurrence polygon shapefiles, helped generate the global salt marsh estimation (McOwen et al., 2017).

Except for the coastal tidal wetland products, two thematic wetland products (TROP-SUBTROP Wetland and GLWD contained various wetland sub-categories), three global land-cover products (GlobeLand30, GLC_FCS30, and CCI_LC contained an independent layer), and the time-series 30-m water dynamic dataset (JRC_GSW) were combined to determine the inland maximum wetland extents and generate the wetland training samples after using a series of refinement rules given in Section 3. Specifically, the TROP-SUBTROP was produced by combining the hydrological model and annual time series of soil moisture, mainly covering the tropics and sub-tropics (40°N ~ 60°S) with a resolution of 231 m (Gumbricht, 2015). The GLWD, combining the GIS functionality and a variety of existing maps and information, was developed with 12 wetland sub-categories at a resolution of 1 km (Lehner and Döll, 2004). The JRC_GSW dynamic water dataset achieved a

producer accuracy of 98.5% for these seasonal waters (Pekel et al., 2016) and was used to identify inundated pixels. Furthermore, three global land-cover products, simultaneously containing wetland layer and non-wetland land-cover types, were used to determine the non-wetland samples and then served as the auxiliary datasets to improve the confidence of inland wetland samples.

Table 1. The characteristics of 13 global wetland products with various spatiotemporal resolutions (unit of area: million km$^2$)

| Dataset name and reference | Wetland categories | Year | Resolution | Total area | Coverage |
|---|---|---|---|---|---|
| World atlas of mangroves (WAM) Spalding (2010) | Mangrove | 2010 | 1:1000000 | 0.152 | Global |
| Global mangrove watch (GWM) Thomas et al. (2017) | | 1996-2016 | ~25m | ~0.136 | Global |
| A global biophysical typology of mangroves (GBTM) Worthington et al. (2020) | | 1996-2016 | ~25m | ~0.136 | Global |
| Continuous global mangrove forest cover (CGMFC) Hamilton and Casey (2016) | | 2000-2010 | 30 m | 0.083 | Global |
| Global distribution of mangroves USGS (GDM_USGS) Giri et al. (2011) | | 2011 | 30 m | ~0.138 | Global |
| Global distribution of tidal flat ecosystems Murray et al. (2019) | Tidal flat | 1984-2016 | 30 m | 0.124~0.132 | 60°S~60°N |
| Global distribution of saltmarsh McOwen et al. (2017) | Salt marsh | 1973-2015 | 1:10,000 | ~0.05 | Global |
| Tropical and subtropical wetland distribution (CIFOR) Gumbricht (2015) | Open water, mangrove, swamps, fens, riverine, floodplains, marshes | 2011 | ~231 m | 4.7 | 60°S~40°N |
| Global lakes and wetlands database (GLWD) Lehner and Döll (2004) | Lake, reservoir, river, marsh, swamps, coastal tidal wetland, saline wetland, and peatland | 2004 | ~1 km | 10.7~12.7 | Global |
| JRC-GSW Pekel et al. (2016) | Water | 1984-2021 | 30 m | ~4.46 | Global |
| ESA CCI_LC Defourny et al. (2018) | Swamps, mangrove, and Shrub or herbaceous cover wetlands | 1992-2020 | 300 m | 6.1 | Global |
| GlobeLand30 Chen et al. (2015) | Wetland | 2000-2020 | 30 m | 7.01~7.17 | Global |
| GLC_FCS30 Zhang et al. (2021b) | Wetland | 2015, 2020 | 30 m | 6.36 | Global |

## 2.3 Global 30 m tree cover product

The global 30-m forest cover change in tree cover (GFCC30TC) data in 2015 was produced by downscaling the 250-m MODIS VCF (Vegetation Continuous Fields) tree cover product using Landsat imagery and then incorporating the MODIS cropland layer to guarantee the tree cover accuracy in agricultural areas (Sexton et al., 2016; Sexton et al., 2013). This product was used to accurately distinguish between inland swamp and marsh wetlands because both of them reflected obvious vegetation spectra characteristics. It was validated to achieve an overall accuracy of 91%; the average producer and user accuracy for stable forests were 92.5% and 95.4%, respectively (Sexton et al., 2016; Townshend et al., 2012).

## 2.4 National wetland products

Three national wetland products including: NLCD (National Land Cover Database) (Homer et al., 2020), NWI (National Wetlands Inventory) (Wilen and Bates, 1995) and CLC (CORINE Land Cover) (Büttner, 2014), were used as the comparative datasets to analyze the performance of developed global wetland maps in Section 6.2. Specifically, the NLCD contained open water, woody wetlands and emergent herbaceous wetlands, the NWI contained eight sub-categories (estuarine and marine deep-water, estuarine and marine wetland, freshwater emergent wetland, freshwater forest/shrub wetland, freshwater pond, lake, other, and Riverine), and the CLC identified the wetlands into nine sub-categories as: inland marshes, peat bogs, salt marshes, saline, intertidal flats, water courses, water bodies, coastal lagoons, estuaries, as well as sea and oceans.

## 3. Collecting training samples and determining maximum wetland extents

In this study, after considering the applicability of moderate resolution (10–30 m) imagery, their practical use for ecosystem management, and the available pre-existing global wetland dataset, the fine wetland classification system, containing eight sub-categories (three coastal tidal sub-categories and five inland sub-categories), was proposed to comprehensively depict the spatial patterns of global wetlands (Table 2). Specifically, the sub-categories of coastal tidal wetlands consist of mangroves, salt marshes, and tidal flats. By importing the vegetation and water cover information associated with this land cover, these categories were widely recognized in many previous studies (Wang et al., 2021; Zhang et al., 2022b). The inland wetland types shared similar characteristics and were grouped into swamp, marsh, and flooded flat. Meanwhile, in order to capture saline soils and halophytic plant species along saline lakes, the inland saline wetland, inherited from the Global Lakes and Wetlands Dataset (GLWD) (Lehner and Döll, 2004), was also imported. Lastly, the permanent water, including lakes, rivers and streams that are always flooded, was widely identified as a wetland layer in previous studies (Davidson, 2014; Dixon et al., 2016; Hu et al., 2017b) and was also added into our fine wetland classification system.

**Table 2**. The description of wetland classification system in this study

| Category I | Category II | Description |
|---|---|---|
| Tidal wetland | Mangrove | The forest or shrubs which grow in the coastal blackish or saline water |
| | Salt marsh | Herbaceous vegetation (grasses, herbs and low shrubs) in the upper coastal intertidal zone |
| | Tidal flat | The tidal flooded zones between the coastal high and low tide levels including mudflats and sandflats. |
| Inland wetland | Swamp | The forest or shrubs which grow in the inland freshwater |
| | Marsh | Herbaceous vegetation (grasses, herbs and low shrubs) grows in the freshwater |
| | Flooded flat | The non-vegetated flooded areas along the rivers and lakes |
| | Saline | Characterized by saline soils and halophytic (salt tolerant) plant species along saline lakes |
| | Permanent water | Lakes, rivers and streams that are always flooded |

Many studies have explained that the quality and confidence of training samples directly affected the classification performance (Zhang et al., 2021b; Zhu et al., 2016). The previously mentioned process of collecting sufficient training samples via visual interpretation was time-consuming and involved a lot of manual labor. Fortunately, a variety of regional and global wetland products have been developed and released over the past few decades (Table 1), and many studies have demonstrated that deriving training samples from existing

products could be used for large-area classification and mapping (Huang et al., 2021; Zhang et al., 2021b). Therefore, we propose to combine existing global wetland datasets to independently derive coastal/inland wetland training samples and their maximum distribution extents (Figure 2).

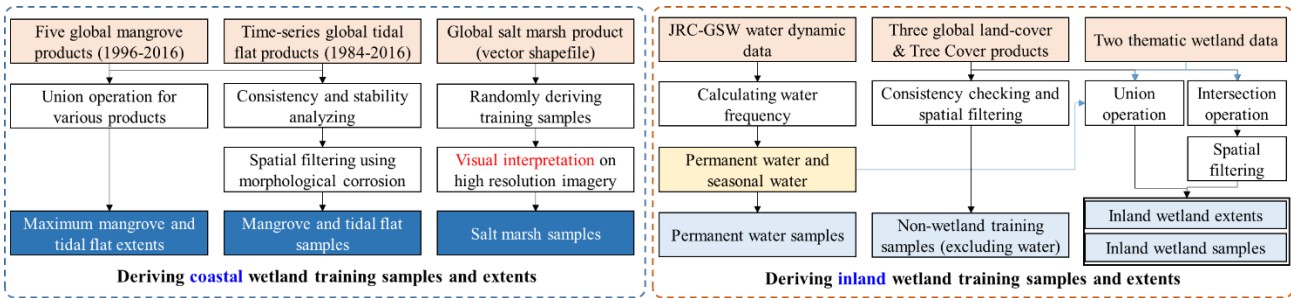

**Figure 2**. The flowchart of deriving coastal and inland wetland samples from multiple pre-existing datasets

**3.1 Deriving coastal tidal wetland training samples and maximum extents**

        This study divided the coastal tidal wetlands into three sub-categories: mangrove forest, salt marsh, and tidal flat. The previously existing products have been collected in Table 1. For the mangrove training samples, we collected five global mangrove products with different spatiotemporal resolutions, all of which achieved fulfilling performances. For example, Hamilton and Casey (2016) stated that their continuous mangrove forest

cover (CGMFC) dataset could cover 99% of all mangrove forests from 2000 to 2012, and Thomas et al. (2017) validated their Global Mangrove Watch (GMW) products from 1996 to 2016 and reached an overall accuracy of 95.25%. Therefore, we first measure the temporal consistency of the three time-series mangrove forest products (CGMFC, GMW, and GBTM mangroves), and only these temporally stable mangrove forest pixels were selected as the primary candidate points ($P_{mangrove}^{Tstable}$). Meanwhile, to minimize the influence of

classification error in each mangrove forest product, the cross-consistency of five mangrove products was analyzed, and only the pixel, simultaneously identified as mangrove forest in all five products, was labeled as stable and consistent candidate points ($P_{mangrove}^{Tstable,Scons}$). Furthermore, considering that there was a temporal interval between prior mangrove products and our study, and that mangrove deforestation usually followed the pattern of edge-to-center contraction, a morphological erosion filter with a local window of 3×3 was applied to

the $P_{mangrove}^{Tstable,Scons}$ points to further ensure the confidence of mangrove training samples. Lastly, as for the maximum mangrove forest extents ($MaxExtent_{mangrove}$), the union operation was applied to five global mangrove products as shown in Eq. (1).

$$MaxExtent_{mangrove} = M_{WAM} \cup M_{GMW} \cup M_{GBTM} \cup M_{CGMFC} \cup M_{GDM\_USGS} \qquad (1)$$

where $[M_{WAM}, M_{GMW}, M_{GBTM}, M_{CGMFC}, M_{GDM\_USGS}]$ are the spatial distributions of five global mangrove forest products listed in Table 1. It should be noted that these prior mangrove products were demonstrated to

cover almost all mangroves over the world, so the $MaxExtent_{mangrove}$ can be used as the boundary for mangrove mapping; namely, only the pixel within the maximum mangrove extent was labeled as mangrove forest.

Regarding the collection of tidal flat samples, the prior time-series global 30 m tidal flat products ($Gtidalflat$) from 1984 to 2016 were validated to achieve an overall map accuracy of 82.3%, and user accuracies for the non-tidal and tidal flat of 83.3% and 81.1%, respectively (Murray et al., 2019). To ensure the accuracy of tidal flat samples, we first applied temporal consistency analysis to the time series of tidal flat datasets from 2000 to 2016 and identified the temporally stable tidal flat pixels ($P_{tidal}^{Tstable}$) during 16 consecutive years. The reason why we discarded the tidal flat datasets before 2000 was that the available Landsat imagery were sparse and could not accurately capture the high-tidal and low-tidal information, and suffered lower monitoring accuracy. Next, Radoux et al. (2014) found that transition zones between two different land-cover types are likely to be misclassified; therefore, the candidate tidal flat samples $P_{tidal}^{Tstable}$ were further refined by the morphological erosion filter with a local window of 3×3. Furthermore, as a tidal flat is a non-vegetated coastal tidal wetland, we combined the empirical rule (EVI ≥ 0.1, NDVI ≥ 0.2, and LSWI > 0) proposed by Wang et al. (2020) and time-series Landsat imagery in 2020 (approximately 142 thousand Landsat scenes) to exclude all vegetated pixels from tidal flat training samples. Lastly, to derive the maximum tidal flat extents ($MaxExtent_{tidalflat}$), the union operation was applied to the time-series tidal flat products from 1984 to 2016. It should be noted that the Murray's global 30 m tidal flat datasets only covered the regions of 60°N~60°S (Murray et al., 2019), therefore, we used the coastal shorelines ($Line_{coastal}$) to create a 50 km buffer (applied by the Wang et al. (2020) and (Murray et al., 2019)) as the potential tidal flat zones in the high latitude regions (>60°N) as in Eq. (2). It should be noted that we only identified and then retained these tidal flat pixels within the maximum extents by using the classification models in the Section 4.2.

$$MaxExtent_{tidalflat} = \begin{cases} \cup_{t=1984}^{2016} Gtidalflat_{t,s}, & s \in [60°S, \ 60°N] \\ Line_{coastal} \pm 50km, & s \in [60°N, \ 90°N] \end{cases} \tag{2}$$

Compared with the mangrove forest and tidal flat, the pre-existing global or regional salt marsh products were relatively sparse. The global distribution of the salt marsh dataset contained 350,985 individual vector polygons and was the most complete dataset on salt marsh occurrence and extent at the global scale (McOwen et al., 2017). However, after careful review, we found some mislabeled salt marsh polygons, so this dataset cannot be used directly to derive training samples. This study first used the random sampling method to generate 35,099 salt marsh points (approximately 10% of the total polygons) based on prior datasets. We combined the visual interpretation method and high-resolution imagery to check each salt marsh point. After discarding the incorrect and uncertain samples, a total of 32,712 salt marsh points were retained. However, the prior dataset only captured the extent of salt marshes in 99 countries worldwide (McOwen et al., 2017), further noting that the distribution of salt marshes was spatially correlated with tidal flat and mangrove forest (Wang et al., 2021). Consequently, the maximum extents of tidal flat and mangrove forest, in addition to the prior salt marsh extent were used for salt marsh mapping. Meanwhile, as the wetland layer in the global land-cover products (GLC_FCS30, GlobeLand30, and CCI_LC) also covered some coastal tidal wetlands, the saline-water wetland layer in the CCI_LC and the wetland data in other two products closed to the coastal shorelines were also imported as supplement when determining the maximum coastal tidal wetland extents.

### 3.2 Deriving inland wetland training samples and maximum extents

The pre-existing inland wetland datasets usually suffered from lower accuracy compared to coastal tidal wetland products; for example, the wetland layer in the GlobeLand30-2010 and GLC_FCS30-2015 was validated to achieve a user accuracy of 74.9% (Chen et al., 2015) and 43.4% (Zhang et al., 2021b), respectively. Therefore, we first generated high-confidence inland wetland samples and then determined their sub-categories

(swamp, marsh, inland flat, saline wetland and permanent water). Specifically, the consistency analysis of five global wetland datasets (TROP-SUBTROP Wetland, GLWD, CCI_LC, GlobeLand30, and GLC_FCS30) and the temporal stability checking for CCI_LC (1992–2020), GlobeLand30 (2000-2020) and GLC_FCS30 (2015-2020) were applied to identify these temporally stable and high cross-consistency wetland points ($P_{inlandWet}^{Tstable,Scons}$). It should be noted that the coarse wetland products (GLWD, TROP-SUBTROP and CCI_LC) were resampled to 30 m using the nearest neighbor method on the GEE platform and the coastal tidal wetland layers in these products were excluded. Namely, only the pixel identified as inland wetland in all five products was retained. Then, the morphological erosion filter with a local window of $3 \times 3$ was also used to decrease the sampling uncertainty over these land-cover transition areas because the transition zones between two different land-cover types are likely to be misclassified (Lu and Wang, 2021; Radoux et al., 2014).

Afterward, to determine the wetland sub-category for each inland wetland sample, we first used the empirical vegetation rule (EVI ≥ 0.1, NDVI ≥ 0.2, and LSWI > 0) proposed by Wang et al. (2020) and time-series Landsat imagery to split candidate samples into two parts: vegetated wetland samples (swamp and marsh) and non-vegetated wetland samples (flooded flat, saline and permanent water). Then, as the swamp was defined as the forest or shrubs which grow in the inland freshwater, the global 30-m tree cover dataset (GFCC30TC) was adopted to distinguish the swamp and marsh from vegetated wetland samples. Specifically, if the tree cover of the sample was greater than 30% (Hansen et al., 2013), it was labeled as swamp, and the remaining vegetated wetland samples were labeled as marsh. Furthermore, to distinguish between the inland flat, saline samples and permanent water, the saline blocks in the prior GLWD products were first checked by visual interpretation and then imported as the reference dataset to identify all saline wetland samples. The remaining non-vegetated wetland samples were further refined using the time series of the JRC-GSW datasets, only water probability of these remaining samples less than the threshold of 0.95 (suggested by Wang et al. (2020)) were labeled as flooded flat. Lastly, regarding the permanent water samples, the JRC_GSW water dynamic dataset was validated and achieved producer's and user's accuracies of 99.7% and 99.1% for permanent water (Pekel et al., 2016). The permanent water training samples were directly derived from the JRC_GSW dataset without any refinement rules.

Lastly, as for determining the maximum inland wetland extents ($Mextent_{inWet}$), the union operation was conducted to six pre-existing global wetland datasets as in Eq. (3).

$$Mextent_{inWet} = W_{\text{TROP–SUBTROP}} \cup W_{GLWD} \cup W_{CCI\_LC} \cup W_{GLC\_FCS30} \cup W_{Globeland30} \cup W_{JRC\_GSW} \qquad (3)$$

Here, $[W_{\text{TROP–SUBTROP}}, W_{GLWD}, W_{CCI\_LC}, W_{GLC\_FCS30}, W_{Globeland30}]$ were wetland distributions of five pre-existing global wetland products, and $W_{JRC\_GSW}$ was JRC-GSW time-series water dynamic datasets, which identified the inundated probability at a monthly history during 1984-2021(Pekel et al., 2016). It should be noted that the omission error can be ignored for derived maximum inland wetland extents ($Mextent_{inWet}$), because the GLWD and TROP-SUBTROP wetland datasets captured almost all potential wetlands using compilation and model simulation methods (Gumbricht, 2015; Lehner and Döll, 2004).

### 3.3 Deriving non-wetland training samples from prior land-cover products

Except for inland and coastal tidal wetland samples, the non-wetland samples were also necessary because some non-wetland land-cover types were shown to have a similar spectrum to wetlands. For example, swamp and forest or shrubs exhibited the same vegetation reflectance characteristics in optical imagery, and marsh and grassland shared similar spectra curves during the growing season (Zhang et al., 2022b). Except for eight fine

wetland sub-categories training samples, we also divided the non-wetlands into forest/shrubland, grassland, cropland, and others (bare land, impervious surfaces, and snow). To automatically derive these non-wetland samples, the multi-epochs GlobeLand30, GLC_FCS30 and CCI_LC global land-cover products were integrated. Specifically, the temporal stability and cross-consistency analysis were applied to three land-cover products to identify temporally stable forest/shrubland, grassland, cropland, and other candidate samples. Furthermore, the morphological erosion filter with the local window of $3 \times 3$ was also adopted to decrease the sampling uncertainty over land-cover transition areas.

## 3.4 Determining the sample size and distributions using stratified random sampling strategy

Except for the confidence of training samples, many studies also found that the size and distribution of training samples also affected classification performances (Jin et al., 2014; Zhu et al., 2016). As this study aimed to identify wetlands instead of all land-cover types, the equal allocation sample distribution would perform better than the proportional distribution (the sample size determined by the area) (Jin et al., 2014; Zhang et al., 2020). Namely, the approximate proportion of inland wetland, coastal tidal wetland, and non-wetland samples was 5:3:4 in the coexisting areas because the classification system was composed of five inland and three coastal tidal wetland sub-categories and four non-wetland land-cover types. Regarding the sample size, Zhu et al. (2016) had analyzed the quantitative relationships of sample size and the mapping accuracy and found that the mapping accuracies slowly increased and then remained stable with any further increase in the number of samples and suggested using a total of 20,000 samples in the Landsat scene. In this study, we used the stratified random sampling strategy to collect the training samples (excluding salt marsh because it was collected globally using visual interpretation in Section 3.1) at each $5° \times 5°$ geographical grid (corresponding to the local adaptive modeling in the Section 4.2) using an approximate sample size of 2000 for each category. According to our statistics, this study derived exceeding 20 million training samples for mapping global fine wetlands.

## 4. Mapping wetland using the stratified classification strategy and the water-level, phenological features

Figure 3 illustrates the flowchart of the proposed method for generating the global 30-m fine wetland maps. First, we combined the time-series Landsat-8, Sentinel-1 SAR observations and ASTER DEM topographical image to derive multisource and multitemporal features including: various water-level, phenological and three topographical features. Then, the training samples (coastal tidal, inland wetlands and no-wetlands) and derived multisource and multitemporal features were combined to train the stratified random forest classifiers (a classic and widely used machine learning classification model (Breiman, 2001)) at each local region. Next, using the trained random forest models and derived multisource and multitemporal features, we could develop corresponding coastal tidal wetland and inland wetland maps. Finally, the post-processing step was used to generate the global 30 m fine wetland map in 2020.

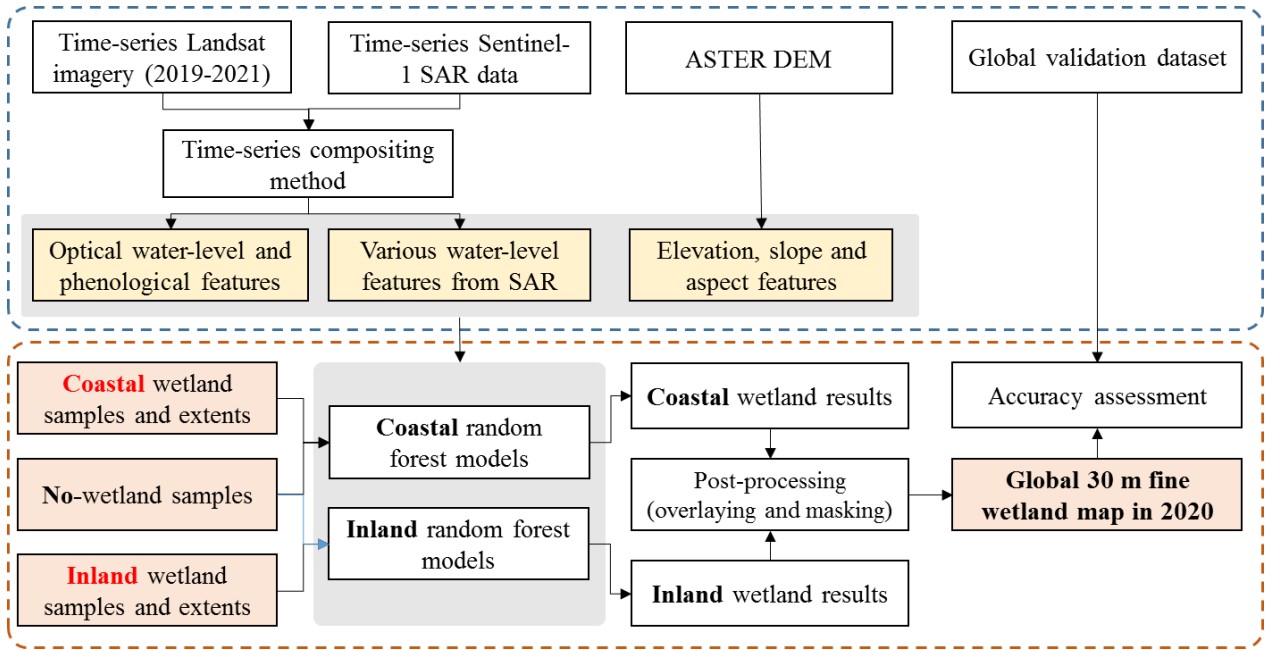

**Figure 3**. The flowchart of wetland mapping using water-level, phenological and topographical features and a stratified classification strategy.

## 4.1 Generating various water-levels and phenological composites

Before generating various water-level and phenological features, four spectral indexes including normalized difference water index (NDWI), land surface water index (LSWI), normalized difference vegetation index 365 (NDVI) and enhanced vegetation index (EVI) were imported because many studies have demonstrated that they were of great help in wetland mapping (Mao et al., 2020; Wang et al., 2020),

$$LSWI = \frac{\rho_{nir} - \rho_{swir1}}{\rho_{nir} + \rho_{swir1}}, NDWI = \frac{\rho_{green} - \rho_{swir1}}{\rho_{green} + \rho_{swir1}}, NDVI = \frac{\rho_{nir} - \rho_{red}}{\rho_{nir} + \rho_{red}}, EVI = 2.5 \times \frac{\rho_{nir} - \rho_{red}}{\rho_{nir} + 6 \times \rho_{red} - 7.5 \times \rho_{blue} + 1} \quad (4)$$

where $\rho_{blue}$, $\rho_{green}$, $\rho_{red}$, $\rho_{nir}$, $\rho_{swir1}$ were the blue, green, red, near-infrared and shortwave infrared bands of Landsat imagery, respectively.

Then, the spectral characteristics of the wetlands would quickly change along with the seasonal or daily 370 water levels of the underlying surface. For example, the tidal flat was the status of seawater at the high tidal stage and mud or sand flats at low tidal stages (Wang et al., 2021); therefore, it was necessary to extract the highest and lowest water-level composites to completely capture these inundated wetlands. Over the past several years, the time-series compositing strategy has been widely used to capture phenological and cloud-free composites (Jia et al., 2020; Ludwig et al., 2019; Murray et al., 2019; Zhang et al., 2021a). In this study, 375 considering that NDWI was sensitive to open surface water and that Zhang et al. (2022b) found a positive relationship between tidal height and NDWI using field survey data, the maximum NDWI compositing was applied to the time-series clear-sky Landsat imagery to capture the optical highest water-level composites illustrated in Figure 4b. As for the lowest water-level features, considering that the tidal/flooded flat or marsh usually reflected higher NDVI and EVI values than water bodies and that Zhang et al. (2022b) also used the 380 field data to demonstrate that there was a negative relationship between tidal-level height and NDVI, the maximum NDVI composite was applied to capture the optical lowest water-level information illustrated in Figure 4a. Considering that optical observations were usually contaminated by clouds, especially during the rainy seasons, and that the SAR back coefficients had a great advantage in the presence of cloud coverage and

were found to be sensitive to the soil moisture, vegetation structure, and inundation information, the time-series Sentinel-1 SAR imagery could be used as a complementary dataset for capturing the highest and lowest water-level composites (DeVries et al., 2020; Li et al., 2020; Mahdianpari et al., 2018). Specifically, as the SAR active transmitting signals were heavily absorbed when they reached the water body, the corresponding SAR back coefficients in the water body had lower values compared to other land-cover types. To capture the high water-level features from the time-series Sentinel-1 imagery, the percentile compositing method using the 5th percentile was applied, as illustrated in Figure 4d. Conversely, the 95th percentile of Sentinel-1 VV and VH were generated to capture the lowest water-level information (Figure 4c). It should be noted that the minimum and maximum percentiles were not used because the time-series Sentinel-1 imagery still contained the residual errors caused by the quantitative processing.

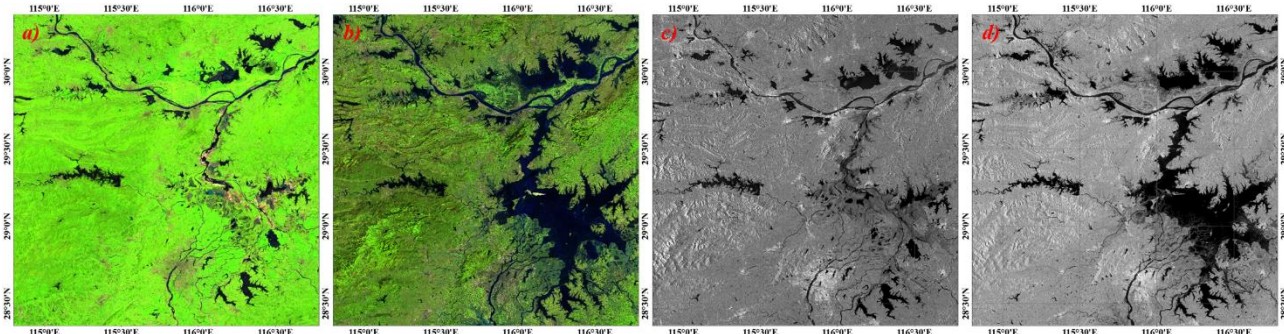

**Figure 4**. The lowest and highest water-level features derived from (a-b) time-series Landsat optical reflectance data and (c-d) the Sentinel-1 SAR imagery using the time-series compositing method in Poyang Lake, China.

Many studies also demonstrated that a multi-temporal phenology was also essential for classifying the vegetated wetlands and excluding these non-wetland land-cover types (Li et al., 2020; Ludwig et al., 2019). There were usually two options for capturing phenological features from time-series Landsat imagery. These included seasonal-based compositing (Zhang et al., 2021a; Zhang et al., 2022a) and percentile-based compositing (Hansen et al., 2014; Zhang and Roy, 2017; Zhang et al., 2021b). The former used the phenological calendar for selecting time-matched imagery. It then adopted the compositing rule to capture the seasonal features, while the latter directly used the statistical distributions to select various percentiles. Azzari and Lobell (2017) quantitatively analyzed the performance of two compositing methods and found that both of them had similar mapping accuracy for land-cover mapping. Meanwhile, the seasonal-based compositing method needed the prior phenological calendar, while the percentile compositing method did not require any prior knowledge or explicit assumptions regarding the timing of the season; therefore, the percentile compositing method was more suitable to generate phenological features. This study composited time-series Landsat reflectance bands and four spectral indexes into five percentiles (15th, 30th, 50th, 70th and 85th) because we wanted to capture as much of the phenological changes in wetlands as possible when comparing to the four seasonal composites (Zhang and Roy, 2017). It should be noted that the minimum and maximum percentiles were excluded because they were usually affected by residual clouds, shadows, and saturated observations.

Lastly, the topographical variables were also important factors for determining the spatial distribution of wetlands (Ludwig et al., 2019; Tootchi et al., 2019). For example, the widely used topographical wetness index (TWI) uses the local slope to reveal soil wetness, which improves wetland classification performance and reduces commission errors within upland areas (Ludwig et al., 2019). Therefore, the elevation, aspect, and slope, calculated from the ASTER GDEM dataset, were included in the multi-sourced features. In summary, a total of

77 multisourced training features (listed in Table 3), including 70 optical features from Landsat imagery, 4 SAR features from Sentinel-1 imagery and 3 topographical features from ASTER GDEM.

**Table 3**. The multisourced and multitemporal training features for wetland mapping.

| Data | Derived training features from multisource remote sensing imagery |
|---|---|
| Landsat | **Water-level features**: the lowest and highest composites with Blue, Green, Red, NIR, SWIR1, SWIR2, LSWI, NDWI, NDVI and EVI bands<br>**Phenological features:** 15th, 30th, 50th, 70th and 85th percentiles with Blue, Green, Red, NIR, SWIR1, SWIR2, LSWI, NDWI, NDVI and EVI bands |
| Sentinel-1 SAR | **Water-level features**: the lowest and highest composites using 5th and 95th percentiles for VV and VH bands. |
| ASTER GDEM | **Topographical features**: elevation, slope and aspect. |

## 4.2 The stratified classification strategy for wetland mapping

Since we have simultaneously extracted the maximum coastal and inland wetland extents when deriving training samples from prior wetland datasets, the stratified classification strategy was adopted to fully use the maximum extent constraint. If a pixel was classified as a coastal tidal wetland outside the maximum coastal tidal wetland extents, it would be identified as a misclassification. Furthermore, there were two ideas for the large-area land-cover mapping including global classification modeling (using one universal model for the whole areas) and local adaptive modeling (using various models for different local zones) (Zhang et al., 2020). For example, Zhang and Roy (2017) demonstrated that local adaptive modeling outperformed the global classification modeling strategy. Therefore, the global land surface was first divided into 961 $5° × 5°$ geographical tiles illustrated in Figure 5, which were inherited from the global 30 m land-cover mapping by (Zhang et al., 2021b). Then, we trained the local adaptive classification models using derived training samples in Section 3 and multisource and multitemporal features (the highest, lowest water-level and phenological composites and topographical variables) at each $5° × 5°$ geographical tile. It should be noted that we used the training samples from neighboring $3 × 3$ geographical tiles to train the classification model and classify the central tile for guaranteeing the spatially continuous transition over adjacent regional wetland maps. Namely, we trained 961 local adaptive classification models and then produced 961 $5° × 5°$ wetland maps. Finally, we spatially mosaiced these 961 regional wetland maps into the global 30 m wetland map in 2020.

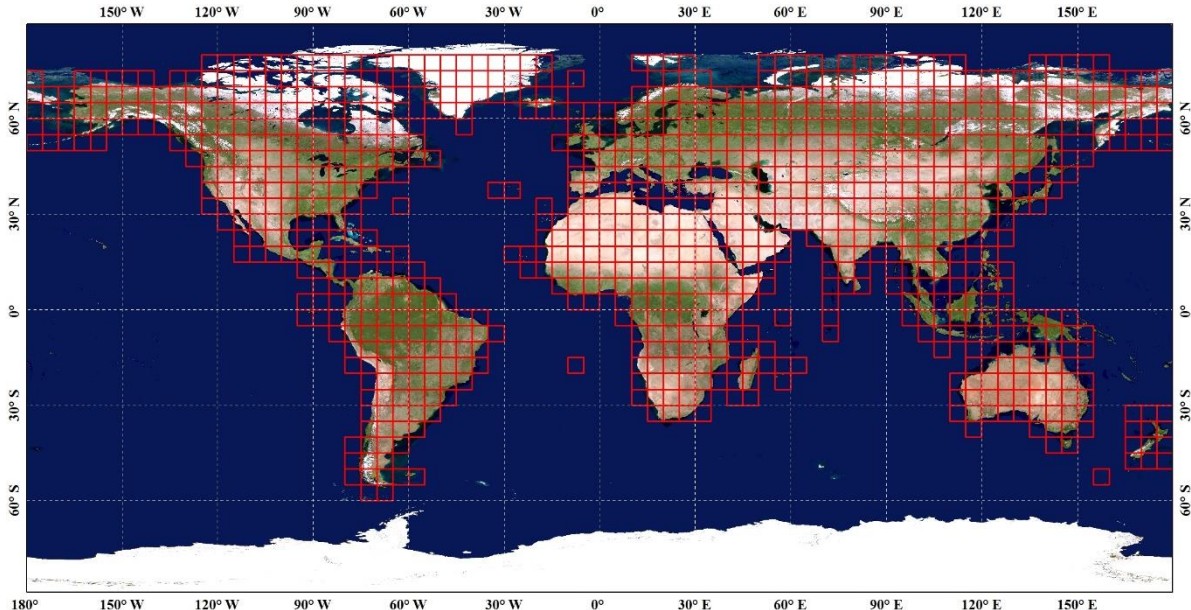

**Figure 5**. The spatial distribution of 961 5° × 5° geographical tiles used for local adaptive modeling, which was inherited from the global 30 m land-cover mapping by (Zhang et al., 2021b). The background imagery came from the National Aeronautics and Space Administration (https://visibleearth.nasa.gov, last access: 10 Nov 2022).

Afterward, the random forest (RF) classifier was demonstrated to have obvious advantages including: dealing with high-dimensional data, robustness for training noise and feature selection, as well as achieving higher classification when compared to other widely used machine learning classifiers (e.g., support vector machines, neural networks, decision trees, etc.) (Belgiu and Drăguţ, 2016; Gislason et al., 2006). Therefore, the RF classifier was selected for mapping inland and coastal tidal wetlands using multi-sourced features on the GEE platform. It should be noted that the RF classifier had two key parameters: the number of selected prediction variables (Mtry) and the number of decision trees (Ntree). Belgiu and Drăguţ (2016) and Zhang et al. (2022b) have demonstrated the quantitative relationship of Ntree against classification accuracy and found that the classification accuracy stabilized when Ntree was greater than 100. Meanwhile, Belgiu and Drăguţ (2016) suggested that the Mtry should take its default value of the square root of the number of all input features. Therefore, the Ntree and Mtry took 100 and the square root of the number of all input features, respectively.

The inland and coastal tidal wetland maps were produced by combining water-label and phenological features, the stratified classification strategy, local adaptive modeling, and the derived wetland and non-wetland training samples. As the inland and coastal tidal wetlands were independently produced, some pixels in the overlapping area of maximum inland and coastal tidal wetland extents were simultaneously labeled as inland wetlands and coastal tidal wetlands. However, as the final global wetland map was a hard classification, these pixels should be post-processed into one label. As the random forest classifier could provide the posterior probability for each pixel, we determined the labels of the confused pixels by comparing the posterior probabilities. In addition, as the tidal flats were demonstrated to overestimate some coastal ponds as the tidal flats, the global lake and reservoir dataset, developed by Khandelwal et al. (2022), was applied to optimize the tidal flat.

### 4.3 Accuracy assessment

To quantitatively analyze the performance of our GWL_FCS30 wetland map, a total of 25,709 validation samples (illustrated in Figure 6), including 10,558 non-wetland points and 15,151 wetland points, were collected. Firstly, as the wetland was sparse land-cover type compared to the non-wetlands (forest, cropland, grassland and bare land), the stratified random strategy was applied to randomly derive validation points at each strata as:

$$n_i = n \times \frac{W_i \times p_i(1-p_i)}{\sum W_i \times p_i(1-p_i)}; \qquad n = \frac{\left(\sum W_i\sqrt{p_i(1-p_i)}\right)^2}{V + \sum W_i p_i(1-p_i)/N} \qquad (5)$$

where $W_i$ and $p_i$ are the area proportion and expected accuracy of class $i$, $n_i$ and $n$ are the sample size of
class $i$ and total sample size, $V$ is the standard error of the estimated overall accuracy, and $N$ is the number of pixel units in the study region. Then, as the wetlands had significant correlation with the water levels (Zhang et al., 2022b), the time-series optical observations archived on the GEE cloud platform were used as the auxiliary dataset to interpret these water-level sensitive wetlands such as: tidal flat and flooded flat. It should be noted that the visual interpretation was implemented on the GEE cloud platform because it archives a large amount of
satellites imagery with various time spans and spatiotemporal resolution (Zhang et al., 2022a). Meanwhile, each validation point is independently interpreted by five experts for minimizing the effect of expert's subjective knowledge, and only these complete agreement points were retained otherwise they were discarded. Then, we employed four metrics typically used to evaluate accuracy, which include the kappa coefficient, overall accuracy, user's accuracy (measuring the commission error), and producer's accuracy (measuring the omission error)
(Gómez et al., 2016; Olofsson et al., 2014), were calculated using 25709 global wetland validation samples.

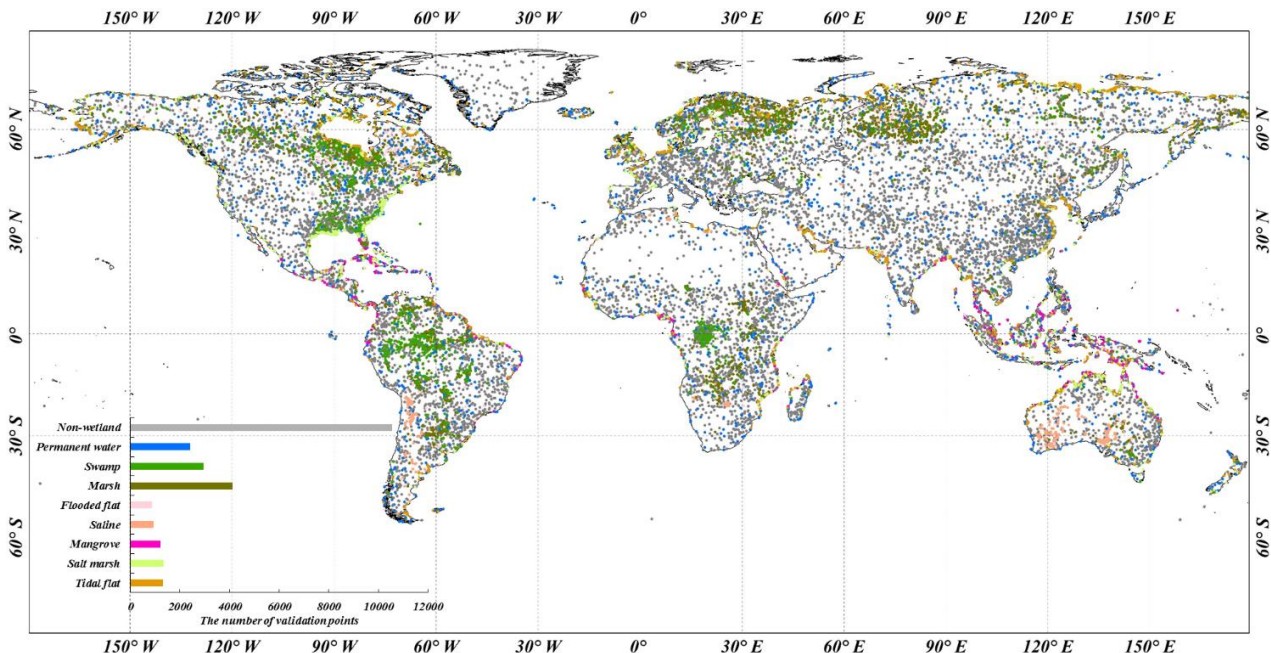

**Figure 6.** The spatial distribution of 25,709 global wetland validation samples using stratified sampling strategy.

## 5. Results

## 5.1 The reliability analysis of derived training samples

This study proposed combining multi-sourced pre-existing wetland products, refinement rules, and expert knowledge to automatically derive these massive inland and coastal tidal wetland training samples globally. To demonstrate the reliability of the derived training samples for wetland mapping, we randomly selected approximately 10,000 points from the sample pool and checked their confidence using visual interpretation. It should be noted that we cannot check all the training samples because the number of derived samples was

massive (exceeding 20 million training samples in Section 3). After a point-to-point inspection, these selected training samples achieved an overall accuracy of 91.53% in 2020. Meanwhile, we also used 10,000 selected wetland training samples and many non-wetland samples to analyze overall and producer's accuracies of coastal and inland wetlands versus number of erroneous training samples. Specifically, we gradually increased the "contaminated" samples by randomly altering the label of a certain percentage of training samples in steps of

0.01, and then used these "contaminated" samples to build the RF classification model. After repeating the process 100 times, the quantitative relationship between mapping accuracies and erroneous samples is illustrated in Fig. 7. Obviously, the overall accuracy and producer's accuracy of wetlands (merging seven sub-categories into one wetland) was insensitive to the erroneous training samples when the percentage of erroneous samples was controlled within 20%. Beyond this threshold, the accuracies slowly decreased along with the increase of

erroneous training samples. Similarly, previous studies by Zhang et al. (2021b) and Zhang et al. (2022a) quantitatively analyzed the relationship between overall accuracy and the erroneous training samples size. They found that the overall accuracy stabilized when the percentage of erroneous training samples was controlled within the threshold and then rapidly decreased after exceeding the threshold. Therefore, the derived training samples in Section 3 were accurate enough to support large-area fine wetland mapping.

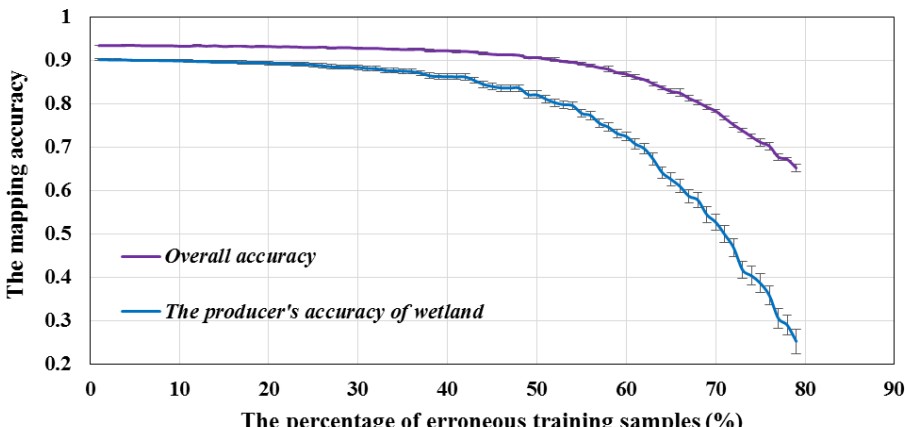


**Figure 7**. The relationship between mapping accuracies with the percentage of erroneous training samples with a step of 1%.

## 5.2 The importance of multi-sourced phenological features for wetland mapping

      The complicated temporal dynamics and spectral heterogeneity caused great uncertainties in wetland
mapping because their spectral characteristics quickly changed with the seasonal or daily water levels of the underlying surface (Ludwig et al., 2019). To quantitatively analyze the importance of these multi-sourced and multi-temporal features, we used the random forest classification model, which calculated the increased mean squared error by permuting the out-of-bag data of a variable while keeping remaining variables constant (Breiman, 2001; Zhang et al., 2020), in an effort to compute their importance. Figure 8 illustrated the importance

of all multi-sourced and phenological features, and it can be found that the phenological features which made the most significant contribution mainly did so because they used the multi-temporal percentiles to comprehensively capture vegetation phenology (EVI and NDVI) and water-level dynamics (NDWI and LSWI) for the various land-cover types. Then, the combination of optical and Sentinel-1 SAR water-level features ranked as the second-most important role in distinguishing the fine wetlands and non-wetlands. Based on the

lowest and highest water-level features in Fig. 4, the highest and lowest water-level features greatly contributed

to determining these water-sensitive wetlands (marsh, tidal flat, and flooded flat). For example, Zhang et al. (2022b) quantitatively analyzed the contribution of multi-sourced features to mapping accuracy. They found that importing water-level features significantly improved the ability to separate tidal flats from non-wetlands. Lastly, three topographical variables also contributed to wetland mapping because the spatial distribution of wetlands had a significant relationship with topography and was mainly distributed in low-lying areas (Zhu and Gong, 2014).

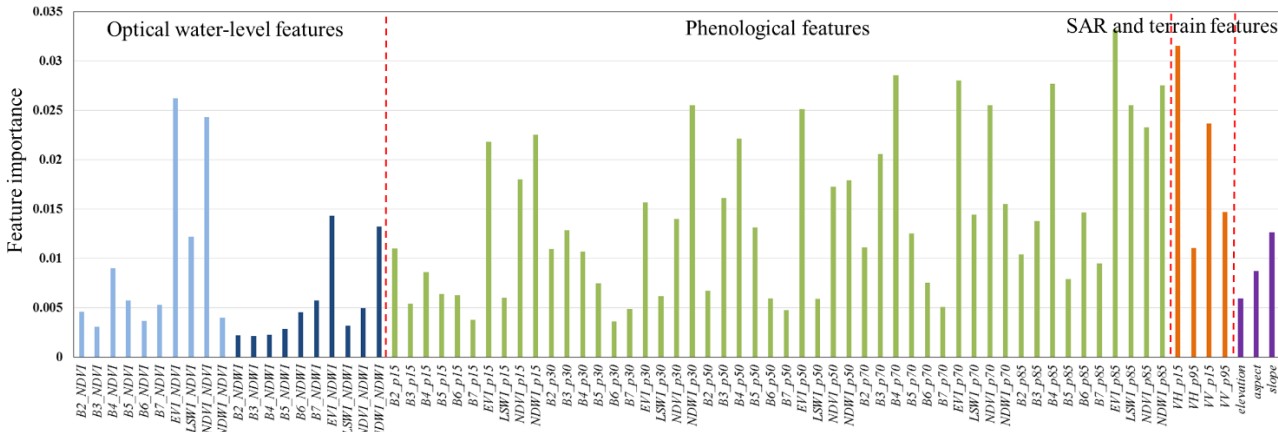

**Figure 8**. The importance of multi-sourced and multi-temporal features derived from the random forest classification model.

## 5.3 The spatial pattern of global wetlands in 2020

Figure 9 illustrates the spatial distributions of our GWL_FCS30 wetland map and their area statistics in latitudinal and longitudinal directions in 2020. Overall, the GWL_FCS30 map accurately captured the spatial patterns of wetlands. It mainly concentrated on the high latitude areas in North Hemisphere and the rainforest areas (Congo Basin and Amazon rainforest in South America). Quantitatively, according to the latitudinal statistics, approximately 72.96% of wetlands were distributed poleward of 40°N (a large number of wetlands are located in Canada and Russia), and 10.6% of wetlands were located in equatorial areas, between 10°S~10°N, within which the Congo and Amazon rainforest wetlands are located. As for the longitudinal direction, there were mainly four statistical peak intervals: 120°W~50°W (Canada wetlands and Amazon wetlands), 15°E~25°E (Congo wetlands), 40°E~55°E (the Caspian Sea), and 60°E~90°E (Russia wetlands). Afterward, to more intuitively understand the performance of our GWL_FCS30 wetland map, four local enlargements in Florida, the Congo Basin, Sundarbans, and Poyang Lake were also illustrated. All of them comprehensively captured the wetland patterns in these local areas. For example, there was significant consistency between our results and Hansen's regional wetland maps in the Congo Basin (Bwangoy et al., 2010); both results indicated that the wetlands occurred closer to major rivers and floodplains. Next, according to the lowest and highest water-level features derived from Sentinel-1 SAR and Landsat optical imagery in Figure 4, the inland wetlands, varied with the water-levels, were also comprehensively identified in the Poyang wetland map (Figure 9d). Figure 9c illustrates the spatial distributions of the world's largest mangrove forest in the Sundarbans (Figure 9c), and the cross-comparison in Figure 14 also demonstrates the great performance of the GWL_FCS30 dataset. Lastly, the Florida wetlands simultaneously contained six sub-categories (mangrove, tidal flat, salt marsh, marsh, permanent water and swamp). These were distributed along the coastlines and rivers and are accurately captured in Figure 9a.

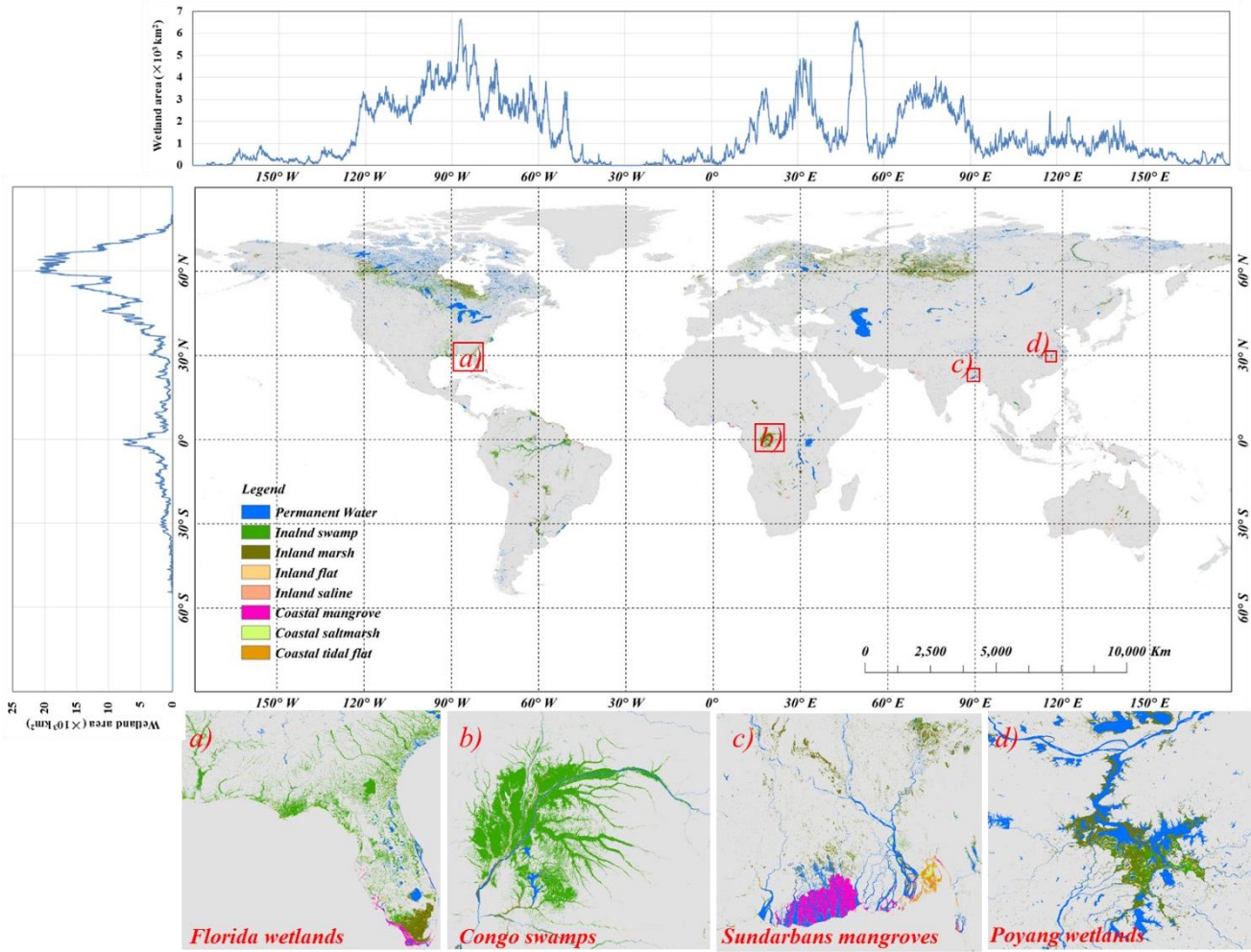

**Figure 9**. The overview of global 30-m fine wetland maps and their area statistics in latitudinal and longitudinal directions in 2020. Four local enlargements in (a) Florida, (b) Congo Basin, (c) Sundarbans, and (d) Poyang Lake were also illustrated.

Figure 10 illustrates the spatial distribution of eight wetland sub-categories after aggregating to the 0.5° × 0.5° grid cell. Intuitively, permanent water body, swamp and marsh accounted for most inland wetlands, while the flooded tidal and inland saline wetlands had obviously lower proportions and the later was only distributed along the surroundings of several saline lakes. In terms of the spatial distribution, it can be found that: 1) the swamp wetlands mainly were concentrated in the Congo and Amazon rainforests, Southern United States, and Northern Canada; 2) most marsh wetlands were located in high latitude areas in the Northern Hemisphere including Northern Canada, Russia, and Sweden; 3) there were significant coexistent relationships between flooded flat, permanent water, swamp, and marsh wetlands. Then, as for three coastal tidal wetlands, the mangrove forests were only found in coastal areas below 30°N and were mainly concentrated in regions between 30°N ~ 30°S, including Southeast Asia, West Africa, and the east coast of South America. The salt marshes and tidal flats shared similar spatial distributions. They were widely distributed globally and can be observed along most coastlines. In addition, the tidal flat distributions were closely related to the slope of coastlines, tidal ranges, and sediment inflows. For example, the tidal flats in Asia and Europe usually were located in the tide-dominated estuaries and deltas. Similarly, Murray et al. (2019) also demonstrated that there were often more tidal flats where the river flowed into the sea.

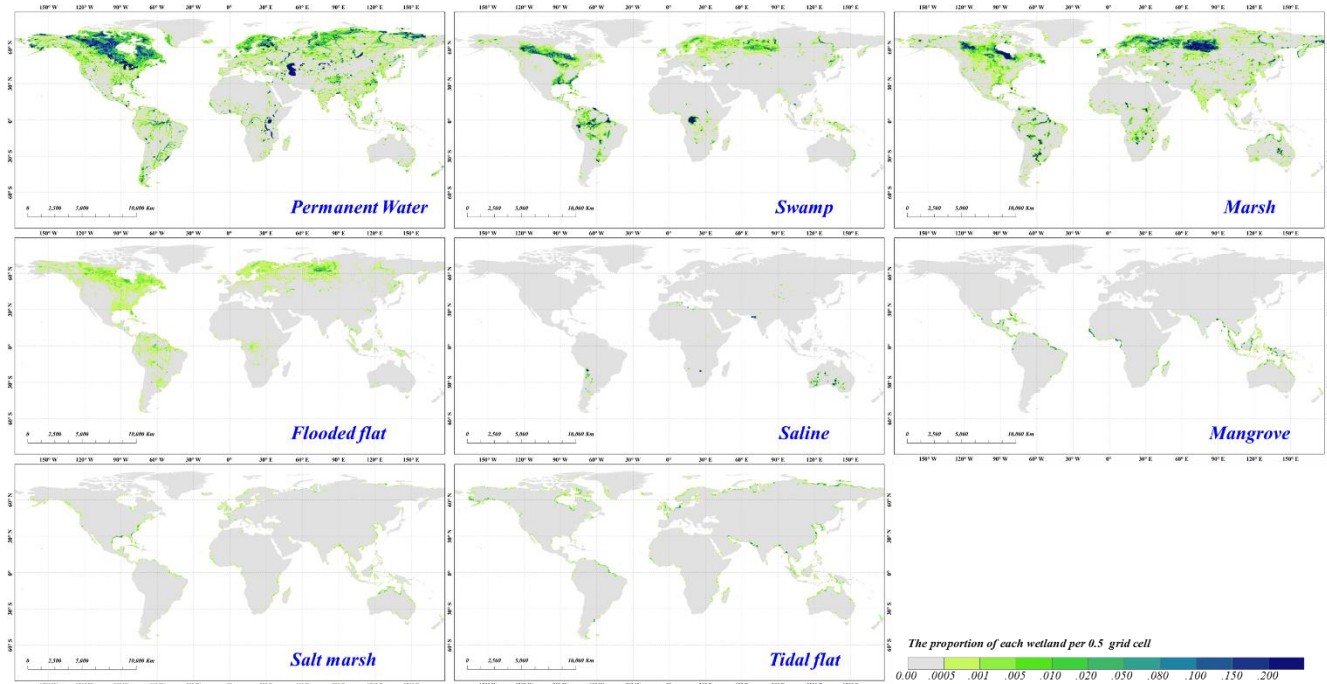

**Figure 10.** The spatial distributions of the eight wetland sub-categories after aggregating them to a resolution of $0.5° \times 0.5°$.

To quantitatively summarize the distribution of the eight sub-category wetlands, the total area and area percentages of eight fine wetland sub-categories over each continent were calculated in Figure 11 and Table 4. The total wetland area was 6.38 million $km^2$, including 6.03 million $km^2$ of inland wetlands and 0.35 million $km^2$ of coastal tidal wetlands, and the distribution of wetlands varied across different continents. Intuitively, approximately 60% of coastal tidal wetlands (tidal flat, salt marsh, and mangrove) and 70% of permanent water, flooded flat and marsh wetlands were distributed in the Northern Hemisphere, especially in the Asian and North American continents. Comparatively, more than 85% of saline wetlands were located in the Southern Hemisphere, especially the Oceania continent. Then, in terms of specific wetland sub-categories, most permanent water concentrated on the Northern Hemisphere especially in North America (nearly 50% of the world's permanent water bodies). The swamp was mainly distributed on the North American, African, and South American continents, which contained many rainforest wetlands, with corresponding swamp areas of 0.39, 0.18, and 0.32 million $km^2$, respectively. Swamp areas in the Oceania continent were the smallest, covering only 6572 $km^2$, mainly because the forest cover in Oceania was smaller than in other continents. The marsh and flooded flats shared similar areal proportions in all six continents and were mainly concentrated in the North Hemisphere (exceeding 70%), where many lakes and rivers were distributed. Next, as the mangrove forests only covered regions south of 30°N and were mostly concentrated in tropical regions near the equator, such as Southeast Asia, East Africa, and Central America, this sub-category was absent in the Europe continent and sparse in the Oceania.

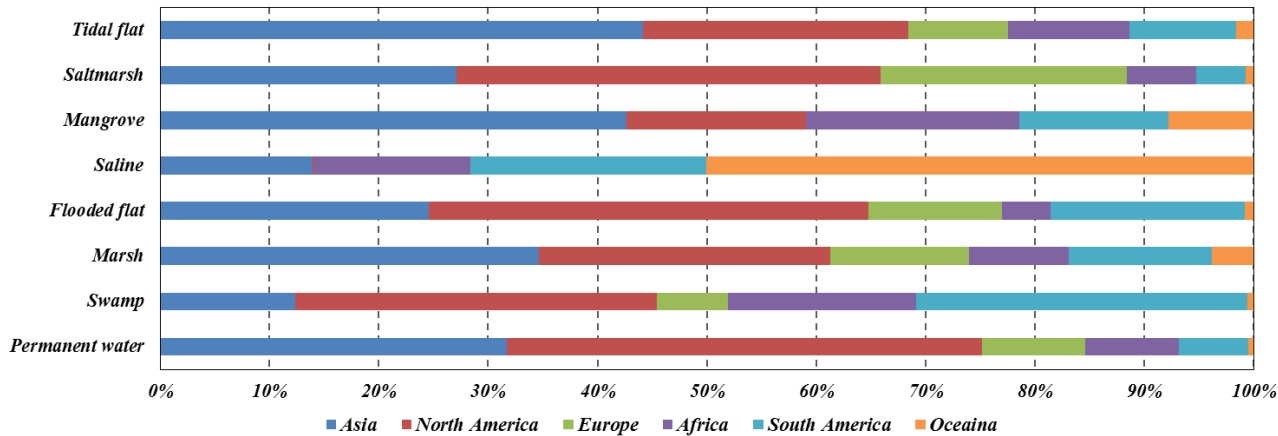

**Figure 11**. The area proportions of eight wetland sub-categories over each continent.

**Table 4**. The total wetland area (unit: $10^4 \times km^2$) of eight wetland sub-categories at six continents and globe.

|  | Permanent water | Swamp | Marsh | Flooded flat | Saline | Mangrove | Saltmarsh | Tidal flat |
|---|---|---|---|---|---|---|---|---|
| Asia | 90.529 | 13.227 | 58.229 | 7.244 | 1.215 | 6.636 | 1.852 | 5.347 |
| North America | 123.754 | 39.314 | 45.350 | 11.867 | 0.008 | 2.590 | 2.619 | 2.697 |
| Europe | 27.111 | 7.010 | 22.513 | 3.601 | 0.005 | 0.000 | 0.717 | 1.408 |
| Africa | 24.214 | 18.393 | 14.926 | 1.318 | 1.248 | 3.105 | 0.688 | 0.731 |
| South America | 18.310 | 32.337 | 21.640 | 5.242 | 1.888 | 2.175 | 0.520 | 1.238 |
| Oceania | 1.330 | 0.657 | 6.151 | 0.233 | 4.355 | 1.219 | 1.094 | 0.875 |
| Total | 285.247 | 110.938 | 168.810 | 29.504 | 8.719 | 15.725 | 7.491 | 12.296 |

## 5.4 Accuracy assessment of global 30 m fine wetland map

Using 25,709 global validation samples, the confusion matrix of the novel GMW_FCS30 wetland map was calculated in Table 5. Overall, our wetland map achieved an overall accuracy of 86.44% and a kappa coefficient of 0.82 across the fine wetland classification system. In terms of the producer's and user's accuracies, the non-wetlands achieved the highest producer's accuracy of 94.24%, mainly because we combined multi-sourced pre-existing wetland datasets to determine the maximum wetland boundary and further used multi-sourced and time-series imagery to distinguish between wetlands and non-wetlands. The permanent water achieved the highest user's accuracy of 95.99% because the permanent water had unique and stable spectra characteristics and the training samples were directly from the JRC_GSW database (Pekel et al., 2016). Then, as for the coastal tidal wetlands, mangrove forest and tidal flat achieved higher accuracies than salt marsh, with producer's accuracies of 91.43% and 88.12% and user's accuracies of 95.69% and 94.81%, respectively. The salt marsh had a lower producer accuracy of 74.09% because its reflectance spectra were affected by both water levels and vegetation cover with considerable spatiotemporal heterogeneity and the sparser prior saltmarsh products were adopted. Next, as for inland sub-categories, the swamp and marsh obviously performed better than the flooded flat, with producer's accuracies of 72.03% and 78.09%, respectively. It can be seen that the confusion between swamp and marsh was the main source of the misclassification error of swamp and that the marsh was simultaneously confused with non-wetland, swamp, and flooded flat because the spectra of marsh changed along with the water levels. For example, the marsh in Poyang Lake, shown in Figure 4b, was flooded at its highest water levels. Then, the flooded flat achieved a low producer accuracy of 65.83% because it usually coexisted with the marsh and shared similar spectral characteristics, so approximately 10.89% of flooded flat points were labeled as the

marsh in our wetland map. The saline wetland was mainly concentrated along the edge of salt lakes and demonstrated great performance in our mapping, with producer's and user's accuracies of 91.96% and 91.66%, respectively.

**Table 5**. The confusion matrix of the global 30 m fine wetland map using 25,709 validation points.

|  | NWT | PW | SWP | MSH | FFT | SAL | MGV | SMH | TFT | Total | P.A. |
|---|---|---|---|---|---|---|---|---|---|---|---|
| **NWT** | 9950 | 17 | 254 | 224 | 39 | 3 | 12 | 33 | 26 | 10588 | 94.24 |
| **PW** | 69 | 2251 | 4 | 15 | 63 | 0 | 0 | 8 | 9 | 2419 | 93.06 |
| **SWP** | 272 | 5 | 2127 | 452 | 74 | 11 | 3 | 9 | 0 | 2953 | 72.03 |
| **MSH** | 546 | 18 | 135 | 3218 | 149 | 18 | 2 | 34 | 1 | 4121 | 78.09 |
| **FFT** | 145 | 21 | 26 | 95 | 574 | 3 | 1 | 5 | 2 | 872 | 65.83 |
| **SAL** | 26 | 1 | 0 | 43 | 5 | 846 | 0 | 0 | 0 | 921 | 91.86 |
| **MGV** | 65 | 4 | 11 | 2 | 2 | 1 | 1109 | 15 | 3 | 1213 | 91.43 |
| **SMH** | 157 | 15 | 6 | 85 | 9 | 30 | 26 | 998 | 22 | 1347 | 74.09 |
| **TFT** | 78 | 13 | 0 | 11 | 7 | 11 | 6 | 29 | 1150 | 1305 | 88.12 |
| **Total** | 11308 | 2345 | 2563 | 4145 | 922 | 923 | 1159 | 1131 | 1213 | 25709 | |
| **U.A.** | 87.99 | 95.99 | 82.99 | 79.56 | 62.26 | 91.66 | 95.69 | 88.24 | 94.81 | | |
| **O.A.** | | | | | 86.44 | | | | | | |
| **Kappa** | | | | | 0.822 | | | | | | |

**Note**: **NWT**: non-wetlands, **PW:** permanent water, **SWP**: swamp, **MSH**: marsh, **FFT**: flooded flat, **SAL**: saline, **SMH**: salt marsh, **MGV**: mangrove forest, **TFT**: tidal flat, **O.A.**: overall accuracy, **P.A.**: producer's accuracy, **U.A.**: user's accuracy.

## 6. Discussion

### 6.1 Cross-comparisons with other global wetland maps

To comprehensively understand the performance of the GWL_FCS30 wetland maps, four existing global wetland datasets (GLC_FCS30, GlobeLand30, CCI_LC, and GLWD), listed in Table 1, were selected. Figure 12 quantitatively illustrates the total wetland area of five products over each continent. Specifically, the total wetland area of different wetland products varied. The GLWD obviously overestimated the wetland area on each continent mainly because it was derived from the compilation model instead of actual remote sensing observations (Lehner and Döll, 2004). Namely, the GLWD classified a large amount of non-wetlands as potential wetlands. The remaining four wetland products, derived from the Landsat and PROBE-V remote sensing imagery, shared a total wetland area of 4.128~7.364 million km$^2$, and our GWL_FCS30 wetland dataset had the total area of 6.387 million km$^2$ among these datasets. The CCI LC wetland layer contained the smallest wetland area of 4.128 million km$^2$, and the estimated area in North America was profoundly lower than the other datasets, mainly because the CCI LC heavily underestimated the wetland distribution in Canada after a comparison with the Canadian Wetland Inventory (Amani et al., 2019). Next, the total wetland area in GlobeLand30 and GLC_FCS30 wetland layer was higher than the developed GWL_FCS30 wetland dataset because some water-level sensitive non-wetlands (such as: irrigated cropland) were also captured in these two datasets.

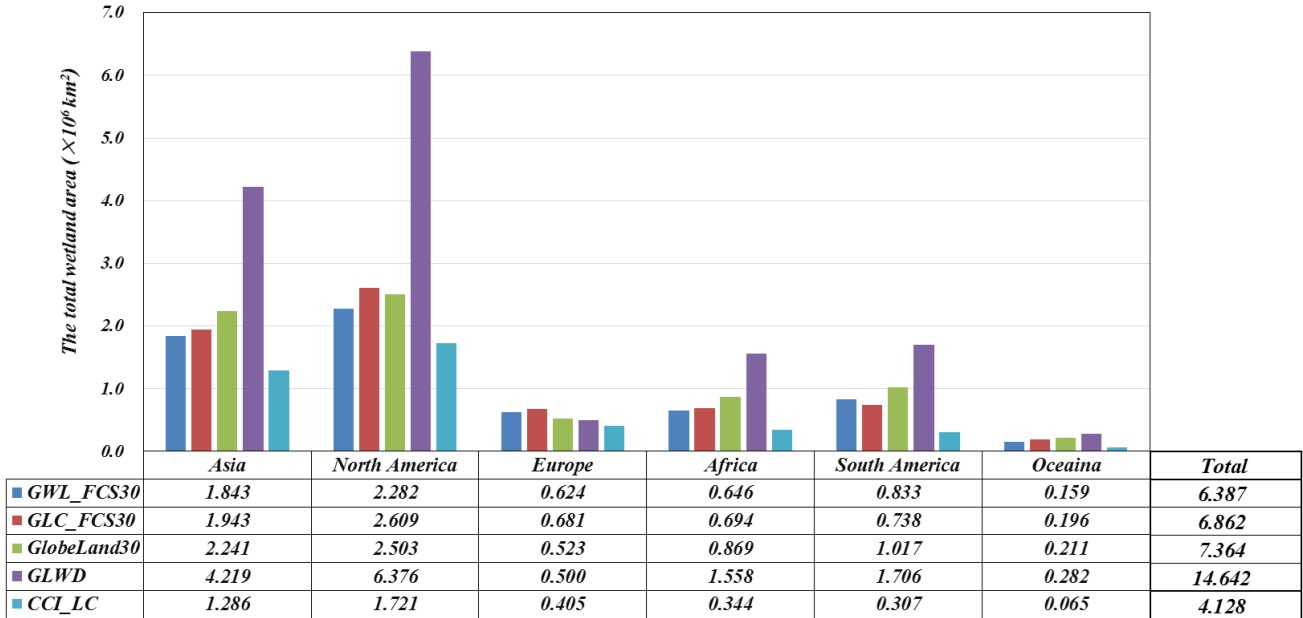

| | Asia | North America | Europe | Africa | South America | Oceaina | Total |
|---|---|---|---|---|---|---|---|
| ■ GWL_FCS30 | 1.843 | 2.282 | 0.624 | 0.646 | 0.833 | 0.159 | 6.387 |
| ■ GLC_FCS30 | 1.943 | 2.609 | 0.681 | 0.694 | 0.738 | 0.196 | 6.862 |
| ■ GlobeLand30 | 2.241 | 2.503 | 0.523 | 0.869 | 1.017 | 0.211 | 7.364 |
| ■ GLWD | 4.219 | 6.376 | 0.500 | 1.558 | 1.706 | 0.282 | 14.642 |
| ■ CCI_LC | 1.286 | 1.721 | 0.405 | 0.344 | 0.307 | 0.065 | 4.128 |

**Figure 12**. The total wetland area (unit: million $km^2$) of five global wetland products on six continents.

Figure 13 illustrates the performances of five wetland products for two typical wetland regions (Poyang Lake in China and Pantanal wetland in Brazil). The reasons for choosing these two regions were that the wetlands in Poyang Lake quickly changed with water levels, and the Pantanal wetland was the largest wetland in the world. Intuitively, the GWL_FCS30 wetland maps had the greatest performance in capturing the spatial 645 patterns of various wetland sub-categories. Comparatively, the GLC_FCS30 wetland layer suffered serious underestimation and misclassification problems in these two regions, which obviously misclassified many water-sensitive wetlands (swamp and marsh) as water bodies in Poyang Lake and also missed a large number of marsh and swamp wetlands in the Pantanal wetland. Zhang et al. (2021b) also stated that the wetland in the GLC_FCS30 suffered from low accuracy because of a lack of enough wetland samples and multi-sourced 650 wetland sensitive features. Then, the GlobeLand30 wetland layer performed better in the Pantanal wetland than in Poyang Lake, which also obviously misclassified many marsh wetlands as water bodies in the Poyang Lake mainly because the low water-level features were not captured during the development of the GlobeLand30 (Chen et al., 2015). In addition, the wetland layer of GlobeLand30 in Pantanal still suffered from the over-estimation problem, and some non-wetlands in Pantanal Wetland Park were mislabeled as wetland, so the 655 wetland layer in the GlobeLand30 only achieved a user's accuracy of 74.87% (Chen et al., 2015). The CCI LC was highly consistent with the GWL_FCS30 wetland maps in spatial distribution when comparing with GLC_FCS30 and Globeland30, however, details show that the wetlands in the CCI LC were still underestimated in the Poyang Lake wetland and overestimated in the Pantanal wetland based on the highest and lowest water-level composites. Lastly, the GLWD dataset significantly overestimated the wetlands in two regions, namely, 660 the mapped marsh area was obviously greater than its actual area and it also misclassified these water-sensitive wetlands as water bodies near Poyang Lake.

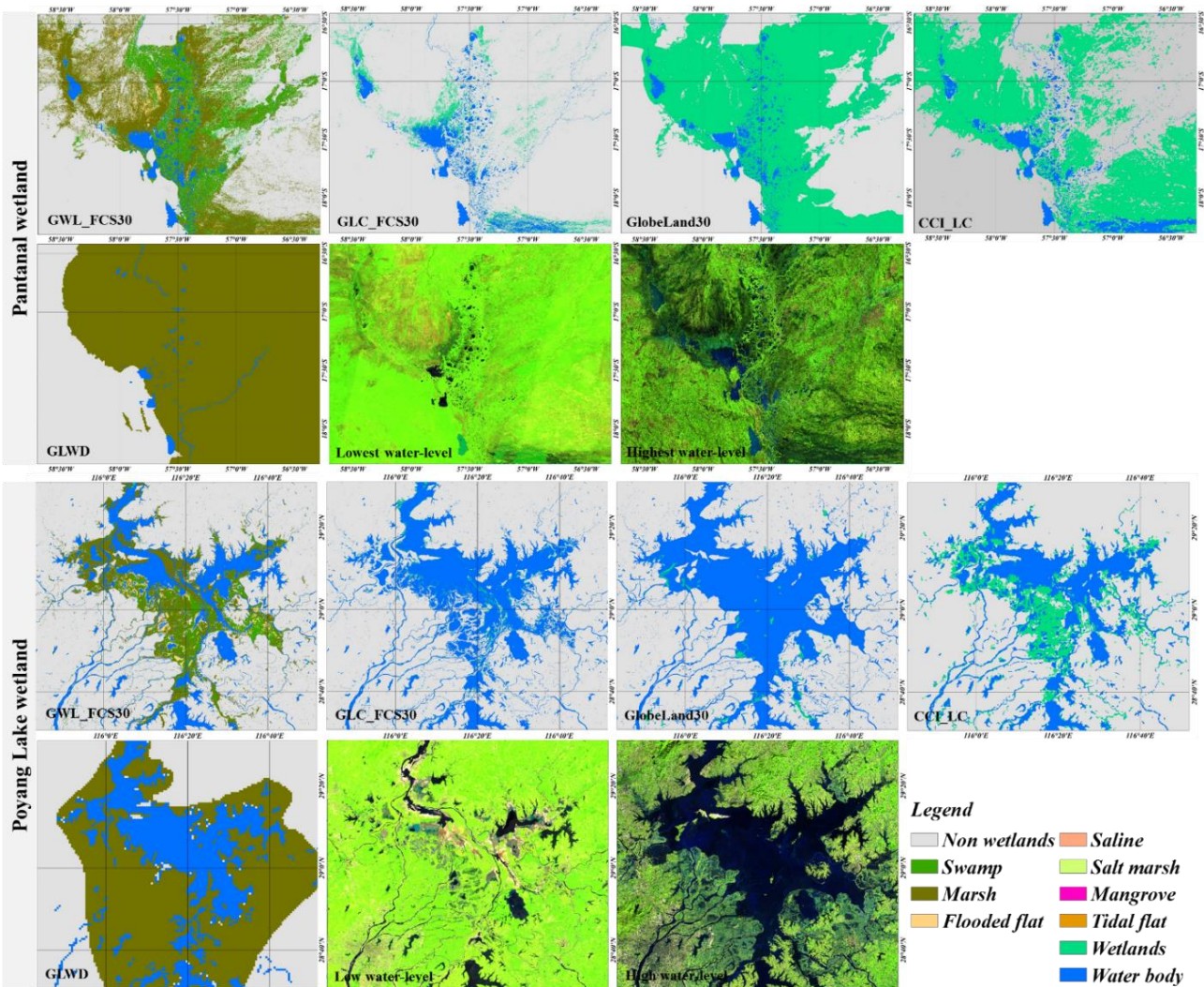

**Figure 13**. The cross-comparisons between our GWL_FCS30 wetland maps with four existing wetland products: GLC_FCS30 generated by Zhang et al. (2021b), GlobeLand30 generated by Chen et al. (2015), CCI LC generated by Defourny et al. (2018) and GLWD generated by Lehner and Döll (2004) at Pantanal and Poyang Lake wetland. The false-color composited Landsat imagery (SWIR1, NIR, and Red bands) at the highest and lowest water levels were also illustrated.

Figure 14 illustrates the comparisons between the GWL_FCS30 map with three widely used global mangrove forest products (Atlas mangrove, GMW_V3 (Global Mangrove Watch Version3), and USGS Mangrove) listed in Table 1 in two typical mangrove regions (coastal Indonesia and Sundarbans). Overall, there was great consistency over four mangrove datasets because the mangrove forest reflected obvious and strong vegetation reflectance characteristics and was easier to identify than other wetland sub-categories. Detailedly, the Atlas mangrove dataset suffers from the underestimation problem; namely, the mangrove area in the Atlas mangrove dataset was obviously lower than the other three products, especially in coastal Indonesia (local enlargements). The USGS mangrove product can comprehensively and accurately capture the spatial distribution of mangroves over two regions. Still, it missed small and isolated fragments of mangrove forests in two regions (green rectangle) based on high-resolution imagery. The GMW_V3 dataset was validated to achieve an overall accuracy of 95.25%, with user and producer accuracies of mangrove forests of 97.5% and 94.0%, respectively (Bunting et al., 2018; Thomas et al., 2017), which shows the greatest agreement with our

GWL_FCS30 maps in this two regions and enlargements. Using the high resolution imagery, it can be found that GWL_FCS30 and GWM_V3 accurately identified the spatial patterns of mangrove forest in both regions.

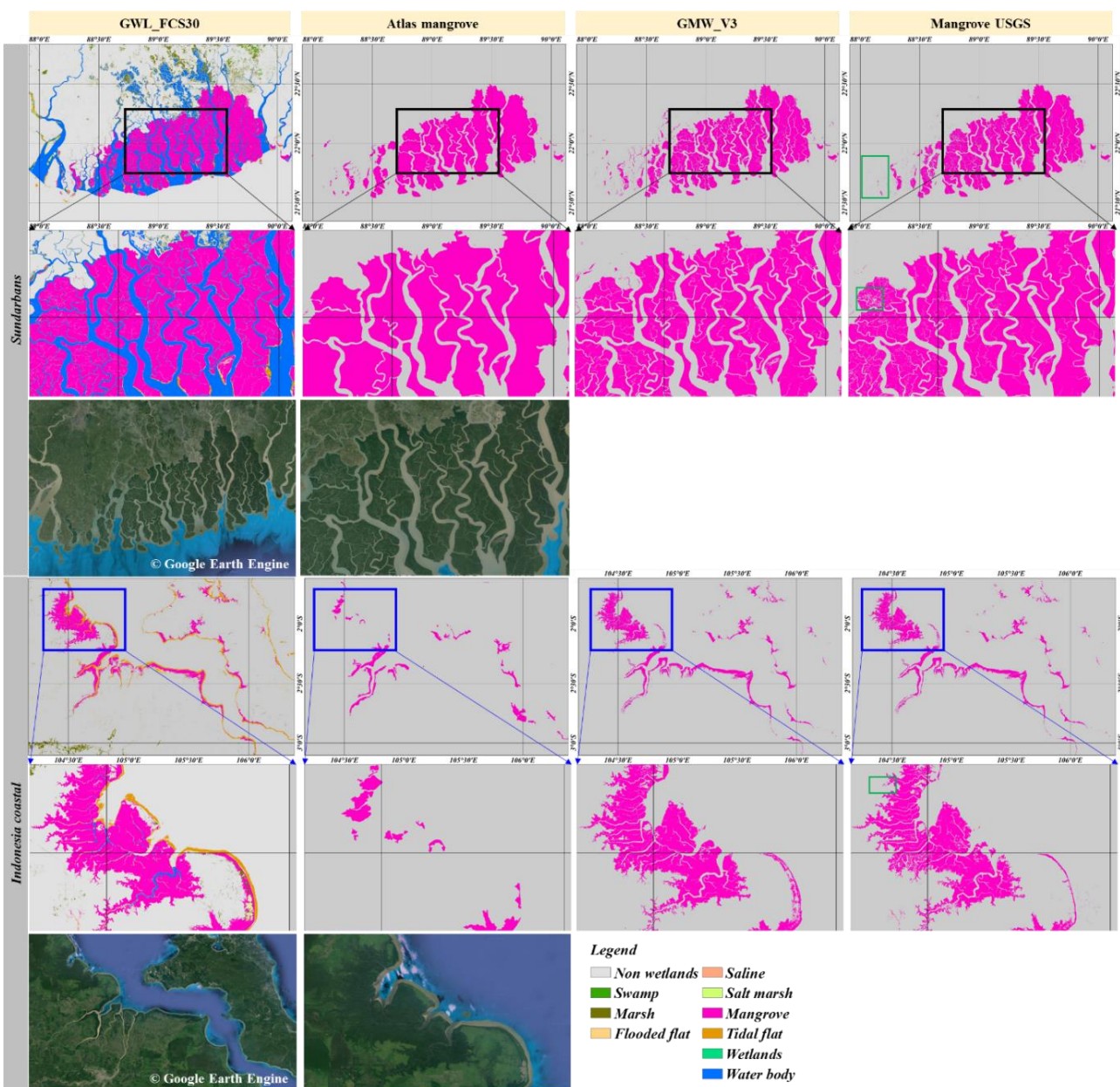

**Figure 14**. The cross-comparisons between our GWL_FCS30 wetland maps with three mangrove products (Atlas mangrove developed by Spalding (2010), GMW_V3 developed by Bunting et al. (2022) and Mangrove
USGS developed by Giri et al. (2011)) in Sundarbans and coastal Indonesia. The high-resolution imagery came from the Google Earth Engine platform (https://earthengine.google.com; last access: 16 May 2022).

  Figure 15 illustrated the comparisons between GWLF_CS30 tidal flat layer with the Murray's tidal flat V1.1 in 2016 (Murray et al., 2019) and the updated Murray's tidal flat V1.2 in 2019 (Murray et al., 2022) in two local regions, and the corresponding highest and lowest tidal-level composites are also listed. Overall, three
products can comprehensively capture the spatial patterns of tidal flats in these two regions, and the GWL_FCS30-2020 and Murray's tidal flat V1.2 performed higher spatial consistency while the Murray's tidal flat V1.1 suffered the obvious omission error in three typical areas (red rectangles). Detailedly, we can find that the Murray's tidal flat products misclassified some coastal ponds and lakes into the tidal flats especially in the

first region while the GWL_FCS30-2020 achieved the best performance and accurately excluded these coastal ponds and lakes. In addition, the GWL_FCS30 also distinguished the salt marshes and tidal flats especially in the Yellow River estuary while the Murray's tidal flat V1.2 database misclassified a lot of salt marshes into the tidal flats.

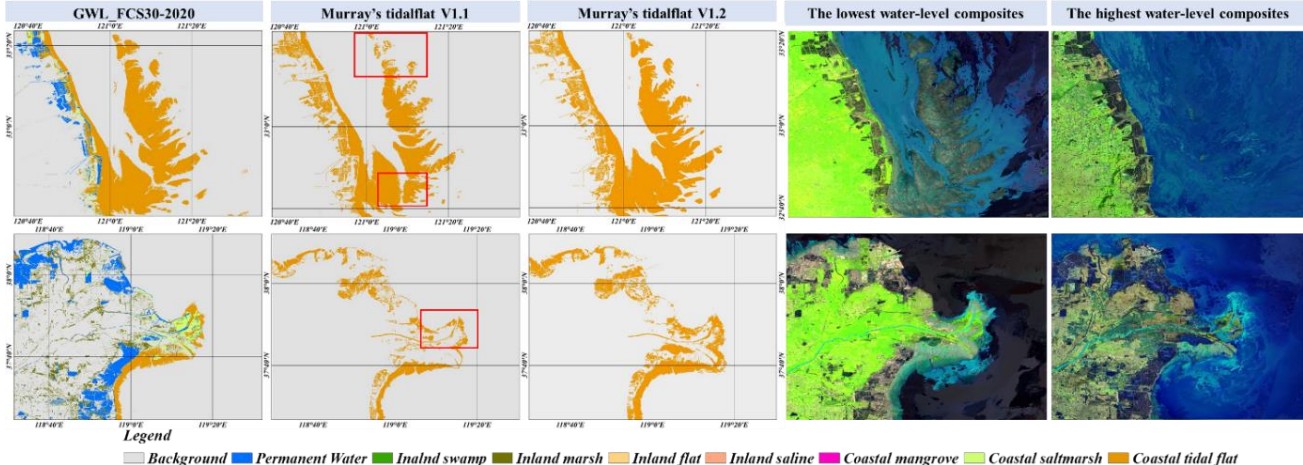

**Figure 15**. The comparisons between the tidal flat of GWL_FCS30 in 2020, Murray's tidal flat V1.1 in 2016 (Murray et al., 2019) , and Murray's tidal flat V1.2 in 2019 (Murray et al., 2022) for two local regions. In each case, the highest and lowest tidal-level composites, composited by SWIR1, NIR, and red bands, are illustrated.

## 6.2 Comparisons with the national wetland products

Using 1835 validation points (from the global validation points in Section 4.3) over the continuous United States, we quantitatively assessed the accuracy metrics of NLCD (National Land Cover Database) with GWL_FCS30 after merging the wetland subcategories into 4 classes in Table 6. Overall, the GWL_FCS30 achieved a higher performance than that of the NLCD mainly because a lot of herbaceous wetlands were misclassified into the open water in the NLCD, so the user's accuracy of herbaceous wetland and producer's accuracy of open water in NLCD was lower than that of GWL_FCS30. Then, as the NWI (National Wetlands Inventory) had different wetland system with the NLCD and GWL_FCS30, we also analyzed the metrics of NWI with GWL_FCS30 after merging into 5 classes. It can be found that the NWI shared similar performances with GWL_FCS30 on the non-wetlands and marine wetlands, but the user's accuracies of forest wetland and herbaceous wetland of NWI were lower than that of GWL_FCS30 mainly because some non-wetlands and open water were overestimated as the wetland in NWI. Similarly, Gage et al. (2020) also demonstrated that the NWI was easier to overestimate the wetland areas.

**Table 6**. The accuracy metrics of NLCD, NWI and GWL_FCS30 using 1835 validation points over the continuous United States

| (a) NLCD vs GWL_FCS30 | | | | | | | | | | | |
|---|---|---|---|---|---|---|---|---|---|---|---|
| | | NWT | Open water | | | Woody wetland | | Emergent herbaceous wetland | | | O.A. | Kappa |
| NLCD | U.A. | 96.46 | 93.98 | | | 77.92 | | 61.97 | | | 83.58 | 0.756 |
| | P.A. | 88.80 | 53.65 | | | 85.96 | | 87.61 | | | | |
| | | NWT | PW | FFT | TFT | SWP | MGV | MSH | | SMH | O.A. | Kappa |
| GWL_FCS30 | U.A. | 90.55 | 94.81 | | | 69.87 | | 87.61 | | | 85.76 | 0.786 |
| | P.A. | 85.99 | 95.52 | | | 77.97 | | 88.36 | | | | |
| (b) NWI vs GWL_FCS30 | | | | | | | | | | | |

| NWI | | NWT | FPD EMD RVR LKE | FSSW | FEW | EMW | O.A. | Kappa |
|---|---|---|---|---|---|---|---|---|
| | U.A. | 94.45 | 94.74 | 67.58 | 60.25 | 85.71 | 83.49 | 0.762 |
| | P.A. | 84.93 | 63.32 | 86.62 | 82.76 | 91.53 | | |
| GWL_FCS30 | | NWT | PW | SWP | MSH TFT | MGV SMH TFT | O.A. | Kappa |
| | U.A. | 90.55 | 94.74 | 68.96 | 80.75 | 90.08 | 85.23 | 0.789 |
| | P.A. | 85.99 | 95.45 | 76.76 | 78.78 | 94.98 | | |

Note: NWT: non-wetlands, PW: permanent water, SWP: swamp, MSH: marsh, FFT: flooded flat, SMH: salt marsh, MGV: mangrove forest, TFT: tidal flat, FPD: Freshwater Pond, EMD: Estuarine and Marine Deepwater, RVR: Riverine, LKE: Lake, FSSW: Freshwater Forested/Shrub Wetland, FEW: Freshwater Emergent Wetland, EMW: Estuarine and Marine Wetland, O.A.: overall accuracy, P.A.: producer's accuracy, U.A.: user's accuracy.

Figure 16 illustrated the comparisons between our GWL_FCS30-2020, NLCD wetland layer and NWI in San Francisco and Florida. It should be noted that the ocean was excluded in the GWL_FCS30-2020 while NLCD and NWI still retained. Overall, three wetland products performed great spatial consistency and accurately captured the spatial patterns of wetlands over two regions. From the perspective of diversity of wetland sub-category, the GWL_FCS30 and NWI had obvious advantages over the NLCD which simply divided the wetlands into open water, woody wetlands and emergent herbaceous wetlands. Afterwards, the NWI had the largest wetland areas in the San Francisco because it included the irrigated cropland (red color) while the other two datasets excluded irrigated cropland. Then, the local enlargement showed that the GWL_FCS30 and NWI also had better performance than NLCD, because they comprehensively captured the coastal tidal wetlands, and our GWL_FCS30 further distinguished the tidal flats and salt marshes which also demonstrated that GWL_FCS30 performed better than NWI over the coastal tidal wetlands. In the Florida, the NWI and GWL_FCS30 accurately divided the inland and coastal tidal wetlands and the GWL_FCS30 further identified the coastal tidal wetlands into the mangrove forest. Meanwhile, the local enlargement also demonstrated the great consistency of three wetland products. However, it can be found that there was obvious difference between GWL_FCS30 and NWI over the wetland categories, in which GWL_FCS30 classified most inland wetlands into marshes while NWI classified them as emergent wetlands and forest/shrub wetlands, mainly because of the differences in the definition of the classification system (GWL_FCS30 defined those low shrubs that grown in the freshwater as marsh, in Table 1).

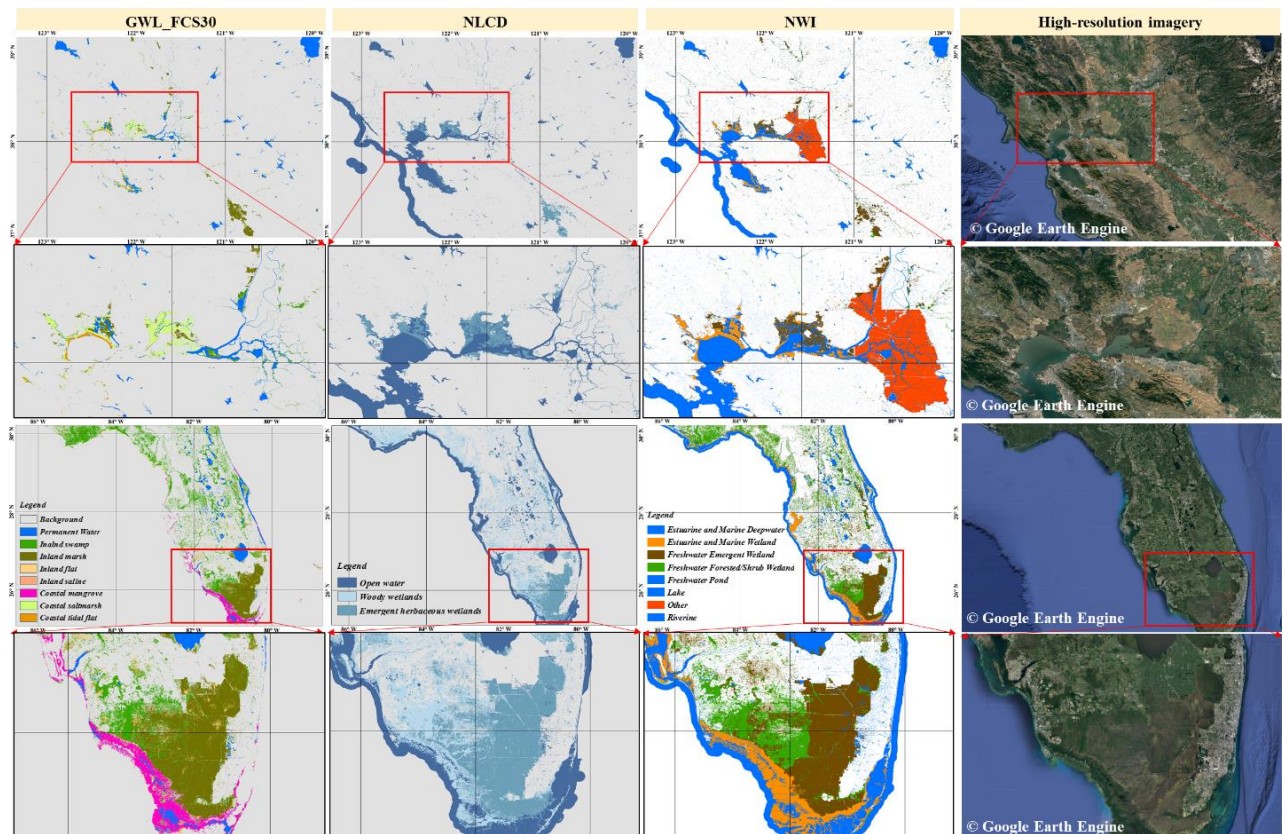

**Figure 16**. The comparisons between GWL_FCS30 in 2020, National Land Cover Database (NLCD) wetland Layer (Homer et al., 2020) and National Wetlands Inventory (NWI, https://www.fws.gov/program/national-wetlands-inventory, last access: Nov 12, 2022) in San Francisco and Florida. The high-resolution imagery came from the Google Earth Engine platform (https://earthengine.google.com; last access: 12 Nov 2022).

Table 7 illustrated the accuracy metrics of CLC (CORINE Land Cover) and GWL_FCS30 after merging the wetland categories over the European Union area using 1996 validation points from the global validation points in Section 4.3. Overall, the GWL_FCS30 performed better than the CLC and the former mainly had lower commission errors than that of the CLC for salt marsh and tidal flat. To intuitively understand the overestimation of tidal flat, Figure 17 illustrated the comparison between our GWL_FCS30-2020 and CLC wetland layer in 2018 over the Nordic, in which mainly distributed in tidal flats and open water, and these tidal flats gathered around the coastline. In term of specific wetland subcategory, it can be found that the CLC database had larger tidal flat area than that of the GWL_FCS30, however, the lowest tidal-level composite from time-series Landsat imagery indicated that the CLC overestimated the tidal flats in the region. For example, the local enlargement showed that a lot of permanent ocean pixels were wrongly labelled as the tidal flats in CLC and accurately identified as ocean in the GWL_FCS30. The comparison also demonstrated why the CLC had low user's accuracy of 62.90% for tidal flat and producer's accuracy of 57.76% for water bodies. Then, the local enlargement also indicated that the total area of salt marsh in CLC was lower than that of GWL_FCS30 (green rectangles), namely, some salt marshes were wrongly labelled as tidal flat and water body, so the accuracy metrics in Table 7 showed the user's accuracy of salt marsh in CLC was 35.86%.

**Table 7**. The accuracy metrics between CLC and GWL_FCS30 after merging the wetland categories

| CLC | NWT | WC | WB | CL | ET | SO | Peat bogs & Inland marshes | SMH | TFT | O.A. | Kappa |
|---|---|---|---|---|---|---|---|---|---|---|---|

| | | NWT | PW | SWP | MSH | FFT | SMH | TFT | O.A. | Kappa |
|---|---|---|---|---|---|---|---|---|---|---|
| | U.A. | 92.94 | 94.81 | 68.63 | | | 35.86 | 62.90 | 80.75 | 0.706 |
| | P.A. | 82.80 | 57.76 | 83.93 | | | 91.23 | 75.00 | | |
| | | NWT | PW | SWP | MSH | FFT | SMH | TFT | O.A. | Kappa |
| GWL_FCS30 | U.A. | 91.22 | 88.02 | 80.98 | | | 86.21 | 94.35 | 88.10 | 0.816 |
| | P.A. | 88.54 | 97.69 | 80.82 | | | 91.91 | 97.50 | | |

**Note**: NWT: non-wetlands, WC: water courses, WB: water bodies, CL: coastal lagoons, ET: estuaries, SO: sea and ocean, PW: permanent water, SWP: swamp, MSH: marsh, FFT: flooded flat, SAL: saline, SMH: salt marsh, MGV: mangrove forest, TFT: tidal flat, O.A.: overall accuracy, P.A.: producer's accuracy, U.A.: user's accuracy.

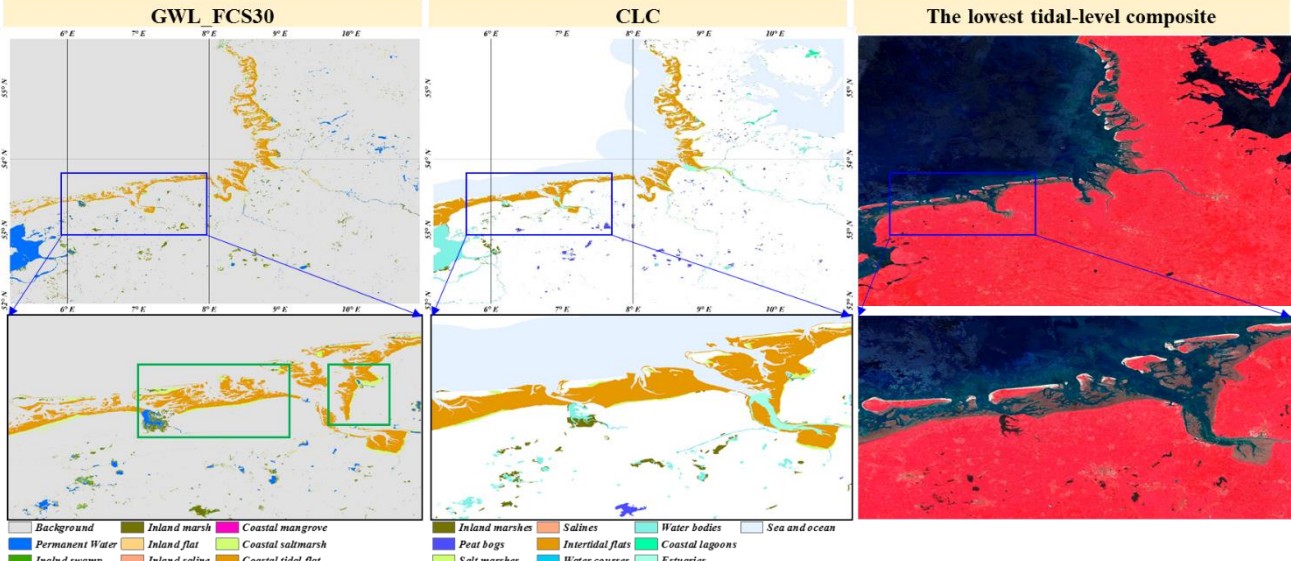

**Figure 17**. The comparisons between GWL_FCS30 and CORINE Land Cover (CLC) wetland layer in 2018 (https://land.copernicus.eu/pan-european/corine-land-cover/ clc2018?tab=metadata, last access: Nov 22, 2022). The lowest tidal-level Landsat composite, composited by NIR, red, and green bands, was illustrated.

## 6.3 The limitations and prospects of our global fine wetland map

It should be noted there were still many uncertainties and limitations to the proposed method and global wetland maps. First, the proposed method used continuous Landsat reflectance and Sentinel-1 SAR imagery to capture various water-level information. Still, it might fail when the available Landsat observations were sparse and lacked the aid of Sentinel-1 SAR data, especially before 2000. Thus, our future work would focus on combining a richer multi-sourced data source, including MODIS, Sentinel-2, SPOT, and PALSAR imagery, to develop a more robust wetland mapping method. For example, Chen et al. (2018) integrated Landsat and MODIS observations to successfully monitor the wetland dynamics from 2000 to 2014 using a spatiotemporal adaptive fusion model. Then, in this study, we combined the multisourced wetland products and their practical use for ecosystem management to define a fine wetland classification system containing eight sub-categories, however, there are still many wetland sub-categories, such as: submergent vegetation (nymphaea), groundwater-dependent wetlands (karst and cave systems) and seagrass beds (Richardson et al., 2022), cannot be captured because remote sensing observations usually had poor performance on penetrating water body and then capturing underwater characteristics, and there was currently no prior dataset for global underwater wetlands. Meanwhile, some coastal swamps (except for mangrove), which were usually overlooked at most coastal wetland mapping (Murray et al., 2022; Zhang et al., 2022), were also missed in the GWL_FCS30, mainly

because there are no global or large-area coastal swamp dataset can be imported and the coastal swamp is also sparser than the mangrove forest in the low and middle latitudes. So, our further work will pay more attention to combine multisourced auxiliary datasets, such as hydrological data, bathymetry depth and climate data, to map these special wetland sub-categories in a targeted manner.

We combined the pre-existing global wetland products to derive the training samples and maximum extents; however, the salt marsh and saline samples still used the visual interpretation method to ensure their reliability because of lacking sufficient pre-existing global products. Additionally, it was found that the producer's accuracy of salt marsh and saline in Table 4 was relatively poor compared with other sub-categories mainly because visual interpretation cannot provide massive and geographically distributed salt marsh and saline training samples. Namely, this study cannot comprehensively capture the regional adaptive reflectance characteristics of salt marsh and saline. Fortunately, many studies have built expert knowledge of these sub-categories over recent years. For example Mao et al. (2020) combined multi-scale segmentation, multiple normalized indices, and rule-based classification methods to develop a wetland map of China with an overall classification accuracy of 95.1%. Similarly, Wang et al. (2020) used the four widely used spectral indices to successfully identify three sub-categories within coastal tidal wetlands. Thence, our further work would attach more effort on the spectral characteristics of salt marsh and saline wetlands and build expert knowledge of them for automatically deriving their training samples.

In addition, we used the derived maximum extents as the boundary for identifying inland and coastal tidal wetlands, in other words, we assumed that the derived maximum extents contained all inland and coastal tidal wetlands with zero omission error. Actually, the inland maximum extents in Eq. (3) fulfilled the assumption of zero omission error, because the GLWD and TROP-SUBTROP products, produced by the compilation and model simulation method (Gumbricht, 2015; Lehner and Döll, 2004), can capture most wetland areas at the expense of a higher commission error. For example, the Figure 13 illustrated the cross-comparisons between our GWL_FCS30 wetland maps with four existing wetland products, and the GLWD obviously overestimated the inland wetlands. On the other hand, the union of five global wetland datasets in Eq. (3) also minimized the omission error of each dataset for inland wetland sub-categories. Next, as for the maximum mangrove forest extents (Eq. (1)), as the high producer's and user's accuracies were achieved by five prior mangrove products (explained in Section 2.2) and the time-series mangrove products were integrated that these missed mangroves may be complemented by other products or time-series products, the derived maximum extents also can be considered as zero omission error and covered almost all mangrove forests. Recently, Bunting et al. (2022) developed the newest mangrove products covering 1996-2020, it can be used as another important prior dataset in our further works for deriving the maximum mangrove extents. Lastly, the maximum tidal flat extents, derived from time-series Murray's products from 1985~2016 by using the union operation (Eq. (2)), can also contain almost all tidal flats because previous studies demonstrated that they suffered higher commission error than the omission error (Jia et al., 2021; Zhang et al., 2022b).The missed tidal flats would concentrate on these newly increased tidal flats during 2016-2020, fortunately, the new time-series global tidal flat products during 1999-2019 was developed (Murray et al., 2022) and can be used as an important supplement in our further work for deriving the maximum tidal flat extent with zero omission error.

## 7. Data availability

The GWL_FCS30 wetland dataset in 2020 was freely available at https://doi.org/10.5281/zenodo.7340516 (Liu et al. 2022). It was composed of 961 5°×5° geographical grid tiled files, and each tiled file was stored using

the geographical projection system with a spatial resolution of 30-meter in the GeoTIFF format. The fine
wetland subcategory information was labeled as 0, 180, 181, 182, 183, 184, 185 186 and 187, representing the
non-wetland, permanent water, swamp, marsh, flooded flat, saline, mangrove forest, salt marsh and tidal flat,
respectively. The validation samples are available upon request.

## 8. Conclusions

Over the past few decades, many global and regional wetland products have been developed; however, an
accurate global 30-m wetland dataset, with fine wetland categories and coverage of both inland and coastal
zones, is still lacking. In this study, the time-series Landsat reflectance and Sentinel-1 SAR imagery, together
with the stratified classification strategy and local adaptive random forest classification algorithm, were
successfully integrated to produce the first global 30-m wetland product with a fine classification system in
2020. The wetlands were classified into four inland wetlands (swamp, marsh, flooded flat, and saline) and three
coastal tidal wetlands (mangrove, salt marsh, and tidal flat). The produced wetland dataset, GWL_FCS30,
accurately captured the spatial patterns of seven wetland sub-categories with an overall accuracy of 86.44% and
a kappa coefficient of 0.822 for the fine wetland classification system with lower omission and commission
errors compared to other global products. The quantitative statistical analysis showed that the global wetland
area reached 6.38 million $km^2$, including 6.03 million $km^2$ of inland wetlands and 0.35 million $km^2$ of coastal
tidal wetlands. Approximately 72.96% of wetlands were distributed poleward of 40°N. Therefore, the proposed
method is suitable for large-area fine wetland mapping, and the GWL_FCS30 dataset can serve as an accurate
wetland map that could potentially provide vital support for wetland management.

## Acknowledgements

This research has been supported by the Innovative Research Program of the International Research Center of
Big Data for Sustainable Development Goals (Grant No. CBAS2022ORP03), the National Natural Science
Foundation of China (grant no. 41825002) and National Earth System Science Data Sharing Infrastructure (grant
no. 2005DKA32300).

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
