# Peer review of "GWL\_FCS30: global 30 m wetland map with fine classification system using multi-sourced and time-series remote sensing imagery in 2020"

_Earth System Science Data, 2022_

## Author Comment (AC1)

**Response to comments**

**Paper #:** essd-2022-180
**Title:** GWL_FCS30: global 30 m wetland map with fine classification system using multi-sourced and time-series remote sensing imagery in 2020
**Journal**: Earth System Science Data

**Reviewer #1**

In terms of the whole workload, there is no doubt that this paper carried out a lot of data processing and analysis. The submitted MS also has good writing. However, I still have some concerns about the method and accuracy of classification results. My major comments are as below.

Great thanks for the positive comments. The manuscript has been greatly improved based on your and two other reviewers' comments and suggestions.

1. Besides the mangrove, how did you consider other coastal swamp in the classification system?

Great thanks for the comment and this great question. I am sorry we do not consider the coastal swamp in the manuscript because: 1) Ramsar convention only defines the coastal wetlands into: unvegetated tidal flats, saltmarshes, coastal deltas, mangroves, seagrass beds and coral reefs, it can be found that the **coastal tree-related wetlands** only include mangrove forest and **no other coastal swamp**. 2) We currently defined coastal wetlands into mangrove, saltmarsh and tidal flat **because there is almost no global/regional coastal swamp products can be used**. And the coastal wetland system of three subcategories (mangrove, salt marsh and tidal flat) is also widely recognized in many previous studies (Murray et al. 2022, Zhang et al., 2022).

Murray, N. J., Worthington, T. A., Bunting, P., Duce, S., Hagger, V., Lovelock, C. E., ... & Lyons, M. B. (2022). High-resolution mapping of losses and gains of Earth's tidal wetlands. Science, 376(6594), 744-749.

Zhang, Z., Xu, N., Li, Y., & Li, Y. (2022). Sub-continental-scale mapping of tidal wetland composition for East Asia: A novel algorithm integrating satellite tide-level and phenological features. Remote Sensing of Environment, 269, 112799.

2. As mentioned, the wetlands have clear seasonal changes within a year?

Great thanks for the comment. After carefully checking our manuscript, the manuscript didn't state that **the wetland have clear seasonal changes**, instead, we emphasize that the spectra variability of wetlands is simultaneously affected by the **water-level and phenology changes**. Therefore, we use time-series Landsat and Sentinel-1 to generate the lowest and highest water-levels:

The spectral characteristics of the wetlands would quickly change along with the seasonal or daily water levels of the underlying surface. For example, the tidal flat was the status of seawater at the high tidal stage and mud or sand flats at low tidal stages (Wang et al., 2021); therefore, it was necessary to **extract the highest and lowest water-level composites to completely capture these inundated wetlands**. Over the past several years, the time-series compositing strategy has been widely used to capture phenological and cloud-free composites. Derived the phenological features from time-series Landsat imagery as:

Many studies also demonstrated that a multi-temporal phenology was also essential for classifying the vegetated wetlands and excluding these non-wetland land-cover types (Li et al., 2020; Ludwig et al., 2019). There were

usually two options for capturing phenological features from time-series Landsat imagery. These included seasonal-based compositing (Zhang et al., 2021a; Zhang et al., 2022a) and percentile-based compositing (Hansen et al., 2014; Zhang and Roy, 2017; Zhang et al., 2021b). The former used the phenological calendar for selecting time-matched imagery. It then adopted the compositing rule to capture the seasonal features, while the latter directly used the statistical distributions to select various percentiles. Azzari and Lobell (2017) quantitatively analyzed the performance of two compositing methods and found that both of them had similar mapping accuracy for land-cover mapping. Meanwhile, the seasonal-based compositing method needed the prior phenological calendar, while the percentile compositing method did not require any prior knowledge or explicit assumptions regarding the timing of the season; therefore, the percentile compositing method was more suitable to generate phenological features. This study composited time-series Landsat reflectance bands and four spectral indexes into five percentiles (15th, 30th, 50th, 70th and 85th). It should be noted that the minimum and maximum percentiles were excluded because they were usually affected by residual clouds, shadows, and saturated observations.

How did you define the final date for the wetland product?

Great thanks for the comment. The global wetland product is developed for the nominal year of 2020, because the time-series Landsat imagery and Sentinel-1 SAR observations, used in this study, are mainly around 2020. It has been explained in the manuscript as:

First, all available Landsat imagery, including Landsat 7 ETM+ and Landsat 8 OLI missions, **during 2019–2021 was obtained for the nominal year of 2020 via the Google Earth Engine platform** for minimizing the influence of frequent cloud contamination in the tropics and snow and ice in the high latitudes.

The description of Sentinel-1: All the **time-series Sentinel-1 imageries archived on the GEE platform in 2020** in Interferometric Wide Swath mode with a dual-polarization of VV and VH were used.

3. Besides the time-series feature? Did you consider other features

Thanks for the comment. In this study, we used the time-series Landsat imagery to generate the lowest and highest water-level composites and multiple phenological features, used the time-series Sentinel-1 imagery to generate the lowest and highest water-level features, and used the ASTER GDEM to derive elevation, slope and aspect. To intuitively understand all training features, the 77 multisourced training features were listed in a table as:

In summary, a total of 77 multisource training features (listed in Table 3), including 70 optical features from Landsat imagery, 4 SAR features from Sentinel-1 imagery and 3 topographical features from ASTER GDEM.

**Table 3**. The multisourced and multitemporal training features for wetland mapping.

| Data | Derived training features from multisource remote sensing imagery |
|---|---|
| Landsat | **Water-level features**: the lowest and highest composites with Blue, Green, Red, NIR, SWIR1, SWIR2, LSWI, NDWI, NDVI and EVI bands
**Phenological features:** 5th, 30th, 50th, 70th and 85th percentiles with Blue, Green, Red, NIR, SWIR1, SWIR2, LSWI, NDWI, NDVI and EVI bands |
| Sentinel-1 SAR | **Water-level features**: the lowest and highest composites using 5th and 95th percentiles for VV and VH bands. |
| ASTER GDEM | **Topographical features**: elevation, slope and aspect. |

4. How did you training the model? One model for the global wetland or one model per grid? There are distinctly phenological differences for the different wetland types and even the same wetland type. Please clarify it.

Great thanks for the comment. We used 961 local adaptive classification models in 961 5° × 5° geographical tiles after considering the phenological differences for the different wetland types and even the same wetland type at spatial dimension. The local adaptive modeling has been strengthen as:

Since we have simultaneously extracted the maximum coastal and inland wetland extents when deriving training samples from prior wetland datasets, the stratified classification strategy was adopted to fully use the maximum extent constraint. If a pixel was classified as a coastal wetland outside the maximum coastal wetland extents, it would be identified as a misclassification. Furthermore, there were two ideas for the large-area land-cover mapping including global classification modeling (using one universal model for the whole areas) and local adaptive modeling (using various models for different local zones) (Zhang et al., 2020). For example, Zhang and Roy (2017) demonstrated that local adaptive modeling outperformed the global classification modeling strategy. Therefore, the global land surface was first divided into 961 5° × 5° geographical tiles illustrated in Figure 5, which were inherited from the global 30 m land-cover mapping by (Zhang et al., 2021b). Then, **we trained the local adaptive classification models using derived training samples in Section 3 and multisource and multitemporal features (the highest, lowest water-level and phenological composites and topographical variables) at each 5° × 5° geographical tile.** It should be noted that we used the training samples from neighboring 3 × 3 geographical tiles to train the classification model and classify the central tile for guaranteeing the spatially continuous transition over adjacent regional wetland maps. Namely, we trained 961 local adaptive classification models and then produced 961 5° × 5° wetland maps. Finally, we spatially mosaiced these 961 regional wetland maps into the global 30 m wetland map in 2020.

[Figure]

Figure 5. The spatial distribution of 961 5° × 5° geographical tiles used for local adaptive modeling, which was inherited from the global 30 m land-cover mapping by (Zhang et al., 2021b). The background imagery came from the National Aeronautics and Space Administration (https://visibleearth.nasa.gov, last access: 10 Nov 2022).

5. The authors used Sentinel SAR data, why do not produce the 10 m resolution product based on Sentinel 2?

Great thanks for the suggestion. Yes, we can combine the time-series Sentinel-1 and Sentinel-2 to develop the global 10m wetland maps, however, the reasons why we developed the global 30m wetland products are the following:

1) The spatial resolution of most prior global wetland products in Table 1 is 30 m, the derived training samples can be directly applied in the Landsat imagery for wetland mapping. If we used the derived training samples to Sentinel-1 and Sentinel-2, we must consider the spatial scale matching problem.

2) Compared to the Landsat imagery, the Sentinel-2 preprocessing method is not yet mature, namely, these bad quality (cloud, shadow, snow and ice) cannot be completely identified. It might be transferred to the wetland mapping especially in the tropics (cloudy regions) and high latitudes areas (frequent snow and ice covering).

3) The global 10-m wetland mapping also means 10 times the amount of computation when comparing to the global 30 m wetland mapping. Although the GEE provides free computation and storage ability, the time consumption is also a factor that cannot be ignored in global wetland mapping

4) Our further works would focus on the spatiotemporal dynamics of global wetlands over long time spans, which is also a hot spot in wetland monitoring today, however, the Sentinel-2 has shorter time span than the Landsat (since 1984).

Based on the above factors, we choose the Landsat imagery as our main data to develop the global 30 m fine wetland mapping.

---

## Author Comment (AC2)

**Response to comments**
**Paper #:** essd-2022-180
**Title:** GWL_FCS30: global 30 m wetland map with fine classification system using multi-sourced and time-series remote sensing imagery in 2020
**Journal**: Earth System Science Data

**Reviewer #2**

The submitted manuscript provides a global wetland map including inland and tidal sub-classes based on remote sensing data. Currently, we are still lacking a multi-class global wetland data including inland and tidal wetlands simultaneously, and the map produced by this work provides valuable information for related wetland studies. The manuscript is well-written and easy to follow. Above all, I recommend their publication provided that a moderate revision is carried out.

Great thanks for the positive comments. The manuscript has been further improved based on your and other two reviewers' comments and suggestions.

1. Wetlands are classified as inland or coastal wetlands in this study, and the latter includes mangroves, salt marshes, and tidal flats. For these three wetland types, the term "tidal wetlands" is more appropriate than "coastal wetlands", for example, in Murray et al., 2022. Coastal wetlands include other terrestrial and shoreline constituents like riparian wetlands and tidal freshwater marshes, but not just mangroves, salt marshes and tidal flats. As such, I suggest using "tidal wetlands" to make the classification system more accurate.

Great thanks for your useful suggestion. The 'coastal wetland' has been changed as the 'tidal wetland' in our fine wetland classification system as:

**Table 2**. The description of wetland classification system in this study

| Category I | Category II | Description |
| --- | --- | --- |
| **Tidal wetland** | Mangrove | The forest or shrubs which grow in the coastal blackish or saline water |
| | Salt marsh | Herbaceous vegetation (grasses, herbs and low shrubs) in the upper coastal intertidal zone |
| | Tidal flat | The tidal flooded zones between the coastal high and low tide levels including mudflats and sandflats. |
| Inland wetland | Swamp | The forest or shrubs which grow in the inland freshwater |
| | Marsh | Herbaceous vegetation (grasses, herbs and low shrubs) grows in the freshwater |
| | Flooded flat | The non-vegetated flooded areas along the rivers and lakes |
| | Saline | Characterized by saline soils and halophytic (salt tolerant) plant species along saline lakes |
| | Permanent water | Lakes, rivers and streams that are always flooded |

2. Section 2.4: This section is about generating validation samples, thus should be moved to the "Accuracy assessment" section as a validation step. Another thing is how did the authors determine the size of total validation samples (i.e., 18,701)?

Great thanks for the comment. First, Based on the suggestion, the section 2.4 of how to generate the global validation samples has been moved to the Section 4.3 Accuracy Assessment.

Then, as for how to determine the size of total validation samples, we combined the stratified random sampling method and the proportions of various land-cover types to determine the sample size of each land-cover type based on the work of Foody et al. (2009) and Olofsson et al. (2014) as:

$$n = \frac{\left( \sum W_h \sqrt{p_h(1-p_h)} \right)^2}{V + \sum W_h P_h (1-P_h)/N}$$

where $N$ is the number of pixel units in the study region; $V$ is the standard error of the estimated overall accuracy that we would like to achieve, $V = (d/t)^2$ ($t$ = 1.96 for a 95% confidence interval, $t$ = 2.33 for a 97.5% confidence interval, and $d$ is the desired half-width of the confidence interval); $W_h$ is the weight distribution of class $h$; $p_h$ is the producer's accuracy. These sample size calculations should be repeated for a variety of choices of $V$ and $p_h$ before reaching a final decision. We try to achieve producer's accuracies of 0.9 of non-wetland class and 0.8 of the seven wetland classes. Meanwhile, using the parameters of $d$ = 0.0125, $t$ = 2.33, the sample size can be determined as approximately 18500. In addition, there is a little uncertainty for interpreting the validation points, so we randomly generate 20000 validation points over the globe and then discard 1299 uncertain points (these disagreement points over five experts), so a total of 18701 validation points are used to assess the GWL_FCS30-2020 performance.

Pontus Olofsson, G. M. F. (2014). Good practices for estimating area and assessing accuracy of land change. Remote Sensing of Environment, 148(25), 42-57, https://doi.org/10.1016/j.rse.2014.02.015.
Foody, Giles M. "Sample size determination for image classification accuracy assessment and comparison." International Journal of Remote Sensing 30.20 (2009): 5273-5291.

This amount seems disproportionately less than the number of training samples (more than 20 million).

As for the unbalance of the training samples and validation samples, it is mainly because our training and validation samples are completely independent. Specifically, **we combined many pre-existing global wetland datasets to automatically derive the training samples over the globe** while **the validation points must be interpreted by visual interpretation**. As we all known, collecting validation points through visual interpretation is time-consuming and labor-intensive, therefore, we cannot to interpret a large amount of validation points.

(3) Lines 250-255: The tidal flat samples were collected from the global tidal flat map (Murray et al., 2019), and thus would suffer from the inherent error of the data. Several studies found that Murray's tidal flat map failed to distinguish between nearshore ponds and tidal flats, mainly because these ponds also have water-level variations (Jia et al., 2021; Zhang et al., 2022). The error of commission (i.e., classifying ponds into tidal flats) is also indicated in the tidal flat map generated by this study, as shown in the upper panels of Fig. 13. I suggest the authors mask out ponds and lakes from their tidal flat map because it would substantially improve the accuracy. There is a new dataset that provides global lakes and reservoirs may be helpful: Khandelwal et al. 2022.

Great thanks for the comment and useful suggestion. Yes, we agree that the Murray's tidal flat suffered the commission error especially over the nearshore ponds. Based on your suggestion, the new global lakes and reservoirs dataset is used to further optimize tidal flat layer in our GWL_FCS30.

In addition, as the tidal flats were demonstrated to overestimate some coastal pones as the tidal flats, the global lake and reservoir dataset, developed by Khandelwal et al. (2022), was applied to optimize the tidal flat.

The local comparisons in the Figure 16 shows that the updated GWL_FCS30 dataset has better performance than Murray's tidal flat products in excluding these ponds and lakes.

[Figure]

Figure 16. The comparisons between the tidal flat of GWL_FCS30 in 2020, Murray's tidal flat V1.1 in 2016 (Murray et al., 2019), and Murray's tidal flat V1.2 in 2019 (Murray et al., 2022) for two local regions. In each case, the highest and lowest tidal-level composites, composited by SWIR1, NIR, and red bands, are illustrated.

(4) Line 286: These thresholds proposed by Wang et al. 2020 were designed for tidal wetlands, but their application in this study was to inland wetlands. Therefore, the authors need to prove that these thresholds have robust performance in mapping inland wetlands.

Great thanks for the comment. Yes, the rule of 'EVI≥0.1, NDVI≥0.2, and LSWI>0' is referenced from the work of Wang et al. (2020) in tidal wetland mapping, actually, whether the rule is also suitable for inland wetlands has been demonstrated on the work of Xiao et al. (2009) and Hao et al. (2022) who used these thresholds to identify the vegetated land-cover types over the inland regions.

Wang, X., Xiao, X., Zou, Z., Hou, L., Qin, Y., Dong, J., Doughty, R. B., Chen, B., Zhang, X., Chen, Y., Ma, J., Zhao, B., and Li, B.: Mapping coastal wetlands of China using time series Landsat images in 2018 and Google Earth Engine, ISPRS J Photogramm Remote Sens, 163, 312-326, https://doi.org/10.1016/j.isprsjprs.2020.03.014, 2020.

Xiao, Xiangming, et al. "A simple algorithm for large-scale mapping of evergreen forests in tropical America, Africa and Asia." Remote Sensing 1.3 (2009): 355-374.

Hao, Ying-Ying, et al. "A cascading reaction by hydrological spatial dynamics alternation may be neglected." Environmental Research Letters 17.8 (2022): 084034.

Meanwhile, we also use these thresholds to split the vegetated and non-vegetated areas over several inland regions (including: Poyang Lake, Caspian Sea, Congo Rainforests and so on), Figure S1 illustrates that these thresholds are also robust in splitting vegetated and non-vegetated land-cover types in inland areas. For example, in the First panel over Poyang Lake, the non-vegetated areas (water body, impervious surfaces) are both clearly excluded and these cropland, forest and grassland are completely included. In the second panel over semi-arid region, the bare area and water body are masked while the sparse vegetation (upper left) and inland marsh are included. The third panel in the Congo rainforests, these small rivers and reservoirs are accurately captured.

[Figure]

Figure S1. The vegetated and non-vegetated masks (white and black) over three typical inland areas using the rule of 'EVI≥0.1, NDVI≥0.2, and LSWI>0'.

(5) Line 297, Equation 3: This maximum extent of inland wetlands also contains tidal wetlands (since the wetland layer in the global land cover data failed to distinguish them), so how did the authors ensure that the generated samples from inland wetland have corrected labels?

Great thanks for the comment. Yes, the maximum extent of inland wetlands also contains a small amount of tidal wetlands. However, we derive inland training samples from five inland wetland products using a series of refinement measures **instead of directly generating from the inland maximum wetland extents**. Specifically, the consistency analysis of five global wetland datasets (TROP-SUBTROP Wetland, GLWD, CCI_LC, GlobeLand30, and GLC_FCS30) and the temporal stability checking for CCI_LC (1992–2020), GlobeLand30 (2000-2020) and GLC_FCS30 (2015-2020) were applied to identify these temporally stable and high cross-consistency wetland points. It should be noted that the coarse wetland products (GLWD, TROP-SUBTROP and CCI_LC) were resampled to 30 m using the nearest neighbor method on the GEE platform and the coastal wetland layers in these products were excluded. Namely, only the pixel identified as inland wetland in all five products was retained. Then, the morphological erosion filter with a local window of 3 × 3 was also used to decrease the sampling uncertainty over these land-cover transition areas because the transition zones between two different land-cover types are likely to be misclassified. The details of how to derive inland training samples has been strengthen as:

The pre-existing inland wetland datasets usually suffered from lower accuracy compared to coastal wetland products; for example, the wetland layer in the GlobeLand30-2010 and GLC_FCS30-2015 was validated to achieve a user accuracy of 74.9% (Chen et al., 2015) and 43.4% (Zhang et al., 2021b), respectively. Therefore, **we first generated high-confidence inland wetland samples and then determined their sub-categories (swamp, marsh, inland flat, saline wetland and permanent water). Specifically, the consistency analysis of five global wetland datasets (TROP-SUBTROP Wetland, GLWD, CCI_LC, GlobeLand30, and GLC_FCS30) and the temporal stability checking for CCI_LC (1992–2020), GlobeLand30 (2000-2020) and GLC_FCS30 (2015-2020) were applied to identify these temporally stable and high cross-consistency wetland points ($P_{inlandWet}^{Tstable,Scons}$).** It should be noted that the coarse wetland products (GLWD, TROP-SUBTROP and CCI_LC) were resampled to 30 m using the nearest neighbor method on the GEE platform and the coastal wetland layers in these products were excluded. **Only the pixel identified as inland wetland in all five products was retained. Then, the morphological erosion filter with a local window of $3 \times 3$ was also used to decrease the sampling uncertainty over these land-cover transition areas because the transition zones between two different land-cover types are likely to be misclassified** (Lu and Wang, 2021; Radoux et al., 2014).

Afterward, to determine the wetland sub-category for each inland wetland sample, we first used the empirical vegetation rule (EVI ≥ 0.1, NDVI ≥ 0.2, and LSWI > 0) proposed by Wang et al. (2020) and time-series Landsat imagery to split candidate samples into two parts: vegetated wetland samples (swamp and marsh) and non-vegetated wetland samples (flooded flat, saline and permanent water). Then, as the swamp was defined as the forest or shrubs which grow in the inland freshwater, the global 30-m tree cover dataset (GFCC30TC) was adopted to distinguish the swamp and marsh from vegetated wetland samples. Specifically, if the tree cover of the sample was greater than 30% (Hansen et al., 2013), it was labeled as swamp, and the remaining vegetated wetland samples were labeled as marsh. Furthermore, to distinguish between the inland flat, saline samples and permanent water, the saline blocks in the prior GLWD products were first checked by visual interpretation and then imported as the reference dataset to identify all saline wetland samples. The remaining non-vegetated wetland samples were further refined using the time series of the JRC-GSW datasets, only water probability of these remaining samples less than the threshold of 0.95 (suggested by Wang et al. (2020)) were labeled as flooded flat. Lastly, regarding the permanent water samples, the JRC_GSW water dynamic dataset was validated and achieved producer's and user's accuracies of 99.7% and 99.1% for permanent water (Pekel et al., 2016). The permanent water training samples were directly derived from the JRC_GSW dataset without any refinement rules.

Lastly, although the maximum extent of inland wetlands (Eq. (3)) contains tidal wetlands, our post-processing method also minimize this issue in Section 4.2 as:

As the inland and coastal tidal wetlands were independently produced, some pixels in the overlapping area of maximum inland and coastal wetland extents were simultaneously labeled as inland wetlands and coastal wetlands. However, as the final global wetland map was a hard classification, **these pixels should be post-processed into one label. As the random forest classifier could provide the posterior probability for each pixel, we determined the labels of the confused pixels by comparing the posterior probabilities.**

(6) Section 4.2: The description for obtaining training samples is unclear. What are the strata here, wetland classes or 5°×5° tiles?

Great thanks for the comment. The description of how to obtain the training samples has been strengthen by your and other reviewer's suggestions. Specifically, we further adjust the Section 3 (Deriving training samples and determining maximum wetland extents) into four parts. In the first three parts, we separately introduce how to derive coastal tidal wetland samples in Section 3.1, inland wetland samples in Section 3.2, and non-wetland samples in Section 3.3, and determine the sample size and distributions. We think the updated manuscript in Section 3 is easier to follow.

As for 'What are the strata here' in Section 4.2, we actually simultaneously consider the wetland classes and 5°×5° tiles. To make the local adaptive and stratified modeling more intuitive, the Section 4.2 has been strengthen as:

Since we have simultaneously extracted the maximum coastal and inland wetland extents when deriving training samples from prior wetland datasets, the stratified classification strategy was adopted to fully use the maximum extent constraint. If a pixel was classified as a coastal tidal wetland outside the maximum coastal tidal wetland extents, it would be identified as a misclassification. Furthermore, there were two ideas for the large-area land-cover mapping including global classification modeling (using one universal model for the whole areas) and local adaptive modeling (using various models for different local zones) (Zhang et al., 2020). For example, Zhang and Roy (2017) demonstrated that local adaptive modeling outperformed the global classification modeling strategy. Therefore, the global land surface was first divided into 961 5° × 5° geographical tiles illustrated in Figure 5, which were inherited from the global 30 m land-cover mapping by (Zhang et al., 2021b). Then, we trained the local adaptive classification models using derived training samples in Section 3 and multisource and multitemporal features (the highest, lowest water-level and phenological composites and topographical variables) at each 5° × 5° geographical tile. It should be noted that we used the training samples from neighboring 3 × 3 geographical tiles to train the classification model and classify the central tile for guaranteeing the spatially continuous transition over adjacent regional wetland maps. Namely, we trained 961 local adaptive classification models and then produced 961 5° × 5° wetland maps. Finally, we spatially mosaiced these 961 regional wetland maps into the global 30 m wetland map in 2020.

[Figure]

Figure 5. The spatial distribution of 961 5° × 5° geographical tiles used for local adaptive modeling, which was inherited from the global 30 m land-cover mapping by (Zhang et al., 2021b). The background imagery came from the National Aeronautics and Space Administration (https://visibleearth.nasa.gov, last access: 10 Nov 2022).

In addition, the training samples selected from the maximum wetland extent may be of low quality. The authors do explain that the map accuracy is insensitive to low-quality samples within a 20% threshold, but it's still missing a map representing the percentage of real erroneous samples. I think the training samples need to be filtered according to some criterion before classification to improve their accuracy. I recommend clarifying the process of sample generating and the quality-control procedures.

Great thanks for the comment. Yes, we agree that the quality of training samples is important for accurate wetland mapping. In this study, we have used a lot of rules to guarantee the confidence of training samples instead of directly deriving from maximum wetland extents.

Firstly, as for **the mangrove training samples**:

[revised manuscript text omitted]

Also, a product of global tidal wetland dynamics provided by Murray et al. (2022) could be an important reference for comparison.

Great thanks for the comments. Based on your suggestion, the new global tidal flats in Murray et al. (2022) has been added into the comparisons.

Figure 16 illustrated the comparisons between GWL_FCS30 tidal flat layer with the Murray's tidal flat V 1.1 in 2016 and the updated Murray's tidal flat V1.2 in 2019 (Murray et al., 2022) in two local regions, and the corresponding highest and lowest tidal-level composites are also listed. Overall, three products can comprehensively capture the spatial patterns of tidal flats in these two regions, and the GWL_FCS30-2020 and Murray's tidal flat V1.2 performed higher spatial consistency while the Murray's tidal flat V1.1 suffered the obvious omission error in three typical areas (red rectangles). Detailedly, we can find that the Murray's tidal flat products misclassified some coastal ponds and lakes into the tidal flats especially in the first region while the GWL_FCS30-2020 accurately excluded these ponds and lakes.

[Figure]

Figure 16. The comparisons between the tidal flat of GWL_FCS30 in 2020, Murray's tidal flat V1.1 in 2016 (Murray et al., 2019), and Murray's tidal flat V1.2 in 2019 (Murray et al., 2022) for two local regions. In each case, the highest and lowest tidal-level composites, composited by SWIR1, NIR, and red bands, are illustrated.

(8) Figure 8 lacks a legend.

Great thanks for pointing out the problem. The legend has been added in the Figure 8 as:

[Figure]

Figure 11. The area proportions of eight wetland sub-categories over each continent.

---

## Author Comment (AC3)

**Response to comments**

**Paper #:** essd-2022-180
**Title:** GWL_FCS30: global 30 m wetland map with fine classification system using multi-sourced and time-series remote sensing imagery in 2020
**Journal**: Earth System Science Data

**Reviewer #3**

The authors developed a global wetland mapping product based on multiple approaches in the GEE environment, called the GWL_FCS30. They reported some 3.6 million km2 of global wetlands, making the data freely available. The authors' efforts are laudable, yet I have many concerns about the presentation and the analyses themselves that preclude my acceptance of this paper for publication. For the presentation, I would argue that the paper itself is overly long and dense. The approaches could be more clearly articulated and sign-posted for the readers. Parts that are results are introduced in the Discussion section (e.g., some validation data, as I note below) and the length of the paper makes it a long slog. However, my main issues are with the analytical approaches and base assumption.

Great thanks for the comment. The manuscript has been greatly improved based on your and two other reviewers' comments.

First, the authors introduce wetlands in the very first sentence using the Ramsar Convention definition to include waters up to 6 m in depth. Then, they go on to conduct their analysis but exclude any and all inland open waters as they are assumed to be greater than 6 m in depth. They backstop their findings on global wetland abundance by stating at L535 "the estimated total wetland area in this study was more reasonable [than four previous analyses] because permanent water bodies with depths of more than six meters were not considered wetlands, according to the RAMSAR (sic) Convention…". The assumption that any and all open water on the global landmass is >6m in depth – and hence not possibly a wetland – does not resonate. Yes, larger and deeper lakes could be greater than 6m. But open waters, especially smaller ones are frequently considered wetlands and are typically <6m in water depth (see, e.g., China's State Forestry Administration [www.forestry.gov.cn] or a recent paper by Ye et al. (2022, https://doi.org/10.3390/w14071152); see also the Canadian Wetland Inventory [https://open.canada.ca/data/en/dataset/09f46d71-6feb-4f8f-8eb5-a58a58b06af5] or the United States National Wetlands Inventory [https://www.fws.gov/program/national-wetlands-inventory] identifying open waters as a wetland type). The point is that the Ramsar definition of wetlands is used, but then a major type of wetlands are excluded. The authors must acknowledge this in their study. For instance, it could be noted in the title and should definitely be noted in the abstract. I do wish that the authors would redo their analysis and incorporate open waters as a wetland type to include a major wetland type in their global analysis, alas.

Great thanks for pointing out this issue and giving useful suggestion. The **permanent water body** has been added into our fine wetland classification system in method Section as:
In this study, after considering the applicability of moderate resolution (10–30 m) imagery, their practical use for ecosystem management, and the available pre-existing global wetland dataset, the fine wetland classification system, containing eight sub-categories (three coastal tidal sub-categories and five inland sub-categories), was proposed to comprehensively depict the spatial patterns of global wetlands (Table 2). Specifically, the subcategories of coastal tidal wetlands consist of mangroves, salt marshes, and tidal flats. By importing the vegetation and water cover information associated with this land cover, these categories were widely recognized in many previous studies (Wang et al., 2021; Zhang et al., 2022b). The inland wetland types shared similar characteristics and were grouped into swamp, marsh, and flooded flat. Meanwhile, in order to capture saline soils and halophytic plant species along saline lakes, the inland saline wetland, inherited from the Global Lakes and Wetlands Dataset (GLWD) (Lehner and Döll, 2004), was also imported. **Lastly, the permanent water, including lakes, rivers and streams that are always flooded, was widely identified as a wetland layer in previous studies (Davidson, 2014; Dixon et al., 2016; Hu et al., 2017b)**.

**Table 2**. The description of wetland classification system in this study

| Category I | Category II | Description |
|---|---|---|
| Tidal wetland | Mangrove | The forest or shrubs which grow in the coastal blackish or saline water |
| | Salt marsh | Herbaceous vegetation (grasses, herbs and low shrubs) in the upper coastal intertidal zone |
| | Tidal flat | The tidal flooded zones between the coastal high and low tide levels including mudflats and sandflats. |
| Inland wetland | Swamp | The forest or shrubs which grow in the inland freshwater |
| | Marsh | Herbaceous vegetation (grasses, herbs and low shrubs) grows in the freshwater |
| | Flooded flat | The non-vegetated flooded areas along the rivers and lakes |
| | Saline | Characterized by saline soils and halophytic (salt tolerant) plant species along saline lakes |
| | **Permanent water** | **Lakes, rivers and streams that are always flooded** |

Meanwhile, after adding the permanent water into our wetland system, the Result Section has been revised as: Figure 9 illustrates the spatial distributions of our GWL_FCS30 wetland map and their area statistics in latitudinal and longitudinal directions in 2020. Overall, the GWL_FCS30 map accurately captured the spatial patterns of wetlands. It mainly concentrated on the high latitude areas in North Hemisphere and the rainforest areas (Congo Basin and Amazon rainforest in South America). Quantitatively, according to the latitudinal statistics, approximately 72.96% of wetlands were distributed poleward of 40°N (a large number of wetlands are located in Canada and Russia), and 10.6% of wetlands were located in equatorial areas, between 10°S~10°N, within which the Congo and Amazon rainforest wetlands are located. As for the longitudinal direction, there were mainly four statistical peak intervals: 120°W~50°W (Canada wetlands and Amazon wetlands), 15°E~25°E (Congo wetlands), 40°E~55°E (the Caspian Sea), and 60°E~90°E (Russia wetlands). Afterward, to more intuitively understand the performance of our GWL_FCS30 wetland map, four local enlargements in Florida, the Congo Basin, Sundarbans, and Poyang Lake were also illustrated. All of them comprehensively captured the wetland patterns in these local areas. For example, there was significant consistency between our results and Hansen's regional wetland maps in the Congo Basin (Bwangoy et al., 2010); both results indicated that the wetlands occurred closer to major rivers and floodplains. Next, according to the lowest and highest water-level features derived from Sentinel-1 SAR and Landsat optical imagery in Figure 4, the inland wetlands, varied with the water-levels, were also comprehensively identified in the Poyang wetland map (Figure 9d). Figure 9c illustrates the spatial distributions of the world's largest mangrove forest in the Sundarbans (Figure 9c), and the

cross-comparison in Figure 14 also demonstrates the great performance of the GWL_FCS30 dataset. Lastly, the Florida wetlands simultaneously contained six sub-categories (mangrove, tidal flat, salt marsh, marsh, permanent water and swamp). These were distributed along the coastlines and rivers and are accurately captured in Figure 9a.

[Figure]

Figure 9. The overview of global 30-m fine wetland maps and their area statistics in latitudinal and longitudinal directions in 2020. Four local enlargements in (a) Florida, (b) Congo Basin, (c) Sundarbans, and (d) Poyang Lake were also illustrated.

Figure 10 illustrates the spatial distribution of eight sub-category wetlands after aggregating to the 0.5° × 0.5° grid cell. Intuitively, permanent water body, swamp and marsh accounted for most inland wetlands, and all of them showed significant spatial coexistence, in which they mainly concentrated on the . In contrast, flooded tidal wetlands had obviously lower proportions, and the inland saline type was only distributed along the surroundings of several saline lakes. In terms of the spatial distribution, it can be found that: 1) the swamp wetlands mainly were concentrated in the Congo and Amazon rainforests, Southern United States, and Northern Canada; 2) most marsh wetlands were located in high latitude areas in the Northern Hemisphere including Northern Canada, Russia, and Sweden; 3) there were significant coexistent relationships between flooded flat, swamp, and marsh wetlands. Similar to coastal wetlands, the mangrove forests were only found in coastal areas below 30°N and were mainly concentrated in regions between 30°N ~ 30°S, including Southeast Asia, West Africa, and the east coast of South America. The salt marshes and tidal flats shared similar spatial distributions.

They were widely distributed globally and can be observed along most coastlines. In addition, the tidal flat distributions were closely related to the slope of coastlines, tidal ranges, and sediment inflows. For example, the tidal flats in Asia and Europe usually were located in the tide-dominated estuaries and deltas. Similarly, Murray et al. (2019) also demonstrated that there were often more tidal flats where the river flowed into the sea.

[Figure]

Figure 10. The spatial distributions of the eight wetland sub-categories after aggregating them to a resolution of $0.5° \times 0.5°$.

A further issue I have with this paper is that the data are considered mis-classified if they occur as wetlands in an area outside the [wetland type] maximum extent. However, this max extent assumes that all the previous analyses had zero omission error.

Great thanks for the comment. In this study, as the maximum extents of inland/coastal wetlands derived by combining several global prior products, the omission error in each prior product might be complemented by other products. For example, the inland maximum extent is derived from five products (TROP-SUBTROP Wetland, GLWD, CCI_LC, GlobeLand30, and GLC_FCS30). The CCI_LC, GlobeLand30 and GLC_FCS30 had serious omission errors, but the GLWD and TROP-SUBTROP products, produced by the compilation and model simulation method (Gumbricht, 2015; Lehner and Döll, 2004), can capture almost all wetland areas at the expense of a higher commission error. On the other hand, the union of five global wetland datasets in Eq. (3) also minimized the omission error of each dataset for inland wetland sub-categories. Therefore, the derived inland maximum extents actually fulfilled the assumption of zero omission error. The rationality of the maximum extents has been added and discussed in the Discussion Section as:

In addition, we used the derived maximum extents as the boundary for identifying inland and coastal tidal wetlands, in other words, we assumed that the derived maximum extents contained all inland and coastal tidal wetlands with zero omission error. Actually, the inland maximum extents in Eq. (3) fulfilled the assumption of zero omission error, because the GLWD and TROP-SUBTROP products, produced by the compilation and model simulation method (Gumbricht, 2015; Lehner and Döll, 2004), can capture almost all wetland areas at the expense of a higher commission error. For example, the Figure 13 illustrated the cross-comparisons between

our GWL_FCS30 wetland maps with four existing wetland products, and the GLWD obviously overestimated the inland wetlands. On the other hand, the union of five global wetland datasets in Eq. (3) also minimized the omission error of each dataset for inland wetland sub-categories. Next, as for the maximum mangrove forest extents (Eq. (1)), as the high producer's and user's accuracies were achieved by five prior mangrove products (explained in Section 2.2) and the time-series mangrove products were integrated that these missed mangroves may be complemented by other products or time-series products, the derived maximum extents also can be considered as zero omission error and covered almost all mangrove forests. Recently, Bunting et al. (2022) developed the newest mangrove products covering 1996-2020, it can be used as another important prior dataset in our further works for deriving the maximum mangrove extents. Lastly, the maximum tidal flat extents, derived from time-series Murray's products from 1985~2016 by using the union operation (Eq. (2)), can also contain almost all tidal flats because previous studies demonstrated that they suffered higher commission error than the omission error (Jia et al., 2021; Zhang et al., 2022b).The missed tidal flats would concentrate on these newly increased tidal flats during 2016-2020, fortunately, the new time-series global tidal flat products during 1999-2019 was developed (Murray et al., 2022) and can be used as an important supplement in our further work for deriving the maximum tidal flat extent with zero omission error.

Another concern of mine is that their error assessment was done using a relatively paltry number of wetlands for the global extent of their analysis. For instance, they have ~8,000 wetland validation points to cover seven different wetland types. From Figure 2, it appears that ~7,000 of these points are inland "wetlands" versus coastal systems. Even with 7000 points for validation, that seems small considering the global extent of inland systems (swamps, marshes, flooded flats). And ~1000 points are used to validate the global population of saline, salt marsh, mangrove, and tidal flats. Their validation points were visually validated – though the authors explain five experts had to agree on the typology, the disagreements or data supporting those validations are also not presented.

Great thanks for the comment. First, we agree that a large amount of validation points play great role in comprehensively assess the performance of the developed products, however, it should be noted that the collection of validation points, especially for water-level sensitive wetlands with fine classification system, is time-consuming and labor-intensive. In addition, Foody et al. (2009) and Olofsson et al. (2014) have detailedly described how to determine the size of total validation points by using stratified random sampling theory as:

$$ n = \frac{\left( \sum W_h \sqrt{p_h(1 - p_h)} \right)^2}{V + \sum W_h P_h(1 - P_h)/N} $$

where $N$ is the number of pixel units in the study region; $V$ is the standard error of the estimated overall accuracy that we would like to achieve, $V = (d/t)^2$ ($t$ = 1.96 for a 95% confidence interval, $t$ = 2.33 for a 97.5% confidence interval, and $d$ is the desired half-width of the confidence interval); $W_h$ is the weight distribution of class $h$; $p_h$ is the producer's accuracy. These sample size calculations should be repeated for a variety of choices of $V$ and $p_h$ before reaching a final decision. We try to achieve producer's accuracies of 0.9 of non-wetland class and 0.8 of the seven wetland classes. Meanwhile, using the parameters of $d$ = 0.0125, $t$ = 2.33, the sample size can be determined as approximately 18700.

Pontus Olofsson, G. M. F. (2014). Good practices for estimating area and assessing accuracy of land change. Remote Sensing of Environment, 148(25), 42-57, https://doi.org/10.1016/j.rse.2014.02.015.

Foody, Giles M. "Sample size determination for image classification accuracy assessment and comparison." International Journal of Remote Sensing 30.20 (2009): 5273-5291.

In order to make the validation assessment more comprehensive, we also replenish 7008 wetland validation points, including 212 non-wetland points and 6796 wetland points (4538 inland wetland points and 2258 tidal wetland points), and the description of these updated global validation points (25709 points) has been revised as:

To quantitatively analyze the performance of our GWL_FCS30 wetland map, a total of 25,709 validation samples (illustrated in Figure 6), including 10,558 non-wetland points and 15,151 wetland points, were collected. Firstly, as the wetland was sparse land-cover type compared to the non-wetlands (forest, cropland, grassland and bare land), the stratified random strategy was applied to randomly derive validation points at each strata. Then, as the wetlands had significant correlation with the water levels (Zhang et al., 2022b), the time-series optical observations archived on the GEE cloud platform were used as the auxiliary dataset to interpret these water-level sensitive wetlands such as: tidal flat and flooded flat. It should be noted that the visual interpretation was implemented on the GEE cloud platform because it archives a large amount of satellites imagery with various time spans and spatiotemporal resolution (Zhang et al., 2022a). Meanwhile, each validation point is independently interpreted by five experts for minimizing the effect of expert's subjective knowledge, and only these complete agreement points were retained otherwise they were discarded. Then, we employed four metrics typically used to evaluate accuracy, which include the kappa coefficient, overall accuracy, user's accuracy (measuring the commission error), and producer's accuracy (measuring the omission error) (Gómez et al., 2016; Olofsson et al., 2014), were calculated using 25709 global wetland validation samples.

[Figure]

Figure 6. The spatial distribution of 25,709 global wetland validation samples using stratified sampling strategy.

Afterwards, the updated confusion matrix has been revised after replenishing 8007 validation points as:

**Table 5**. The confusion matrix of the global 30 m fine wetland map using 25,709 validation points.

| | NWT | PW | SWP | MSH | FFT | SAL | MGV | SMH | TFT | Total | P.A. |
|---|---|---|---|---|---|---|---|---|---|---|---|

| | NWT | PW | SWP | MSH | FFT | SAL | MGV | SMH | TFT | Total | P.A. |
|---|---|---|---|---|---|---|---|---|---|---|---|
| **NWT** | 9950 | 17 | 254 | 224 | 39 | 3 | 12 | 33 | 26 | 10588 | 94.24 |
| **PW** | 69 | 2251 | 4 | 15 | 63 | 0 | 0 | 8 | 9 | 2419 | 93.06 |
| **SWP** | 272 | 5 | 2127 | 452 | 74 | 11 | 3 | 9 | 0 | 2953 | 72.03 |
| **MSH** | 546 | 18 | 135 | 3218 | 149 | 18 | 2 | 34 | 1 | 4121 | 78.09 |
| **FFT** | 145 | 21 | 26 | 95 | 574 | 3 | 1 | 5 | 2 | 872 | 65.83 |
| **SAL** | 26 | 1 | 0 | 43 | 5 | 846 | 0 | 0 | 0 | 921 | 91.86 |
| **MGV** | 65 | 4 | 11 | 2 | 2 | 1 | 1109 | 15 | 3 | 1213 | 91.43 |
| **SMH** | 157 | 15 | 6 | 85 | 9 | 30 | 26 | 998 | 22 | 1347 | 74.09 |
| **TFT** | 78 | 13 | 0 | 11 | 7 | 11 | 6 | 29 | 1150 | 1305 | 88.12 |
| **Total** | 11308 | 2345 | 2563 | 4145 | 922 | 923 | 1159 | 1131 | 1213 | 25709 | |
| **U.A.** | 87.99 | 95.99 | 82.99 | 79.56 | 62.26 | 91.66 | 95.69 | 88.24 | 94.81 | | |
| **O.A.** | | | | | | 86.44 | | | | | |
| **Kappa** | | | | | | 0.822 | | | | | |

**Note**: **NWT**: non-wetlands, **PW:** permanent water, **SWP**: swamp, **MSH**: marsh, **FFT**: flooded flat, **SAL**: saline, **SMH**: salt marsh, **MGV**: mangrove forest, **TFT**: tidal flat, **O.A.**: overall accuracy, **P.A.**: producer's accuracy, **U.A.**: user's accuracy.

I would argue that there exist multiple independent data layers that could be used to provide a much greater assessment of their relative accuracy (perhaps in addition their visual validation). For instance, the Chinese SFA, Canadian CWI, US NWI are all available datasets for validation. Within the US, there's also the National Land Cover Data (e.g., Wickham et al. 2018 that has the contiguous US land cover at 30 m pixel resolution, including both wetlands AND permanent water; https://doi.org/10.1080/01431161.2017.1410298).

Great thanks for the comment. Based on your suggestion, the comparisons at national scale between GWL_FCS30, NWI and NLCD, and CLC databases have been added in the Section 6.2. As for the Canadian CWI and Chinese SFA, we are temporarily unable to obtain sufficient data for comparative analysis and then use the ESA CORINE Land Cover database for another comparative data.

[revised manuscript text omitted]

Lastly, the Discussion section should focus on their position in the data libraries of the world and not have more results within (e.g., why do they have relatively few wetlands versus other global data?).

Great thanks for the comment. Based on this comment and later suggestions, the two results sections about training samples and feature importance have been moved to the Results Section. The update Discussion section focus on analyzing the performance of GWL_FCS30 with other wetland products (including: inland wetland products, coastal wetland products and national wetland databases).

Ultimately, there's excitement and possibility with these data – the inclusion of multiple data layers and stacks in a random forest analysis within the GEE is exciting, especially considering the abundance of spatial data available for analyses. Yet while the authors have presented a welcome analysis, I find they leave enough to be desired to suggest a major revision to a) shorten, b) clarify approaches so that they can be repeated, c)

appropriately and abundantly defend their approach to not include any open waters as a wetland type (which I do not agree with), d) place their findings against other datasets through accuracy analyses (e.g., CWI, NLCD, NWI, etc.) such that readers can determine that this data layer is better to use than those that have come before. We're lacking that confidence at this juncture, at least from my point of view.

Great thanks for the comments and suggestions.

(a) Since we need to fully consider that the method can be repeated at details and some cross-comparisons also are strengthen by other reviewers' comments, we try our best to shorten some redundant statements in the Method (in Section 4.3 Accuracy Assessment), Results (Section 5.2 Importance of multisourced features) and Discussion (Section 6.1 Cross-comparisons with other global products).

(b) Based on your later comments and suggestions, the method has been greatly strengthen, the details have also been added. The specific revisions have been explained in the following comments.

(c) Based on your comment and later suggestions, the permanent water has been added into our fine wetland system, and the detailed replay has been answered in the Comment 1.

(d) As for the comparisons with other dataset (including: CWI, NLCD and NWI), comparisons at national scale between GWL_FCS30, NWI and NLCD, and CLC databases have been added in the Section 6.2 and the accuracy analyses are also added, and the detailed replay has also been answered in the Comment 3.

**Specific comments**

L43 Ramsar is a city in Iran and not an abbreviation to be capitalized.

Thanks for the comment. It has been corrected.

L108: Is there indeed "…no 30-m dataset covering both inland and coastal wetlands" until now? One could argue that the authors introduce ~8 different data layers doing that. For instance, the ESA products, the CCI, etc. Tootchi et al. (2019), referenced in this paper, have Table 1, "Summary of water body, wetland, and related proxy maps and datasets from the literature" that summarize the state of the literature in 2019, too. ESA recently released a worldcover database at 10-m – how does this contrast to the authors' analyses (and ESA includes herbaceous wetlands and mangroves as specific land covers; https://esa-worldcover.org/en ).

Great thanks for pointing out the inaccurate statement. Yes, the statement in the Line 108 is inaccurate, the sentence has been revised as:

"Due to the complicated temporal dynamics and spatial and spectral heterogeneity of wetlands, there is very few global **thematic** wetland dataset covering both inland and coastal regions **with fine classification system and high spatial resolution**, which cause that global 30 m wetland mapping with a fine classification system remains a challenging task."

In addition, as your mentioned, although the ESA WorldCover dataset contains herbaceous wetlands and mangroves, we find that the herbaceous wetlands suffered serious omission errors and the mangrove layer also had lower performance than the global mangrove thematic datasets. Therefore, we give up to use the ESA WorldCover10 dataset to derive our training samples in this study. And the reasons why the WorldCover10 had poor performance in wetland mapping because **their classification algorithms were not specifically designed for the wetland environment**.

Recently, with the improvement of computing power and storage abilities, three global 30-m land-cover products (including GlobeLand30 (Chen et al., 2015), FROM_GLC (Gong et al., 2013) and GLC_FCS30 (Zhang et al., 2021b)) **and several 10-m land-cover products (WorldCover (Zanaga et al., 2021), Dynamic**

**World (Brown et al., 2022) and FROM_GLC10 (Gong et al., 2019))**, containing an independent wetland layers, were produced, **but their classification algorithms were not specifically designed for the wetland environment, so the wetland usually suffered from low accuracy in these products**.

L119 Why 2019-2021? I recognize that the authors ended up with nearly 800,000 LS images, yet since the GEE can handle so much, why stop there? It's not a fault, but the authors should explain why this time period was selected versus any other available time period.

Thanks for the comment. We used the time-series Landsat imagery during 2019-2021 for the nominal year of 2020 for **minimizing the influence of frequent cloud contamination in the tropics and snow and ice in the high latitudes.** The reason why we only used the Landsat imagery during 2019~2021 because they can guarantee the sufficient observation even in the tropics illustrated in Figure 1. The reasons have been added as: "First, all available Landsat imagery during 2019–2021 was obtained for the nominal year of 2020 via the Google Earth Engine platform for minimizing the influence of frequent cloud contamination in the tropics and snow and ice in the high latitudes."

L123 what are saturated pixels? How does CFMask assist that (vs cloud, cloud shadow, and snow)?

Thanks for the comment. The 'saturated pixels' represents these pixels whose surface reflectance exceeds the theoretical value of 1 especially for ETM+ imagery. And the CFmask algorithm has been explained as:

"And these 'bad quality' observations (shadow, cloud, snow, and saturated pixels) in Landsat imagery were masked using CFmask cloud detection method, **which built a series of decision rules, using temperature, spectral variability, brightness and geometric relationship between cloud and shadow, to identify these 'poor quality' pixels and achieved the overall accuracy of 96.4%** (Zhu et al., 2015; Zhu and Woodcock, 2012)"

L124 Which Landsat platforms were used? Which LS satellite data were used? What sort of processing was done on the LS images? Which bands were used? Etc. etc.

Thanks for the comment. The Landsat 7 ETM+ and Landsat 8 OLI imagery are used, and the pro-processing order in the Landsat imagery has been introduced in the manuscript as: 1) atmospheric correction using LaSRC method; 2) masking 'poor quality' observations using Fmask method.

all available Landsat imagery, including Landsat 7 ETM+ and Landsat 8 OLI missions, during 2019–2021 was obtained for the nominal year of 2020 via the Google Earth Engine platform for minimizing the influence of frequent cloud contamination in the tropics and snow and ice in the high latitudes. To minimize the effect of atmosphere, each Landsat image was **atmospherically corrected** to the surface reflectance by the United States Geological Survey using Land Surface Reflectance Code (LaSRC) method (Vermote et al., 2016) and then archived on the GEE platform. And these **'bad quality' observations** (shadow, cloud, snow, and saturated pixels) in Landsat imagery were masked using CFmask cloud detection method, which built a series of decision rules, using temperature, spectral variability, brightness and geometric relationship between cloud and shadow, to identify these 'poor quality' pixels and achieved the overall accuracy of 96.4% (Zhu et al., 2015; Zhu and Woodcock, 2012).

Then, in this study, six optical bands, including: blue, green, red, NIR (near infrared), SWIR1 (Shortwave Infrared 1) and SWIR2 (Shortwave Infrared 2), are used. The supplement information has been added as:

In this study, six optical bands, including: blue, green, red, NIR (near infrared), SWIR1 (shortwave infrared 1) and SWIR2 (shortwave infrared 2) bands, were used for wetland mapping. Totally, 764,239 Landsat scenes, including Landsat 7 ETM+ and Landsat 8 OLI missions, were collected to capture various water-level and phenological features presented in Section 4.

L125 LS images were used to select the "water level" or the presence of inundation as inferred from reflectance values?

Great thanks for the comment. Yes, we used multitemporal compositing method from time-series Landsat imagery to capture the highest water-level and lowest water-level composites according to the spectral characteristics of water body and other land-cover types. It has been detailedly descripted in the Section 4.1, for example, the figure 4 illustrated the presence of inundation status in the Poyang Lake using time-series Landsat imagery.

[Figure]

**Figure 4**. The lowest and highest water-level features derived from (a-b) time-series Landsat optical reflectance data and (c-d) the Sentinel-1 SAR imagery using the time-series compositing method in Poyang Lake, China.

L126 These are not necessarily clear sky, but they are images that passed through the CFMask filter. Please clarify in text.

Great thanks for pointing out the mistake. Yes, **all Landsat imagery** during 2019-2021 were used and then these 'poor quality' pixels would be masked using CFmask method.

The Figure 1 illustrated the availability of clear-sky observations after masking 'poor quality' pixels, namely, we actually count the frequency of these clear observations at each pixel instead of the frequency of Landsat scenes. So, the statement has been revised as:

Figure 1a illustrates the spatial distribution of all clear-sky observations for all Landsat scenes, and it can be seen that there were more than 10 clear observations after masking these 'poor quality' observations at each region even if in the tropics.

L135 How did the authors discern what were sufficient Sentinel-1 images to "capture the temporal dynamics of wetlands"? What are those temporal dynamics? Seasonal? Intermittent inundation from rainstorms? Please clarify in text.

Great thanks for the comment. As Sentinel-1 SAR platform is **immune to the cloud and shadow and has a revisit cycle of 6 days**, the time-series Sentinel-1 imagery in 2020 are sufficient to capture water-level dynamics. The "temporal dynamics" refers to the water-level dynamics. The statement has been revised as:

Figure 1b also illustrates the spatial distribution of all available Sentinel-1 SAR imagery, there were enough Sentinel-1 SAR observations in each area to capture the water-level dynamics of wetlands **because it was immune to the cloud and shadow and had a revisit time of 6 days after launching the Sentinel-1B mission**.

L138 How were the ASTER data used as ancillary information? Please specify how these data on slope, aspect, etc. were used here for the purposes of the paper.

Great thanks for the comment. The elevation, slope and aspect, derived from the ASTER dataset, are the input features to train the random forest models, because many studies have demonstrated that the topography would directly affect the spatial distribution of wetlands, which are mainly distributed in low-lying areas. It has been explained as:

Figure 3 illustrates the flowchart of the proposed method for generating the global 30-m fine wetland maps. First, we combined the time-series Landsat-8, Sentinel-1 SAR observations and ASTER DEM topographical image to derive multisource and multitemporal features including: various water-level, phenological and **three topographical features**. Then, the training samples (coastal tidal, inland wetlands and no-wetlands) and **derived multisource and multitemporal features** were combined to train the stratified random forest classifiers (a classic and widely used machine learning classification model (Breiman, 2001)) at each local region. Next, using the trained random forest models and derived multisource and multitemporal features, we could develop corresponding coastal tidal wetland and inland wetland maps.

As I see later that it was used in the random forest, the authors need to introduce to the readers that a random forest approach is used and conduct a literature review noting the utility of random forest and limitations.

Thanks for the comment. The random forest approach is a classic and widely used machine learning method, it has been reviewed in many studies (Gislason et al., 2006; Belgiu et al., 2016; Boulesteix et al., 2012), so it was not the focus of this article. The disadvantages and disadvantages of random forest are listed below:

The advantages of the random forest has been introduced in the manuscript as: 1) dealing with high-dimensional data, 2) robustness for training noise and feature selection, 3) achieving higher classification when compared to other widely used machine learning classifiers.

Afterward, the random forest (RF) classifier was demonstrated to have obvious advantages including: **dealing with high-dimensional data, robustness for training noise and feature selection, as well as achieving higher classification when compared to other widely used machine learning classifiers (e.g., support vector machines, neural networks, decision trees, etc.)** (Belgiu and Drăguţ, 2016; Gislason et al., 2006).

As for the disadvantages of the RF are: 1) it surely does a good job at classification but not as for regression problem as it does not gives precise continuous nature prediction; 2) it can feel like a black box approach for a statistical modelers we have very little control on what the model does. **However, these two drawbacks can be ignored for land-cover classifications, so it is currently the most popular machine learning algorithm and is widely used in land cover classifications at various scale (region, nation, continent and globe).**

Gislason, P. O., Benediktsson, J. A., and Sveinsson, J. R.: Random Forests for land cover classification, Pattern Recognition Letters, 27, 294-300, https://doi.org/10.1016/j.patrec.2005.08.011, 2006.

Belgiu, M. and Drăguţ, L.: Random forest in remote sensing: A review of applications and future directions, ISPRS Journal of Photogrammetry and Remote Sensing, 114, 24-31, 2016.

Boulesteix, Anne‐Laure, et al. "Overview of random forest methodology and practical guidance with emphasis on computational biology and bioinformatics." Wiley Interdisciplinary Reviews: Data Mining and Knowledge Discovery 2.6 (2012): 493-507.

L142 Figure 1 would be much clearer if it were a vertical panel of a) over b) versus a) next to b). Please modify. Also please change the caption to clarify that that images were not necessarily 'clear sky' but did otherwise pass the CFMask filter. See, e.g., L395.

Great thanks for the suggestion. The layout of the figure has been revised as:

[Figure]

Figure1. The spatial distribution of clear observations after masking these 'poor quality' observations during 2019-2021 (a), and availability of time-series Sentinel-1 SAR observations in 2020 (b).

L165 The JRC_GSW data layer does not identify wetlands per se but identifies inundated pixels. Therefore it is inaccurate to say that the JRC captured "wetlands around rivers, ponds, etc." because the data layer would include rivers and ponds – or any pixel that was deemed to be inundated by the Pekel et al. (2016) algorithm. Please revise to acknowledge these data from Pekel identify inundated pixels.

Great thanks for the comment and suggestion. Yes, the statement in manuscript is inaccurate, and JRC_GSW dataset is used to identify these inundated pixels, so it has been revised as:

The JRC_GSW dynamic water dataset achieved a producer accuracy of 98.5% for these seasonal waters (Pekel et al., 2016) and was used to **identify inundated pixels.**

Note this also comes up with L281 wherein the authors state they are "excluding permanent water bodies". Why? Permanent water bodies are a massive abundance of the global wetland data layers (e.g., in addition to the Ramsar Convention definition used earlier, see also

Davidson, N. C. 2014. How much wetland has the world lost? Long-term and recent trends in global wetland area. Marine and Freshwater Research65: 934-941

Dixon, M. J. R. et al.2016. Tracking global change in ecosystem area: the Wetland Extent Trends index. Biological Conservation 193: 27-35

Hu, S. et al.2017. Global wetlands: Potential distribution, wetland loss, and status. Science of the Total Environment586: 319-327

Thanks for the comment. Based on your useful suggestion, the permanent water has been added in our wetland classification system as:

The inland wetland types shared similar characteristics and were grouped into swamp, marsh, and flooded flat. Meanwhile, in order to capture saline soils and halophytic plant species along saline lakes, the inland saline wetland, inherited from the Global Lakes and Wetlands Dataset (GLWD) (Lehner and Döll, 2004), was also imported. Lastly, the permanent water, including lakes, rivers and streams that are always flooded, was widely identified as a wetland layer in previous studies (Davidson, 2014; Dixon et al., 2016; Hu et al., 2017b).

**Table 2**. The description of wetland classification system in this study

| Category I | Category II | Description |
|---|---|---|
| Tidal wetland | Mangrove | The forest or shrubs which grow in the coastal blackish or saline water |
| | Salt marsh | Herbaceous vegetation (grasses, herbs and low shrubs) in the upper coastal intertidal zone |
| | Tidal flat | The tidal flooded zones between the coastal high and low tide levels including mudflats and sandflats. |
| Inland wetland | Swamp | The forest or shrubs which grow in the inland freshwater |
| | Marsh | Herbaceous vegetation (grasses, herbs and low shrubs) grows in the freshwater |
| | Flooded flat | The non-vegetated flooded areas along the rivers and lakes |
| | Saline | Characterized by saline soils and halophytic (salt tolerant) plant species along saline lakes |
| | **Permanent water** | **Lakes, rivers and streams that are always flooded** |

L169 Table 1 – considering this product is a global data layer, it would be useful to the readers to see the relative abundance of wetlands that each of these named datasets have identified. Furthermore, it's important to note if indeed these are global products (versus near-global products, such as those within the latitudinal bands of 60N and 60S, for instance). Also convert the arc-seconds to meters (at the equator) for consistency between the data products.

Great thanks for the comment. The total area and spatial coverage of these prior wetland datasets has been added and the arc-second unit has been converted to length unit as:

**Table 1**. The characteristics of 13 global wetland products with various spatiotemporal resolutions (unit of area: million $km^2$)

| Dataset name and reference | Wetland categories | Year | Resolution | Total area | Coverage |
|---|---|---|---|---|---|
| World atlas of mangroves (WAM) Spalding (2010) | Mangrove | 2010 | 1:1000000 | 0.152 | Global |
| Global mangrove watch (GWM) Thomas et al. (2017) | | 1996-2016 | ~25m | ~0.136 | Global |
| A global biophysical typology of mangroves (GBTM) Worthington et al. (2020) | | 1996-2016 | ~25m | ~0.136 | Global |
| Continuous global mangrove forest cover (CGMFC) | | 2000-2010 | 30 m | 0.083 | Global |

| | | | | | |
|---|---|---|---|---|---|
| Hamilton and Casey (2016) | | | | | |
| Global distribution of mangroves USGS (GDM_USGS) Giri et al. (2011) | | 2011 | 30 m | ~0.138 | Global |
| Global distribution of tidal flat ecosystems Murray et al. (2019) | Tidal flat | 1984-2016 | 30 m | 0.124~0.132 | 60°S~60°N |
| Global distribution of saltmarsh McOwen et al. (2017) | Salt marsh | 1973-2015 | 1:10,000 | ~0.05 | Global |
| Tropical and subtropical wetland distribution Gumbricht (2015) | Open water, mangrove, swamps, fens, riverine, floodplains, marshes | 2011 | ~231 m | 4.7 | 60°S~40°N |
| Global lakes and wetlands database (GLWD) Lehner and Döll (2004) | Lake, reservoir, river, marsh, swamps, coastal wetland, saline wetland, and peatland | 2004 | ~1 km | 10.7–12.7 | Global |
| JRC-GSW Pekel et al. (2016) | Water | 1984-2021 | 30 m | ~4.46 | Global |
| ESA CCI_LC Defourny et al. (2018) | Swamps, mangrove, and Shrub or herbaceous cover wetlands | 1992-2020 | 300 m | 6.1 | Global |
| GlobeLand30 Chen et al. (2015) | Wetland | 2000-2020 | 30 m | 7.01~7.17 | Global |
| GLC_FCS30 Zhang et al. (2021b) | Wetland | 2015, 2020 | 30 m | 6.36 | Global |

L189 How many of the 18,701 data validation points did NOT have complete agreement between the five validation experts? Noting here that 8,355 points were used to discern amongst the seven classes of wetlands. Relative to the other possible ways to assess their study – and convince people to use it – this number of validation points is very small. Too small, by my assessment.

Great thanks for the comment. Approximately 1/10 validation points (1291points) have been discarded because of the disagreement between five interpreters. Yes, we agree that a large amount of validation points play great role in comprehensively assess the performance of the developed products, however, it should be noted that the collection of validation points, especially for water-level sensitive wetlands with fine classification system, is time-consuming and labor-intensive. In addition, Foody et al. (2009) and Olofsson et al. (2014) had detailedly described how to determine the size of total validation points by using stratified random sampling theory as:

$$n = \frac{\left(\sum W_h \sqrt{p_h(1 - p_h)}\right)^2}{V + \sum W_h P_h(1 - P_h)/N}$$

where $N$ is the number of pixel units in the study region; $V$ is the standard error of the estimated overall accuracy that we would like to achieve, $V = (d/t)^2$ ($t = 1.96$ for a 95% confidence interval, $t = 2.33$ for a 97.5% confidence interval, and $d$ is the desired half-width of the confidence interval); $W_h$ is the weight distribution of class $h$; $p_h$ is the producer's accuracy. These sample size calculations should be repeated for a variety of choices of $V$ and $p_h$ before reaching a final decision. We try to achieve producer's accuracies of 0.9 of non-wetland class and 0.8 of the seven wetland classes. Meanwhile, using the parameters of $d = 0.0125$, $t = 2.33$, the sample size can be determined as approximately 18500. In addition, there is a little uncertainty for interpreting the validation points, so we randomly generate 20000 validation points over the globe and then

discard 1299 uncertain points (these disagreement points over five experts), so a total of 18701 validation points are used to assess the GWL_FCS30-2020 performance.

In order to make the validation assessment more comprehensive, we also replenish 7008 wetland validation points, including 212 non-wetland points and 6796 wetland points, and the description of these updated global validation points (25709 points) has been revised as:

To quantitatively analyze the performance of our GWL_FCS30 wetland map, a total of 25,709 validation samples, including 10,558 non-wetland points and 15,151 wetland points, were collected by combining high-resolution imagery, time-series Landsat and Sentinel observations and visual interpretation method. Firstly, as the wetland was sparse land-cover type compared to the non-wetlands (forest, cropland, grassland and bare land), the stratified random strategy was applied to randomly derive validation points at each strata. Then, as the wetlands had significant correlation with the water levels (Zhang et al., 2022b), the time-series optical observations archived on the GEE cloud platform were used as the auxiliary dataset to interpret these water-level sensitive wetlands such as: tidal flat and flooded flat. It should be noted that the visual interpretation was implemented on the GEE cloud platform because it archives a large amount of satellites imagery with various time spans and spatiotemporal resolution (Zhang et al., 2022a). Meanwhile, each validation point is independently interpreted by five experts for minimizing the effect of expert's subjective knowledge, and only these complete agreement points were retained otherwise they were discarded. Figure 6 intuitively illustrated the spatial distribution of global wetland validation points, it can be found that the distribution of wetland points accurately revealed the spatial patterns of global wetlands.

[Figure]

Figure 6. The spatial distribution of 25,709 global wetland validation samples using stratified sampling strategy.

L207 There are many wetland definitions. That the Ramsar definition is quoted, noting that it includes waters to the depth of 6 m, suggests that open waters should be a wetland type in this analysis. I recognize that flooded flats – located along rivers and lakes – are included. But what of lakes themselves? Ponds? Smaller waters that

are important to the global wetland data layer? Are these considered lakes? This is an important factor to consider when assessing global wetland coverage.

Great thanks for the comment. Based on your suggestion, the open waters have been included in our updated wetland classification system as the "permanent water", which mainly includes lakes, rivers and streams that are always flooded. The revised wetland classification system as:

The inland wetland types shared similar characteristics and were grouped into swamp, marsh, and flooded flat. Meanwhile, in order to capture saline soils and halophytic plant species along saline lakes, the inland saline wetland, inherited from the Global Lakes and Wetlands Dataset (GLWD) (Lehner and Döll, 2004), was also imported. **Lastly, the permanent water, including lakes, rivers and streams that are always flooded, was widely identified as a wetland layer in previous studies (Davidson, 2014; Dixon et al., 2016; Hu et al., 2017b).**

**Table 2**. The description of wetland classification system in this study

| Category I | Category II | Description |
|---|---|---|
| Tidal wetland | Mangrove | The forest or shrubs which grow in the coastal blackish or saline water |
| | Salt marsh | Herbaceous vegetation (grasses, herbs and low shrubs) in the upper coastal intertidal zone |
| | Tidal flat | The tidal flooded zones between the coastal high and low tide levels including mudflats and sandflats. |
| Inland wetland | Swamp | The forest or shrubs which grow in the inland freshwater |
| | Marsh | Herbaceous vegetation (grasses, herbs and low shrubs) grows in the freshwater |
| | Flooded flat | The non-vegetated flooded areas along the rivers and lakes |
| | Saline | Characterized by saline soils and halophytic (salt tolerant) plant species along saline lakes |
| | **Permanent water** | **Lakes, rivers and streams that are always flooded** |

Another consideration would be submergent vegetation. The marsh class is noted as including grasses, herbs, and low shrubs. What about, say, ponds covered with Nymphaea spp (lily pads)? What about Potamogeton spp. growing submersed in the water? Are these not wetland species? Wetland scientists would say they are. Here's a good reference in re: this discussion:

Richardson, D. C., et al. 2022. A functional definition to distinguish ponds from lakes and wetlands. Scientific Reports 12(1): 10472.

Great thanks for the comment. Yes, we agree the submergent vegetation can be considered as a special wetland sub-category, however, the remote sensing observations have poor ability to penetrate water body and then capture these underground vegetation characteristics. Namely, we cannot identify these submergent vegetation at global scale using remote sensing observations, therefore, our future work would pay attention on these special wetland categories, it has been added in the Discussion as:

Then, in this study, we combined the multisourced wetland products and their practical use for ecosystem management to define a fine wetland classification system containing eight sub-categories, however, there are still many wetland sub-categories, such as: submergent vegetation (nymphaea), groundwater-dependent wetlands (karst and cave systems) and seagrass beds (Richardson et al., 2022), cannot be captured because

remote sensing observations usually had poor performance on penetrating water body and then capturing underwater characteristics, and there was currently no prior dataset for global underwater wetlands. So, our further work would pay attention to combine multisourced auxiliary datasets, such as hydrological data, bathymetry depth and climate data, for targeted monitoring these special wetland sub-categories.

L252 Provide a number of LS images used for this analysis.

Thanks for the comment. The total number of Landsat imagery for distinguish salt marsh and tidal flat is 140902, it also added in the manuscript as:

as a tidal flat is a non-vegetated coastal wetland, we combined the empirical rule (EVI $\geqslant$ 0.1, NDVI $\geqslant$ 0.2, and LSWI > 0) proposed by Wang et al. (2020) and time-series Landsat imagery in 2020 **(approximately 142 thousand Landsat scenes)** to exclude all vegetated pixels from tidal flat training samples.

L257 Clarify – 50 km buffer along the coastal zone between 60N – 90N are salt marsh? That seems to be quite excessive, a 50 km buffer. Please clarify.

Great thanks for the comment. The 50 km buffer is only the maximum boundary for tidal flat and salt marsh between 60N – 90N, namely, the both of them are impossible to be outside this buffer area. Actually, we then used the classification method to identify these salt marsh and tidal flat pixels within the region.

In addition, as for the buffer radius of 50 km, it is used in the works of Wang et al. (2020) and (Murray et al., 2019)) for tidal flat mapping.

Wang, X., Xiao, X., Zou, Z., Hou, L., Qin, Y., Dong, J., Doughty, R. B., Chen, B., Zhang, X., Chen, Y., Ma, J., Zhao, B., and Li, B.: Mapping coastal wetlands of China using time series Landsat images in 2018 and Google Earth Engine, ISPRS J Photogramm Remote Sens, 163, 312-326, https://doi.org/10.1016/j.isprsjprs.2020.03.014, 2020.

Murray, N. J., Phinn, S. R., DeWitt, M., Ferrari, R., Johnston, R., Lyons, M. B., Clinton, N., Thau, D., and Fuller, R. A.: The global distribution and trajectory of tidal flats, Nature, 565, 222-225, https://doi.org/10.1038/s41586-018-0805-8, 2019.

The description about the 50 km buffer has been strengthened in the manuscript as:

therefore, we used the coastal shorelines ($Line_{coastal}$) to create a 50 km buffer (applied by the Wang et al. (2020) and (Murray et al., 2019)) as the potential tidal flat zones in the high latitude regions (>60°N) as in Eq. (2). It should be noted that we only identified and retained these tidal flat pixels within the maximum extents by using the classification models in the Section 4.2.

L258 What's the proportion of overlap between the different data layers? A spatial correlation table/matrix should be presented to the readers (see, e.g., Tootchi et al. 2019, supplemental information Table S1).

Thanks for the comment. The overlap proportions of 6 coastal wetland products have been calculated in the Table S1 as:

**Table S1**. The overlap proportions of six coastal wetland products

|  | GDM_USGS | GWM | GBTM | WAM | McOwen's saltmarsh | Murry's tidalflat |
|---|---|---|---|---|---|---|
| GDM_USGS | 1.000 | 0.775 | 0.776 | 0.700 | 0.027 | 0.147 |
| GWM | 0.828 | 1.000 | 0.997 | 0.788 | 0.031 | 0.155 |
| GBTM | 0.825 | 0.992 | 1.000 | 0.787 | 0.032 | 0.154 |

| | | | | | |
|---|---|---|---|---|---|
| WAM | 0.661 | 0.697 | 0.699 | 1.000 | 0.024 | 0.134 |
| McOwen's saltmarsh | 0.073 | 0.081 | 0.082 | 0.071 | 1.000 | 0.151 |
| Murry's tidalflat | 0.153 | 0.152 | 0.152 | 0.148 | 0.057 | 1.000 |

L270 These data were imported…and what was done with them?

Thanks for the comment. How to import the CCI_LC, GLC_FCS30 and GlobeLand30 has been added as:

as the wetland layer in the global land-cover products (GLC_FCS30, GlobeLand30, and CCI_LC) also covered some coastal wetlands, the saline-water wetland layer in the CCI_LC and the wetland data closed to the coastal shorelines in other two products were also imported as supplement when determining the maximum coastal wetland extents.

L296 The GLWD data are at 1 km pixel. How did the authors include 1 km data plus all the 30-m data products? What's the final resolution of these data? Also, what's the proportion of the overlap between them (a spatial correlation table/matrix would be interesting here).

Thanks for the comment. The GLWD, TROP-SUBTROP Wetland, CCI_LC, with spatial resolutions of 231m~1 km, are resampled to 30 m using the nearest neighbor sampling method on the GEE platform, thus, the derived maximum inland wetland extends is the spatial resolution of 30 m.

Specifically, the consistency analysis of five global wetland datasets (TROP-SUBTROP Wetland, GLWD, CCI_LC, GlobeLand30, and GLC_FCS30) and the temporal stability checking for CCI_LC (1992–2020), GlobeLand30 (2000-2020) and GLC_FCS30 (2015-2020) were applied to identify these temporally stable and high cross-consistency wetland points ($P_{inlandWet}^{Tstable,Scons}$). **It should be noted that the coarse wetland products (GLWD, TROP-SUBTROP and CCI_LC) were resampled to 30 m using the nearest neighbor method on the GEE platform.**

Then, overlap proportions of 6 inland wetland products have been calculated in the Table S2 as:

**Table S2**. The overlap proportions of six inland wetland products

| | CIFOR | GLWD | JRC-GSW | CCI_LC | GlobeLand30 | GLC_FCS30 |
|---|---|---|---|---|---|---|
| CIFOR | 1.000 | 0.406 | 0.172 | 0.341 | 0.213 | 0.194 |
| GLWD | 0.105 | 1.000 | 0.186 | 0.343 | 0.234 | 0.215 |
| JRC-GSW | 0.093 | 0.386 | 1.000 | 0.434 | 0.135 | 0.108 |
| CCI_LC | 0.132 | 0.513 | 0.308 | 1.000 | 0.187 | 0.160 |
| GlobeLand30 | 0.223 | 0.957 | 0.223 | 0.487 | 1.000 | 0.817 |
| GLC_FCS30 | 0.231 | 0.992 | 0.235 | 0.496 | 0.897 | 1.000 |

L338 This is the first time that the use of random forest was noted.

Thanks for the comment. The random forest classification model is a classic and widely used machine learning classifier for land-cover mapping. To make readers to understand the random forest, it has been explained as:

Figure 3 illustrates the flowchart of the proposed method for generating the water-level, phenological and three topographical features and producing the global 30-m fine wetland maps using the stratified random forest (**a classic and widely used machine learning classification model (Breiman, 2001)**) modeling strategy.

Breiman, L.: Random Forests, Machine Learning, 45, 5-32, https://doi.org/10.1023/a:1010933404324, 2001.

In L138, I questioned how the ASTER data were used – I suggest revising the methods to introduce the reader early on to the overall approach (i.e., letting them know that the RF algorithm was used).

Great thanks for the comment. The ASTER GDEM elevation and derived slope and aspect were used as auxiliary information for training the random forest classification models, and further used as the auxiliary features for wetland mapping. Based on your suggestion, we briefly introduced the overall approach in Section 4 as:

Figure 3 illustrates the flowchart of the proposed method for generating the global 30-m fine wetland maps. **First, we combined the time-series Landsat-8, Sentinel-1 SAR observations and ASTER DEM topographical image to derive multisource and multitemporal features including: various water-level, phenological and three topographical features. Then, the training samples (coastal tidal, inland wetlands and no-wetlands) and derived multisource and multitemporal features were combined to train the stratified random forest (a classic and widely used machine learning classification model (Breiman, 2001)) models at each local region. Next, using the trained random forest models and derived multisource and multitemporal features, we could develop corresponding coastal tidal wetland and inland wetland maps. Finally, the post-processing step was used to generate the global 30 m fine wetland map in 2020.**

L341 Figure 4 the use of the ASTER DEM includes slope and aspect? Or was slope and DEM used? If the DEM was used, what information within the DEM was used? See, e.g., L400.

Great thanks for pointing out the mistake in the Figure 4. We used three topographical variables (elevation, slope and aspect) derived from the ASTER DEM dataset. The revised figure 4 as:

[Figure]

**Figure 3**. The flowchart of wetland mapping using water-level, phenological and topographical features and a stratified classification strategy.

Furthermore, the Landsat and Sentinel data were used for identifying inundated pixels. NOT for identifying water levels. I recommend changing the heading title in 4.1 as well.

Great thanks for the comment. We used the time-series Landsat imagery to simultaneously capture the water-level composites (the highest and lowest water-level composites) and **multitemporal phenological information** (five temporal percentiles), and used the time-series Sentinel-1 SAR imagery to capture the water-level composites (the highest and lowest water-level composites). Then, the inundated pixels could be identified

by combining the highest and lowest water-level composites in the optical and SAR composites. So, we still think use the "**Generating various water-levels and phenological composites**" might be more suitable.

L393 Ultimately, why were five LS clusters chosen versus three or just the one? Was parsimony considered in the analyses?

Great thanks for the comment. The reasons why we used five percentiles are: 1) the five percentiles had greater performance on capture phenological variability than three and one percentiles, which also suggested by our previous study in Xie et al. (2021); 2) if we used the seasonal compositing method can generate four seasonal composites, we used five percentiles to better capture the phenological variability; 3) these five LS percentiles are used in many phenological-based studies (Hansen et al., 2014; Zhang and Roy, 2017).

This study composited time-series Landsat reflectance bands and four spectral indexes into five percentiles (15th, 30th, 50th, 70th and 85th) **because we wanted to capture as much of the phenological changes in wetlands as possible when comparing to the four seasonal composites (Zhang and Roy, 2017).**

Yes, we consider the parsimony using percentile-based compositing method for capturing phenological variability when comparing with seasonal-based method. It has been explained in the manuscript as:

Azzari and Lobell (2017) quantitatively analyzed the performance of two compositing methods and found that both of them had similar mapping accuracy for land-cover mapping. Meanwhile, the seasonal-based compositing method needed the prior phenological calendar, while the percentile compositing method did not require any prior knowledge or explicit assumptions regarding the timing of the season.

Xie, S.; Liu, L.; Yang, J. Time-Series Model-Adjusted Percentile Features: Improved Percentile Features for Land-Cover Classification Based on Landsat Data. Remote Sens. 2020, 12, 3091. https://doi.org/10.3390/rs12183091

Hansen, M. C., Egorov, A., Potapov, P. V., Stehman, S. V., Tyukavina, A., Turubanova, S. A., Roy, D. P., Goetz, S. J., Loveland, T. R., Ju, J., Kommareddy, A., Kovalskyy, V., Forsyth, C., and Bents, T.: Monitoring conterminous United States (CONUS) land cover change with Web-Enabled Landsat Data (WELD), Remote Sensing of Environment, 140, 466-484, https://doi.org/10.1016/j.rse.2013.08.014, 2014.

Zhang, H. K. and Roy, D. P.: Using the 500 m MODIS land cover product to derive a consistent continental scale 30 m Landsat land cover classification, Remote Sensing of Environment, 197, 15-34, https://doi.org/10.1016/j.rse.2017.05.024, 2017.

L404 This assumes that the maximum extent of the coastal wetlands (equation 1 for mangroves) has zero omission error. I understand why this was done, yet it requires explanation and accounting for the reader here and possibly in the Discussion section as well.

Thanks for the suggestion. The assumption has been added as:

Since we have simultaneously extracted the maximum coastal and inland wetland extents when deriving training samples from prior wetland datasets, the stratified classification strategy was adopted to fully use the maximum extent constraint. Namely, if a pixel was classified as a coastal wetland outside the maximum coastal wetland extents, it would be identified as a misclassification. I**n other words, we assumed there was zero omission error for these derived maximum wetland extents in Eq. (1-3) by merging several prior wetland products**. In the Discussion Section, the maximum extents of the inland and coastal wetlands have also been added and discussed as:

In addition, we used the derived maximum extents as the boundary for identifying inland and coastal tidal wetlands, in other words, we assumed that the derived maximum extents contained all inland and coastal wetlands with zero omission error. Actually, the inland maximum extents in Eq. (3) fulfilled the assumption (zero omission error), because the GLWD and TROP-SUBTROP products, produced by the compilation and model simulation method (Gumbricht, 2015; Lehner and Döll, 2004), can capture almost all wetland areas at the expense of a higher commission error. For example, the Figure 13 illustrated the cross-comparisons between our GWL_FCS30 wetland maps with four existing wetland products, and the GLWD obviously overestimated the inland wetlands. On the other hand, the union of five global wetland datasets in Eq. (3) also minimized the omission error of each dataset for inland wetland sub-categories. As for the mangrove forest, due to the high producer and user accuracies of five prior mangrove products (explained in Section 2.2), the derived maximum mangrove extents (Eq. (1)) can covered almost all mangrove forests because the missed mangroves maybe complemented by other products. Recently, Bunting et al. (2022) developed the newest mangrove products covering 1996-2020, it can be used as the important prior dataset in our further works for deriving the maximum mangrove extents. However, the zero omission error assumption maybe run into problem when targeting tidal flat and saltmarsh. Specifically, the global tidal flat dataset only covered the period of 1984~2016 and the producer's accuracy of tidal flat was 83.0%. Although we used the union operations for time-series Murray's tidal flats during 1984~2016 (Eq. (2)) to include these potential tidal flats, the newly increased tidal flats during 2016-2020 and missed tidal flats in time-series products would be missed in our maximum tidal flat extents in Eq. (2). Fortunately, the new time-series global tidal flat products during 1999-2019 (Murray et al., 2022), which greatly improved the mapping accuracy based on previous time-series tidal flat products, can be used as prior datasets. Lastly, as the global saltmarsh products were sparse, the maximum extents of tidal flat salt marsh were combined for saltmarsh mapping in Section 3.1. However, there was still missed a lot of saltmarshes, so our further work would pay more attention on accurately saltmarsh mapping.

L410 The local adaptive modeling section is too quickly glossed over. Explain more on how this was done. How were the data trained? What were the specifications of the training here? It would be hard for others to replicate the process based on the data provided thus far.

Great thanks for pointing out the problem. The description of the local adaptive modeling has been greatly strengthen as:

Since we have simultaneously extracted the maximum coastal and inland wetland extents when deriving training samples from prior wetland datasets, the stratified classification strategy was adopted to fully use the maximum extent constraint. Namely, if a pixel was classified as a coastal wetland outside the maximum coastal wetland extents, it would be identified as a misclassification. In other words, we assumed there was zero omission error for these derived maximum wetland extents in Eq. (1-3) by merging several prior wetland products. Furthermore, there were two ideas for the large-area land-cover mapping including global classification modeling (using one universal model for the whole areas) and local adaptive modeling (using various models for different local zones) (Zhang et al., 2020). For example, Zhang and Roy (2017) demonstrated that local adaptive modeling outperformed the global classification modeling strategy. Therefore, the global land surface was first divided into 961 $5° \times 5°$ geographical tiles illustrated in Figure 5, which were inherited from the global 30 m land-cover mapping by (Zhang et al., 2021b). Then, we trained the local adaptive classification models using derived training samples in Section 3 and multisource and multitemporal features (the highest, lowest water-level and phenological composites and topographical variables) at each $5° \times 5°$ geographical tile. It should be noted that

we used the training samples from neighboring 3 × 3 geographical tiles to train the classification model and classify the central tile for guaranteeing the spatially continuous transition over adjacent regional wetland maps. Namely, we trained 961 local adaptive classification models and then produced 961 5° × 5° wetland maps. Finally, we spatially mosaiced these 961 regional wetland maps into the global 30 m fine wetland map in 2020.

[Figure]

Figure 5. The spatial distribution of 961 5° × 5° geographical tiles used for local adaptive modeling, which was inherited from the global 30 m land-cover mapping by (Zhang et al., 2021b). The background imagery came from the National Aeronautics and Space Administration (https://visibleearth.nasa.gov, last access: 10 Nov 2022).

How many of the 961 5x5 tiles had zero coverage of wetlands (e.g., mid-ocean tiles)?

Thanks for the comment and interesting question. According to our statistics, there was 41 5x5 tiles had zero coverage of wetlands.

L413 What statistical program was used to conduct the RF analyses? Furthermore, while RF may have advantageous, it also has detractions. Please introduce the "obvious advantageous" for those who are not aware as well as mention some of the drawbacks.

Thanks for the comment. The RF analysis has been conducted on the GEE platform.

Therefore, the RF classifier was selected for mapping inland and coastal tidal wetlands using multi-sourced features **on the GEE platform**.

The advantages of the RF have been listed in the manuscript as: 1) dealing with high-dimensional data, 2) robustness for training noise and feature selection, 3) achieving higher classification when compared to other widely used machine learning classifiers.

Afterward, the random forest (RF) classifier was demonstrated to have obvious advantages including: **dealing with high-dimensional data, robustness for training noise and feature selection, as well as achieving higher classification when compared to other widely used machine learning classifiers** (e.g., support vector machines, neural networks, decision trees, etc.) (Belgiu and Drăguţ, 2016; Gislason et al., 2006).

As for the disadvantages of the RF are: 1) it surely does a good job at classification but not as for regression problem as it does not gives precise continuous nature prediction; 2) it can feel like a black box approach for a statistical modelers we have very little control on what the model does. **However, these two drawbacks can be ignored for land-cover classifications, so it is currently the most popular machine learning algorithm and is widely used in land cover classifications at various scale (region, nation, continent and globe).**

L435 Note that 18k samples were analyzed across the globe. Consider the relative dearth noted in Figure 2 (see summary above).

Great thanks for the comment. Yes, a large amount of validation points can more comprehensively evaluate the performance of developed GWL_FCS30 dataset. However, as mentioned before, the size of validation points in this study is determined by using the stratified random sampling theory proposed by the Foody et al. (2009) and Olofsson et al. (2014) as:

$$ n = \frac{\left( \sum W_h \sqrt{p_h(1 - p_h)} \right)^2}{V + \sum W_h P_h (1 - P_h)/N} $$

where $N$ is the number of pixel units in the study region; $V$ is the standard error of the estimated overall accuracy that we would like to achieve, $V = (d/t)^2$ ($t = 1.96$ for a 95% confidence interval, $t = 2.33$ for a 97.5% confidence interval, and $d$ is the desired half-width of the confidence interval); $W_h$ is the weight distribution of class $h$; $p_h$ is the producer's accuracy. These sample size calculations should be repeated for a variety of choices of $V$ and $p_h$ before reaching a final decision. We try to achieve producer's accuracies of 0.9 of non-wetland class and 0.8 of the seven wetland classes. Meanwhile, using the parameters of $d = 0.0125$, $t = 2.33$, the sample size can be determined as approximately 18500.

Based on your suggestion, we also replenish 7008 wetland validation points, including 212 non-wetland points and 6796 wetland points, so the updated global validation dataset contains 25709 validation points.

L461 The authors need to introduce Figures 7-10 before introducing Figure 11.

Thanks for the comment. Yes, we first introduce Figure 7-10 and then introduce Figure 11.

L538 The point behind the Ramsar Convention's of 6 m was to address depths that diving birds were expected/known to use aquatic systems. It is disingenuous to state that all permanent water bodies have depths >=6 m. This is a possibly fatal flaw in this analysis.

Great thanks for pointing out the issue. The statement has been removed in the revised manuscript, and there was no water depth database derived from remote sensing imagery until now, so the permanent water bodies are also included in the updated GWL_FCS30 products. The statement has been revised as:

To comprehensively understand the performance of the GWL_FCS30 wetland maps, four existing global wetland datasets (GLC_FCS30, GlobeLand30, CCI_LC, and GLWD), listed in Table 1, were selected. Figure 12 quantitatively illustrates the total wetland area of five products over each continent. Specifically, the total wetland area of different wetland products varied. The GLWD obviously overestimated the wetland area on each continent mainly because it was derived from the compilation model instead of actual remote sensing observations (Lehner and Döll, 2004). Namely, the GLWD classified a large amount of non-wetlands as potential wetlands. The remaining four wetland products, derived from the Landsat and PROBE-V remote sensing imagery, shared a total wetland area of 4.128~7.364 million km², and our GWL_FCS30 wetland dataset

had the total area of 6.347 million km$^2$ among these datasets. The CCI LC wetland layer contained the smallest wetland area of 4.128 million km2, and the estimated area in North America was profoundly lower than the other datasets, mainly because the CCI LC heavily underestimated the wetland distribution in Canada after a comparison with the Canadian Wetland Inventory (Amani et al., 2019). Next, the total wetland area in GlobeLand30 and GLC_FCS30 wetland layer was higher than the developed GWL_FCS30 wetland dataset because some water-level sensitive non-wetlands (such as: irrigated cropland) were also captured in these two datasets.

L555 It would be good to see the analyses done in Table 4 for these two areas shown in Figure 10. For instance, the authors have chosen to not include permanent water as a wetland type but yet show 'water body' in their panel map, which implies it was correctly mapped yet it is not a land use type they map.

Great thanks for the comment. Based on your suggestion in previous comments, the 'permanent water body' has been added into our fine classification system.

L576 Figure 10 the panel caption for GWL_FCS30 doesn't match the panel (GWM_FCS30).

Great thanks for pointing out the mistake. The mistake has been corrected.

L630 These selected training sample results should be in the Results section, not here.

Great thanks for the suggestion. This section has moved to the Results Section 5.1.

L634 Was this inclusion of steps noted in the methods? I don't recall it.

Great thanks for the comment. The reliability analysis of the training samples was not included in the method, because the specific processing flow has been explained in this section as:

To demonstrate the reliability of the derived training samples for wetland mapping, we randomly selected approximately 10,000 points from the sample pool and checked their confidence using visual interpretation. It should be noted that we cannot check all the training samples because the number of derived samples was massive (exceeding 20 million training samples in Section 3). After a point-to-point inspection, these selected training samples achieved an overall accuracy of 91.53% in 2020. Meanwhile, we also used 10,000 selected wetland training samples and many non-wetland samples to analyze overall and producer's accuracies of coastal and inland wetlands versus number of erroneous training samples. Specifically, we gradually increased the "contaminated" samples by randomly altering the label of a certain percentage of training samples in steps of 0.01, and then used these "contaminated" samples to build the RF classification model.

L675 These are results and need to be in that section explaining the outcomes of the RF analysis.

Great thanks for the suggestion. This section has moved to the Results Section 5.2.

---

## Author Response (AR2)

Dear Topical Editor and Reviewers:

On behalf of my co-authors, we thank you very much for reviewing our manuscript and giving us a lot of useful comments and suggestions. We appreciate the comments on our manuscript entitled "GWL_FCS30: global 30 m wetland map with fine classification system using multi-sourced and time-series remote sensing imagery in 2020" (essd-2022-180).

We have revised the manuscript carefully according to the comments. All the changes were highlighted (red color) in the manuscript. And the point-by-point response to the comments of the reviewers is also listed below.

Looking forward to hearing from you soon.

Best regards,

Prof. Liangyun Liu

liuly@radi.ac.cn

Institute of Remote Sensing and Digital Earth, Chinese Academy of Sciences

No.9 Dengzhuang South Road, Haidian District, Beijing 100094, China

**Response to comments**

**Paper #:** essd-2022-180
**Title:** GWL_FCS30: global 30 m wetland map with fine classification system using multi-sourced and time-series remote sensing imagery in 2020
**Journal**: Earth System Science Data

**Reviewer #1**

The authors have made substantial improvement on the MS. I agree on the comments from the other reviewers. For the current version, further accuracy assessment could be present by a spatial map documenting the accuracy for each classification block.

Great thanks for the comment. The spatial map documenting the accuracy for each classification block has been illustrated as:

[Figure]

Figure S1. The spatial map documenting the overall accuracy for each 5°×5° spatial tile.

Although mangrove is the dominant type of coastal swamp, I still disagree on the statement that coastal wetland do not conclude other swamp type beyond the mangroves. Over the low latitudes, there are a lot of coastal swamps but they are not mangroves. Currently, the image data over middle- or latitudes are still limited. Impacts from these factors should be discussed more.

Great thanks for the comment and constructive suggestion. It's our mistake that neglecting the coastal swamp besides mangroves, so our further work would pay more attention to accurately capture these coastal swamps. The discussion about the coastal swamp has been added as:

Meanwhile, some coastal swamps (except for mangrove), which were usually overlooked at most coastal wetland mapping (Murray et al., 2022; Zhang et al., 2022), were also missed in the GWL_FCS30, mainly

because there are no global or large-area coastal swamp dataset can be imported and the coastal swamp is also sparser than the mangrove forest in the low and middle latitudes. So, our further work will pay more attention to combine multisourced auxiliary datasets, such as hydrological data, bathymetry depth and climate data, to map these special wetland sub-categories in a targeted manner.

**Reviewer #2**

The authors have addressed all my concerns. I recommend the publishment of this study, but there are still two minor issues:

Great thanks for the positive affirmation. The manuscript has been further improved based on your and another Reviewer's comments.

1) The validation sample size seems inconsistent between the revised manuscript and the response letter. In the response letter, the authors described the determination of the validation sample size in detail, and the size is 18701; but in the revised manuscript, the number has been changed to 25709, and the determination process is still lacking (the manuscript should refer roughly to the determining of the number).

Great thanks for the comment. Yes, it is inconsistent for validation sample size between the revised manuscript and original manuscript. Specifically, we use the stratified sampling theory to determine the sample size of 18701, however, the Reviewer #3 thinks that the sample size especially for wetlands is sparse. So, approximately 7000 validation points have been added.

Then, as for the determination process of sample size has been added in the manuscript as:

To quantitatively analyze the performance of our GWL_FCS30 wetland map, a total of 25,709 validation samples (illustrated in Figure 6), including 10,558 non-wetland points and 15,151 wetland points, were collected. Firstly, as the wetland was sparse land-cover type compared to the non-wetlands (forest, cropland, grassland and bare land), the stratified random strategy was applied to randomly derive validation points at each strata as:

$$n_i = n \times \frac{W_i \times p_i(1-p_i)}{\sum W_i \times p_i(1-p_i)}; \quad n = \frac{\left(\sum W_i \sqrt{p_i(1-p_i)}\right)^2}{V + \sum W_i p_i(1-p_i)/N} \quad (5)$$

where $W_i$ and $p_i$ are the area proportion and expected accuracy of class $i$, $n_i$ and $n$ are the sample size of class $i$ and total sample size, $V$ is the standard error of the estimated overall accuracy, and $N$ is the number of pixel units in the study region. Then, as the wetlands had significant correlation with the water levels (Zhang et al., 2022b), the time-series optical observations archived on the GEE cloud platform were used as the auxiliary dataset to interpret these water-level sensitive wetlands such as: tidal flat and flooded flat.

2) A typo: Line 494, "coastal pones" should be "coastal ponds".

Great thanks for the comment. The mistake has been corrected in the revised manuscript.

[revised manuscript text omitted]